# Implicit Bias in Leaky ReLU Networks Trained on High-Dimensional Data

**Spencer Frei**[*]
UC Berkeley
frei@berkeley.edu

**Gal Vardi**[*]
TTI Chicago and Hebrew University
galvardi@ttic.edu

**Peter L. Bartlett**
UC Berkeley and Google
peter@berkeley.edu

**Nathan Srebro**
TTI Chicago
nati@ttic.edu

**Wei Hu**
University of Michigan
vvh@umich.edu

## Abstract

The implicit biases of gradient-based optimization algorithms are conjectured to be a major factor in the success of modern deep learning. In this work, we investigate the implicit bias of gradient flow and gradient descent in two-layer fully-connected neural networks with leaky ReLU activations when the training data are nearly-orthogonal, a common property of high-dimensional data. For gradient flow, we leverage recent work on the implicit bias for homogeneous neural networks to show that asymptotically, gradient flow produces a neural network with rank at most two. Moreover, this network is an $\ell_2$-max-margin solution (in parameter space), and has a linear decision boundary that corresponds to an approximate-max-margin linear predictor. For gradient descent, provided the random initialization variance is small enough, we show that a single step of gradient descent suffices to drastically reduce the rank of the network, and that the rank remains small throughout training. We provide experiments which suggest that a small initialization scale is important for finding low-rank neural networks with gradient descent.

## 1 Introduction

Neural networks trained by gradient descent appear to generalize well in many settings, even when trained without explicit regularization. It is thus understood that the usage of gradient-based optimization imposes an implicit bias towards particular solutions which enjoy favorable properties. The nature of this implicit regularization effect—and its dependence on the structure of the training data, the architecture of the network, and the particular gradient-based optimization algorithm—is thus a central object of study in the theory of deep learning.

In this work, we examine the implicit bias of gradient descent when the training data is such that the pairwise correlations $|\langle x_i, x_j \rangle|$ between distinct samples $x_i, x_j \in \mathbb{R}^d$ are much smaller than the squared Euclidean norms of each sample: that is, the samples are nearly-orthogonal. As we shall show, this property is often satisfied when the training data is sampled i.i.d. from a $d$-dimensional distribution and $d$ is significantly larger than the number of samples $n$. We will thus refer to such training data with the descriptors 'high-dimensional' and 'nearly-orthogonal' interchangeably.

We consider fully-connected two-layer networks with $m$ neurons where the first layer weights are trained and the second layer weights are fixed at their random initialization. If we denote the first-layer weights by $W \in \mathbb{R}^{m \times d}$, with rows $w_j^\top \in \mathbb{R}^d$, then the network output is given by,

$$f(x; W) := \sum_{j=1}^m a_j \phi(\langle w_j, x \rangle),$$

where $a_j \in \mathbb{R}$, $j = 1, \ldots m$ are fixed. We consider the implicit bias in two different settings: gradient flow, which corresponds to gradient descent where the step-size tends to zero, and standard gradient descent.

---

[*]Equal contribution.

For gradient flow, we consider the standard leaky ReLU activation, $\phi(z) = \max(\gamma z, z)$. Our starting point in this setting is recent work by Lyu & Li (2019); Ji & Telgarsky (2020) that show that, provided the network interpolates the training data at some time, gradient flow on homogeneous networks, such as two-layer leaky ReLU networks, converges (in direction) to a network that satisfies the Karush–Kuhn–Tucker (KKT) conditions for the margin-maximization problem,

$$\min_W {}^{1/2} \|W\|_F^2 \quad \text{s.t.} \quad \forall i \in [n], \; y_i f(x_i; W) \geq 1 .$$

Leveraging this, we show that the asymptotic limit of gradient flow produces a matrix $W$ which is a global optimum of the above problem, and has rank at most 2. Moreover, we note that our assumption on the high-dimensionality of the data implies that it is linearly separable. Our leaky ReLU network $f(\cdot; W)$ is non-linear, but we show that gradient flow converges in direction to $W$ such that the decision boundary is linear, namely, there exists $z \in \mathbb{R}^d$ such that for all $x$ we have $\operatorname{sign}(f(x; W)) = \operatorname{sign}(z^\top x)$. This linear predictor $z$ may not be an $\ell_2$-max-margin linear predictor, but it maximizes the margin approximately (see details in Theorem 3.2).

For gradient descent, we consider a smoothed approximation to the leaky ReLU activation, and consider training that starts from a random initialization with small initialization variance. Our result for gradient flow on the standard leaky ReLU activation suggests that gradient descent with small-enough step size should eventually produce a network for which $W^{(t)}$ has small rank. However, the asymptotic characterization of trained neural networks in terms of KKT points of a margin-maximization problem relies heavily upon the infinite-time limit. This leaves open what happens in finite time. Towards this end, we consider the stable rank of the weight matrix $W^{(t)}$ found by gradient descent at time $t$, defined as $\|W^{(t)}\|_F^2 / \|W^{(t)}\|_2^2$, the square of the ratio of the Frobenius norm to the spectral norm of $W^{(t)}$. We show that after the first step of gradient descent, the stable rank of the weight matrix $W^{(t)}$ reduces from something that is of order $\min(m, d)$ to that which is at most an absolute constant, independent of $m$, $d$, or the number of samples. Further, throughout the training trajectory the stable rank of the network is never larger than some absolute constant.

We conclude by verifying our results with experiments. We first confirm our theoretical predictions for binary classification problems with high-dimensional data. We then consider the stable rank of two-layer networks trained by SGD for the CIFAR10 dataset, which is not high-dimensional. We notice that the scale of the initialization plays a crucial role in the stable rank of the weights found by gradient descent: with default TensorFlow initialization, the stable rank of a network with $m = 512$ neurons never falls below 74, while with a smaller initialization variance, the stable rank quickly drops to 3.25, and only begins to increase above 10 when the network begins to overfit.

### RELATED WORK

**Implicit bias in neural networks.** The literature on the implicit bias in neural networks has rapidly expanded in recent years, and cannot be reasonably surveyed here (see Vardi (2022) for a survey). In what follows, we discuss results which apply to two-layer ReLU or leaky ReLU networks in classification settings.

By Lyu & Li (2019) and Ji & Telgarsky (2020), homogeneous neural networks (and specifically two-layer leaky ReLU networks, which are the focus of this paper) trained with exponentially-tailed classification losses converge in direction to a KKT point of the maximum-margin problem. Our analysis of the implicit bias relies on this result. We note that the aforementioned KKT point may not be a global optimum (see a discussion in Section 3).

Lyu et al. (2021) studied the implicit bias in two-layer leaky ReLU networks trained on linearly separable and symmetric data, and showed that gradient flow converges to a linear classifier which maximizes the $\ell_2$ margin. Note that in our work we do not assume that the data is symmetric, but we assume that it is nearly orthogonal. Also, in our case we show that gradient flow might converge to a linear classifier that does not maximize the $\ell_2$ margin. Sarussi et al. (2021) studied gradient flow on two-layer leaky ReLU networks, where the training data is linearly separable. They showed convergence to a linear classifier based on an assumption called *Neural Agreement Regime (NAR)*: starting from some time point, all positive neurons (i.e., neurons with a positive outgoing weight) agree on the classification of the training data, and similarly for the negative neurons. However, it is unclear when this assumption holds a priori.

Chizat & Bach (2020) studied the dynamics of gradient flow on infinite-width homogeneous two-layer networks with exponentially-tailed losses, and showed bias towards margin maximization w.r.t.

a certain function norm known as the variation norm. Phuong & Lampert (2020) studied the implicit bias in two-layer ReLU networks trained on orthogonally separable data (i.e., where for every pair of labeled examples $(x_i, y_i), (x_j, y_j)$ we have $x_i^\top x_j > 0$ if $y_i = y_j$ and $x_i^\top x_j \leq 0$ otherwise). Safran et al. (2022) proved implicit bias towards minimizing the number of linear regions in univariate two-layer ReLU networks. Implicit bias in neural networks trained with nearly-orthogonal data was previously studied in Vardi et al. (2022). Their assumptions on the training data are similar to ours, but they consider ReLU networks and prove bias towards non-robust networks. Their results do not have any clear implications for our setting.

Implicit bias towards rank minimization was also studied in several other papers. Ji & Telgarsky (2018; 2020) showed that in linear networks of output dimension 1, gradient flow with exponentially-tailed losses converges to networks where the weight matrix of every layer is of rank 1. Timor et al. (2022) showed that the bias towards margin maximization in homogeneous ReLU networks may induce a certain bias towards rank minimization in the weight matrices of sufficiently deep ReLU networks. Finally, implicit bias towards rank minimization was also studied in regression settings. See, e.g., Arora et al. (2019); Razin & Cohen (2020); Li et al. (2020); Timor et al. (2022); Ergen & Pilanci (2021; 2020).

**Training dynamics of neural networks for linearly separable training data.** A series of works have explored the training dynamics of gradient descent when the data is linearly separable (such as is the case when the input dimension is larger than the number of samples, as we consider here). Brutzkus et al. (2017) showed that in two-layer leaky ReLU networks, SGD on the hinge loss for linearly separable data converges to zero loss. Frei et al. (2021) showed that even when a constant fraction of the training labels are corrupted by an adversary, in two-layer leaky ReLU networks, SGD on the logistic loss produces neural networks that have generalization error close to the label noise rate. As we mentioned above, both Lyu et al. (2021) and Sarussi et al. (2021) considered two-layer leaky ReLU networks trained by gradient-based methods on linearly separable datasets. Wang et al. (2019) and Yang et al. (2021) considered the dynamics of variants of GD/SGD algorithms on the hinge loss for ReLU networks for linearly separable distributions.

Another line of work has explored the dynamics of neural network training when the data is sampled i.i.d. from a distribution which is not linearly separable but the training data is linearly separable due to the number of samples being smaller than the input dimension. Cao et al. (2022) studied two-layer convolutional networks trained on an image-patch data model and showed how a low signal-to-noise ratio can result in harmful overfitting, while a high signal-to-noise ratio allows for good generalization performance. Shen et al. (2022) considered a similar image-patch signal model and studied how data augmentation can improve generalization performance of two-layer convolutional networks. Frei et al. (2022a) showed that two-layer fully connected networks trained on high-dimensional mixture model data can exhibit a 'benign overfitting' phenomenon. Frei et al. (2022b) studied the feature-learning process for two-layer ReLU networks trained on noisy 2-xor clustered data and showed that early-stopped networks can generalize well even in high-dimensional settings. Boursier et al. (2022) studied the dynamics of gradient flow on the squared loss for two-layer ReLU networks with orthogonal inputs.

## 2 PRELIMINARIES

**Notation.** For a vector $x$ we denote by $\|x\|$ the Euclidean norm. For a matrix $W$ we denote by $\|W\|_F$ the Frobenius norm, and by $\|W\|_2$ the spectral norm. We denote by $\mathbb{1}[\cdot]$ the indicator function, for example $\mathbb{1}[t \geq 5]$ equals 1 if $t \geq 5$ and 0 otherwise. We denote $\text{sign}(z) = 1$ for $z > 0$ and $\text{sign}(z) = -1$ otherwise. For an integer $d \geq 1$ we denote $[d] = \{1, \ldots, d\}$. We denote by $\mathsf{N}(\mu, \sigma^2)$ the Gaussian distribution. We denote the maximum of two real numbers $a, b$ as $a \vee b$, and their minimum as $a \wedge b$. We denote by $\log$ the logarithm with base $e$. We use the standard $O(\cdot)$ and $\Omega(\cdot)$ notation to only hide universal constant factors, and use $\tilde{O}(\cdot)$ and $\tilde{\Omega}(\cdot)$ to hide poly-logarithmic factors in the argument.

**Neural networks.** In this work we consider depth-2 neural networks, where the second layer is fixed and only the first layer is trained. Thus, a neural network with parameters $W$ is defined as

$$f(x; W) = \sum_{j=1}^m a_j \phi(w_j^\top x) \,,$$

where $x \in \mathbb{R}^d$ is an input, $W \in \mathbb{R}^{m \times d}$ is a weight matrix with rows $w_1^\top, \ldots, w_m^\top$, the weights in the second layer are $a_j \in \{\pm 1/\sqrt{m}\}$ for $j \in [m]$, and $\phi : \mathbb{R} \to \mathbb{R}$ is an activation function. We focus on

the leaky ReLU activation function, defined by $\phi(z) = \max\{z, \gamma z\}$ for some constant $\gamma \in (0, 1)$, and on a smooth approximation of leaky ReLU (defined later).

**Gradient descent and gradient flow.** Let $S = \{(x_i, y_i)\}_{i=1}^n \subseteq \mathbb{R}^d \times \{\pm 1\}$ be a binary-classification training dataset. Let $f(\cdot; W) : \mathbb{R}^d \to \mathbb{R}$ be a neural network parameterized by $W$. For a loss function $\ell : \mathbb{R} \to \mathbb{R}$ the *empirical loss* of $f(\cdot; W)$ on the dataset $S$ is

$$\widehat{L}(W) := \tfrac{1}{n} \sum_{i=1}^n \ell(y_i f(x_i; W)) \,.$$

We focus on the exponential loss $\ell(q) = e^{-q}$ and the logistic loss $\ell(q) = \log(1 + e^{-q})$.

In *gradient descent*, we initialize $[W^{(0)}]_{i,j} \overset{\text{i.i.d.}}{\sim} \mathsf{N}(0, \omega_{\text{init}}^2)$ for some $\omega_{\text{init}} \geq 0$, and in each iteration we update

$$W^{(t+1)} = W^{(t)} - \alpha \nabla_W \widehat{L}(W^{(t)}) \,,$$

where $\alpha > 0$ is a fixed step size.

*Gradient flow* captures the behavior of gradient descent with an infinitesimally small step size. The trajectory $W(t)$ of gradient flow is defined such that starting from an initial point $W(0)$, the dynamics of $W(t)$ obeys the differential equation $\frac{dW(t)}{dt} = -\nabla_W \widehat{L}(W(t))$. When $\widehat{L}(W)$ is non-differentiable, the dynamics of gradient flow obeys the differential equation $\frac{dW(t)}{dt} \in -\partial^\circ \widehat{L}(W(t))$, where $\partial^\circ$ denotes the *Clarke subdifferential*, which is a generalization of the derivative for non-differentiable functions (see Appendix A for a formal definition).

## 3 ASYMPTOTIC ANALYSIS OF THE IMPLICIT BIAS

In this section, we study the implicit bias of gradient flow in the limit $t \to \infty$. Our results build on a theorem by Lyu & Li (2019) and Ji & Telgarsky (2020), which considers the implicit bias in homogeneous neural networks. Let $f(x; \theta)$ be a neural network parameterized by $\theta$, where we view $\theta$ as a vector. The network $f$ is *homogeneous* if there exists $L > 0$ such that for every $\beta > 0$ and $x, \theta$ we have $f(x; \beta\theta) = \beta^L f(x; \theta)$. We say that a trajectory $\theta(t)$ of gradient flow *converges in direction* to $\theta^*$ if $\lim_{t\to\infty} \frac{\theta(t)}{\|\theta(t)\|} = \frac{\theta^*}{\|\theta^*\|}$. Their theorem can be stated as follows.

**Theorem 3.1** (Paraphrased from Lyu & Li (2019); Ji & Telgarsky (2020))**.** *Let $f$ be a homogeneous ReLU or leaky ReLU neural network parameterized by $\theta$. Consider minimizing either the exponential or the logistic loss over a binary classification dataset $\{(x_i, y_i)\}_{i=1}^n$ using gradient flow. Assume that there exists time $t_0$ such that $\widehat{L}(\theta(t_0)) < \frac{\log(2)}{n}$. Then, gradient flow converges in direction to a first order stationary point (KKT point) of the following maximum-margin problem in parameter space:*

$$\min_\theta \, \tfrac{1}{2} \|\theta\|^2 \quad s.t. \quad \forall i \in [n] \; y_i f(x_i; \theta) \geq 1 \,.$$

*Moreover, $\widehat{L}(\theta(t)) \to 0$ and $\|\theta(t)\| \to \infty$ as $t \to \infty$.*

We focus here on depth-2 leaky ReLU networks where the trained parameters is the weight matrix $W \in \mathbb{R}^{m \times d}$ of the first layer. Such networks are homogeneous (with $L = 1$), and hence the above theorem guarantees that if there exists time $t_0$ such that $\widehat{L}(W(t_0)) < \frac{\log(2)}{n}$, then gradient flow converges in direction to a KKT point of the problem

$$\min_W \, \tfrac{1}{2} \|W\|_F^2 \quad \text{s.t.} \quad \forall i \in [n] \; y_i f(x_i; W) \geq 1 \,. \tag{1}$$

Note that in leaky ReLU networks Problem (1) is non-smooth. Hence, the KKT conditions are defined using the Clarke subdifferential. See Appendix A for more details of the KKT conditions. The theorem implies that even though there might be many possible directions $\frac{W}{\|W\|_F}$ that classify the dataset correctly, gradient flow converges only to directions that are KKT points of Problem (1). We note that such a KKT point is not necessarily a global/local optimum (cf. Vardi et al. (2021); Lyu et al. (2021)). Thus, under the theorem's assumptions, gradient flow *may not* converge to an optimum of Problem (1), but it is guaranteed to converge to a KKT point.

We now state our main result for this section. For convenience, we will use different notations for positive neurons (i.e., where $a_j = 1/\sqrt{m}$) and negative neurons (i.e., where $a_j = -1/\sqrt{m}$). Namely,

$$f(x; W) = \sum_{j=1}^{m} a_j \phi(w_j^\top x) = \sum_{j=1}^{m_1} \frac{1}{\sqrt{m}} \phi(v_j^\top x) - \sum_{j=1}^{m_2} \frac{1}{\sqrt{m}} \phi(u_j^\top x) . \qquad (2)$$

Note that $m = m_1 + m_2$. We assume that $m_1, m_2 \geq 1$.

**Theorem 3.2.** *Let $\{(x_i, y_i)\}_{i=1}^{n} \subseteq \mathbb{R}^d \times \{\pm 1\}$ be a training dataset, and let $R_{max} := \max_i \|x_i\|$, $R_{min} := \min_i \|x_i\|$ and $R = R_{max}/R_{min}$. We denote $I := [n]$, $I_+ := \{i \in I : y_i = 1\}$ and $I_- := \{i \in I : y_i = -1\}$. Assume that*

$$R_{min}^2 \geq 3\gamma^{-3} R^2 n \max_{i \neq j} |\langle x_i, x_j \rangle| .$$

*Let $f$ be the leaky ReLU network from (2) and let $W$ be a KKT point of Problem (1). Then, the following hold:*

1. *$y_i f(x_i; W) = 1$ for all $i \in I$.*

2. *$v_1 = \ldots = v_{m_1} := v$ and $u_1 = \ldots = u_{m_2} := u$. Hence, $\mathrm{rank}(W) \leq 2$.*

3. *$v = \frac{1}{\sqrt{m}} \sum_{i \in I_+} \lambda_i x_i - \frac{\gamma}{\sqrt{m}} \sum_{i \in I_-} \lambda_i x_i$ and $u = \frac{1}{\sqrt{m}} \sum_{i \in I_-} \lambda_i x_i - \frac{\gamma}{\sqrt{m}} \sum_{i \in I_+} \lambda_i x_i$, where $\lambda_i \in \left( \frac{1}{2R_{max}^2}, \frac{3}{2\gamma^2 R_{min}^2} \right)$ for every $i \in I$. Furthermore, for all $i \in I$ we have $y_i v^\top x_i > 0$ and $y_i u^\top x_i < 0$.*

4. *$W$ is a global optimum of Problem (1). Moreover, this global optimum is unique.*

5. *$v, u$ is the global optimum of the following convex problem:*

$$\min_{v, u \in \mathbb{R}^d} \frac{m_1}{2} \|v\|^2 + \frac{m_2}{2} \|u\|^2 \qquad (3)$$

$$\forall i \in I_+ : \quad \frac{m_1}{\sqrt{m}} v^\top x_i - \gamma \frac{m_2}{\sqrt{m}} u^\top x_i \geq 1$$

$$\forall i \in I_- : \quad \frac{m_2}{\sqrt{m}} u^\top x_i - \gamma \frac{m_1}{\sqrt{m}} v^\top x_i \geq 1 .$$

6. *Let $z = \frac{m_1}{\sqrt{m}} v - \frac{m_2}{\sqrt{m}} u$. For every $x \in \mathbb{R}^d$ we have $\mathrm{sign}\left(f(x; W)\right) = \mathrm{sign}(z^\top x)$. Thus, the network $f(\cdot; W)$ has a linear decision boundary.*

7. *The vector $z$ may not be an $\ell_2$-max-margin linear predictor, but it maximizes the margin approximately in the following sense. For all $i \in I$ we have $y_i z^\top x_i \geq 1$, and $\|z\| \leq \frac{2}{\kappa + \gamma} \|z^*\|$, where $\kappa := \sqrt{\frac{\min\{m_1, m_2\}}{\max\{m_1, m_2\}}}$, and $z^* := \mathrm{argmin}_{\tilde{z}} \|\tilde{z}\|$ s.t. $y_i \tilde{z}^\top x_i \geq 1$ for all $i \in I$.*

Note that by the above theorem, the KKT points possess very strong properties: the weight matrix is of rank at most 2, there is margin maximization in parameter space, in function space the predictor has a linear decision boundary, there may not be margin maximization in predictor space, but the predictor maximizes the margin approximately within a factor of $\frac{2}{\kappa + \gamma}$. Note that if $\kappa = 1$ (i.e., $m_1 = m_2$) and $\gamma$ is roughly 1, then we get margin maximization also in predictor space. We remark that variants of items 2, 5 and 6 were shown in Sarussi et al. (2021) under a different assumption called *Neural Agreement Regime* (as we discussed in the related work section).[1]

The proof of Theorem 3.2 is given in Appendix B. We now briefly discuss the proof idea. Since $W$ satisfies the KKT conditions of Problem (1), then there are $\lambda_1, \ldots, \lambda_n$ such that for every $j \in [m]$ we have

$$w_j = \sum_{i \in I} \lambda_i \nabla_{w_j} \left( y_i f(x_i; W) \right) = a_j \sum_{i \in I} \lambda_i y_i \phi'_{i, w_j} x_i ,$$

where $\phi'_{i, w_j}$ is a subgradient of $\phi$ at $w_j^\top x_i$. Also we have $\lambda_i \geq 0$ for all $i$, and $\lambda_i = 0$ if $y_i f(x_i; W) \neq 1$. We prove strictly positive upper and lower bounds for each of the $\lambda_i$'s. Since the $\lambda_i$'s are strictly

---

[1]In fact, the main challenge in our proof is to show that a property similar to their assumption holds in every KKT point in our setting.

positive, the KKT conditions show that the margin constraints are satisfied with equalities, i.e., part 1 of the theorem. By leveraging these bounds on the $\lambda_i$'s we also derive the remaining parts of the theorem.

The main assumption in Theorem 3.2 is that $R_{\min}^2 \geq 3\gamma^{-3}R^2 n \max_{i \neq j} |\langle x_i, x_j \rangle|$. In words, this means the squared norms of samples are much larger than the pairwise correlations between different samples, i.e. the training data are nearly orthogonal. Lemma 3.3 below implies that if the inputs $x_i$ are drawn from a well-conditioned Gaussian distribution (e.g., $\mathsf{N}(0, I_d)$), then it suffices to require $n \leq O\left(\gamma^3 \sqrt{\frac{d}{\log n}}\right)$, i.e., $d \geq \tilde{\Omega}\left(n^2\right)$ if $\gamma = \Omega(1)$. Lemma 3.3 holds more generally for a class of subgaussian distributions (see, e.g., Hu et al. (2020, Claim 3.1)), and we state the result for Gaussians here for simplicity.

**Lemma 3.3.** *Suppose that $x_1, \ldots, x_n$ are drawn i.i.d. from a $d$-dimensional Gaussian distribution* $\mathsf{N}(0, \Sigma)$, *where* $\mathrm{Tr}[\Sigma] = d$ *and* $\|\Sigma\|_2 = O(1)$. *Suppose $n \leq d^{O(1)}$. Then, with probability at least* $1 - n^{-10}$ *we have* $\frac{\|x_i\|^2}{d} = 1 \pm O(\sqrt{\frac{\log n}{d}})$ *for all $i$, and* $\frac{|\langle x_i, x_j \rangle|}{d} = O(\sqrt{\frac{\log n}{d}})$ *for all $i \neq j$.*

The proof of Lemma 3.3 is provided in Appendix C. We thus see that for data sampled i.i.d. from a well-conditioned Gaussian, near-orthogonality of training data holds when the training data is sufficiently high-dimensional, i.e. the dimension is much larger than the number of samples.

By Theorem 3.2, if the data points are nearly orthogonal then every KKT point of Problem (1) satisfies items 1-7 there. It leaves open the question of whether gradient flow converges to a KKT point. By Theorem 3.1, in order to prove convergence to a KKT point, it suffices to show that there exists time $t_0$ where $\widehat{L}(W(t_0)) < \frac{\log(2)}{n}$. In the following theorem we show that such $t_0$ exists, regardless of the initialization of gradient flow (the theorem holds both for the logistic and the exponential losses).

**Theorem 3.4.** *Consider gradient flow on a the network from* (2) *w.r.t. a dataset that satisfies the assumption from Theorem 3.2. Then, there exists a finite time $t_0$ such that for all $t \geq t_0$ we have* $\widehat{L}(W(t)) < \log(2)/n$.

We prove the theorem in Appendix D. Combining Theorems 3.1, 3.2 and 3.4, we get the following:

**Corollary 3.5.** *Consider gradient flow on the network from* (2) *w.r.t. a dataset that satisfies the assumption from Theorem 3.2. Then, gradient flow converges to zero loss, and converges in direction to a weight matrix $W$ that satisfies items 1-7 from Theorem 3.2.*

## 4 NON-ASYMPTOTIC ANALYSIS OF THE IMPLICIT BIAS

In this section, we study the implicit bias of gradient descent with a fixed step size following random initialization (refer to Section 2 for the definition of gradient descent). Our results in this section are for the logistic loss $\ell(z) = \log(1 + \exp(-z))$ but could be extended to the exponential loss as well. We shall assume the activation function $\phi$ satisfies $\phi(0) = 0$ and is twice differentiable and there exist constants $\gamma \in (0, 1], H > 0$ such that $0 < \gamma \leq \phi'(z) \leq 1$, and $|\phi''(z)| \leq H$. We shall refer to functions satisfying the above properties as $\gamma$-*leaky, $H$-smooth*. Note that such functions are not necessarily homogeneous. Examples of such functions are any smoothed approximation to the leaky ReLU that is zero at the origin. One such example is: $\phi(z) = \gamma z + (1 - \gamma) \log\left(\frac{1}{2}(1 + \exp(z))\right)$, which is $\gamma$-leaky and $1/4$-smooth (see Figure 3 in the appendix for a side-by-side plot of this activation with the standard leaky ReLU).

We next introduce the definition of stable rank (Rudelson & Vershynin, 2007).

**Definition 4.1.** *The stable rank of a matrix $W \in \mathbb{R}^{m \times d}$ is* $\mathsf{StableRank}(W) = \|W\|_F^2 / \|W\|_2^2$.

The stable rank is in many ways analogous to the classical rank of a matrix but is considerably more well-behaved. For instance, consider the diagonal matrix $W \in \mathbb{R}^{d \times d}$ with diagonal entries equal to 1 except for the first entry which is equal to $\varepsilon \geq 0$. As $\varepsilon \to 0$, the classical rank of the matrix is equal to $d$ until $\varepsilon$ exactly equals 0, while on the other hand the stable rank smoothly decreases from $d$ to $d - 1$. For another example, suppose again $W \in \mathbb{R}^{d \times d}$ is diagonal with $W_{1,1} = 1$ and $W_{i,i} = \exp(-d)$ for $i \geq 2$. The classical rank of this matrix is exactly equal to $d$, while the stable rank of this matrix is $1 + o_d(1)$.

With the above conditions in hand, we can state our main theorem for this section.

**Theorem 4.2.** *Suppose that $\phi$ is a $\gamma$-leaky, $H$-smooth activation. For training data $\{(x_i, y_i)\}_{i=1}^n \subset \mathbb{R}^d \times \{\pm 1\}$, let $R_{max} = \max_i \|x_i\|$ and $R_{min} = \min_i \|x_i\|$, and suppose $R = R_{max}/R_{min}$ is at most an absolute constant. Denote by $C_R := 10R^2/\gamma^2 + 10$. Assume the training data satisfies,*

$$R_{min}^2 \geq 5\gamma^{-2}C_R n \max_{i \neq j} |\langle x_i, x_j \rangle|.$$

*There exist absolute constants $C_1, C_2 > 1$ (independent of $m$, $d$, and $n$) such that the following holds. For any $\delta \in (0,1)$, if the step-size satisfies $\alpha \leq \gamma^2(5nR_{max}^2 R^2 C_R \max(1, H))^{-1}$, and $\omega_{\text{init}} \leq \alpha \gamma^2 R_{min}(72RC_R n\sqrt{md\log(4m/\delta)})^{-1}$, then with probability at least $1-\delta$ over the random initialization of gradient descent, the trained network satisfies:*

1. *The empirical risk under the logistic loss satisfies $\widehat{L}(W^{(t)}) \leq \sqrt{C_1 n/R_{min}^2 \alpha t}$ for $t \geq 1$.*

2. *The $\ell_2$ norm of each neuron grows to infinity: for all $j$, $\|w_j^{(t)}\|_2 \to \infty$.*

3. *The stable rank of the weights is bounded: $\sup_{t \geq 1} \left\{ \mathsf{StableRank}(W^{(t)}) \right\} \leq C_2$.*

We now make a few remarks on the above theorem. We note that the assumption on the training data is the same as in Theorem 3.2 up to constants (treating $\gamma$ as a constant), and is satisfied in many settings when $d \gg n^2$ (see Lemma 3.3).

For the first part of the theorem, we show that despite the non-convexity of the underlying optimization problem, gradient descent can efficiently minimize the training error, driving the empirical risk to zero.

For the second part of the theorem, note that since the empirical risk under the logistic loss is driven to zero and the logistic loss is decreasing and satisfies $\ell(z) > 0$ for all $z$, it is necessarily the case that the spectral norm of the first layer weights $\|W^{(t)}\|_2 \to \infty$. (Otherwise, $\widehat{L}(W^{(t)})$ would be bounded from below by a constant.) This leaves open the question of whether only a few neurons in the network are responsible for the growth of the magnitude of the spectral norm, and part (2) of the theorem resolves this question.

The third part of the theorem is perhaps the most interesting one. In Theorem 3.2, we showed that for the standard leaky ReLU activation trained on nearly-orthogonal data with gradient flow, the asymptotic true rank of the network is at most 2. By contrast, Theorem 4.2 shows that the stable rank of neural networks with $\gamma$-leaky, $H$-smooth activations trained by gradient descent have a constant stable rank after the first step of gradient descent and the rank remains bounded by a constant throughout the trajectory. Note that at initialization, by standard concentration bounds for random matrices (see, e.g., Vershynin (2010)), the stable rank satisfies $\mathsf{StableRank}(W^{(0)}) \approx \Theta(md/(\sqrt{m}+\sqrt{d})^2) = \Omega(m \wedge d)$, so that Theorem 4.2 implies that gradient descent drastically reduces the rank of the matrix after just one step.

The details for the proof of Theorem 4.2 are provided in Appendix E, but we provide some of the main ideas for the proofs of part 1 and 3 of the theorem here. For the first part, note that training data satisfying the assumptions in the theorem are linearly separable with a large margin (take, for instance, the vector $\sum_{i=1}^n y_i x_i$). We use this to establish a proxy Polyak–Lojasiewicz (PL) inequality (Frei & Gu, 2021) that takes the form $\|\nabla \widehat{L}(W^{(t)})\|_F \geq c\widehat{G}(W^{(t)})$ for some $c > 0$, where $\widehat{G}(W^{(t)})$ is the empirical risk under the sigmoid loss $-\ell'(z) = 1/(1 + \exp(z))$. Because we consider smoothed leaky ReLU activations, we can use a smoothness-based analysis of gradient descent to show $\|\nabla \widehat{L}(W^{(t)})\|_F \to 0$, which implies $\widehat{G}(W^{(t)}) \to 0$ by the proxy PL inequality. We then translate guarantees for $\widehat{G}(W^{(t)})$ into guarantees for $\widehat{L}(W^{(t)})$ by comparing the sigmoid and logistic losses.

For the third part of the theorem, we need to establish two things: $(i)$ an upper bound for the Frobenius norm, and $(ii)$ a lower bound for the spectral norm. A loose approach for bounding the Frobenius norm via an application of the triangle inequality (over time steps) results in a stable rank bound that grows with the number of samples. To develop a tighter upper bound, we first establish a structural condition we refer to as a *loss ratio bound* (see Lemma E.4). In the gradient descent updates, each sample is weighted by a quantity that scales with $-\ell'(y_i f(x_i; W^{(t)})) \in (0, 1)$. We show that these $-\ell'$ losses grow at approximately the same rate for each sample throughout training,

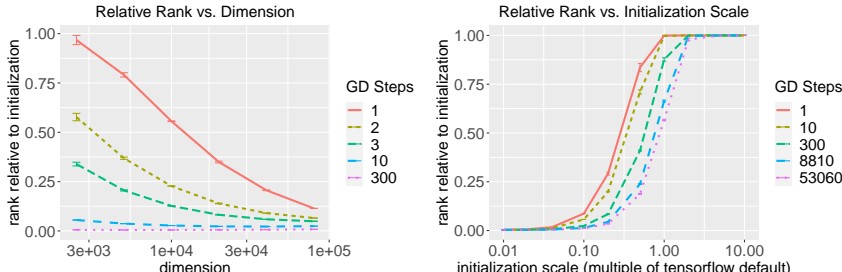

Figure 1: Relative reduction in the stable rank of two-layer nets trained by gradient descent for Gaussian mixture model data (cf. (4)). The rank reduction happens more quickly as the dimension grows (left; initialization scale $50\times$ smaller than default TensorFlow, $\alpha = 0.01$) and as the initialization scale decreases (right; $d = 10^4$, $\alpha = 0.16$).

and that this allows for a tighter upper bound for the Frobenius norm. Loss ratio bounds were key to the generalization analysis of two previous works on benign overfitting (Chatterji & Long, 2021; Frei et al., 2022a) and may be of independent interest. In Proposition E.10 we provide a general approach for proving loss ratio bounds that can hold for more general settings than the ones we consider in this work (i.e., data which are not high-dimensional, and networks with non-leaky activations). The lower bound on the spectral norm follows by identifying a single direction $\widehat{\mu} := \sum_{i=1}^{n} y_i x_i$ that is strongly correlated with every neuron's weight $w_j$, in the sense that $|\langle w_j^{(t)}/\|w_j^{(t)}\|, \widehat{\mu}\rangle|$ is relatively large for each $j \in [m]$. Since every neuron is strongly correlated with this direction, this allows for a good lower bound on the spectral norm.

## 5 IMPLICATIONS OF THE IMPLICIT BIAS AND EMPIRICAL OBSERVATIONS

The results in the preceding sections show a remarkable simplicity bias of gradient-based optimization when training two-layer networks with leaky activations on sufficiently high-dimensional data. For gradient flow, regardless of the initialization, the learned network has a *linear* decision boundary, even when the labels $y$ are some nonlinear function of the input features and when the network has the capacity to approximate any continuous function. With our analysis of gradient descent, we showed that the bias towards producing low-complexity networks (as measured by the stable rank of the network) is something that occurs quickly following random initialization, provided the initialization scale is small enough.

In some distributional settings, this bias towards rather simple classifiers may be beneficial, while in others it may be harmful. To see where it may be beneficial, consider a Gaussian mixture model distribution P, parameterized by a mean vector $\mu \in \mathbb{R}^d$, where samples $(x, y) \sim$ P have a distribution as follows:
$$y \sim \mathsf{Uniform}(\{\pm 1\}), \quad x|y \sim y\mu + z, \quad z \sim \mathsf{N}(0, I_d). \tag{4}$$
The linear classifier $x \mapsto \mathrm{sign}(\langle \mu, x \rangle)$ performs optimally for this distribution, and so the implicit bias of gradient descent towards low-rank classifiers (and of gradient flow towards linear decision boundaries) for high-dimensional data could in principle be helpful for allowing neural networks trained on such data to generalize well for this distribution. Indeed, as shown by Chatterji & Long (2021), since $\|x_i\|^2 \approx d + \|\mu\|^2$ while $|\langle x_i, x_j \rangle| \approx \|\mu\|^2 + \sqrt{d}$ for $i \neq j$, provided $\|\mu\| = \Theta(d^\beta)$ and $d \gg n^{\frac{1}{1-2\beta}} \vee n^2$ for $\beta \in (0, 1/2)$, the assumptions in Theorem 4.2 hold. Thus, gradient descent on two-layer networks with $\gamma$-leaky, $H$-smooth activations, the empirical risk is driven to zero and the stable rank of the network is constant after the first step of gradient descent. In this setting, Frei et al. (2022a) recently showed that such networks have small generalization error. This shows that the implicit bias towards classifiers with constant rank can be beneficial in distributional settings where linear classifiers can perform well.

On the other hand, the same implicit bias can be harmful if the training data come from a distribution that does not align with this bias. Consider the noisy 2-xor distribution $\mathcal{D}_{\mathsf{xor}}$ in $d \geq 3$ dimensions defined by $x = z + \xi$ where $z \sim \mathsf{Uniform}(\{\pm\mu_1, \pm\mu_2\})$, where $\mu_1, \mu_2$ are orthogonal with identical norms, $\xi \sim \mathsf{N}(0, I_d)$, and $y = \mathrm{sign}(|\langle \mu_1, x \rangle| - |\langle \mu_2, x \rangle|)$. Then every linear classifier achieves 50% test error on $\mathcal{D}_{\mathsf{xor}}$. Moreover, provided $\|\mu_i\| = \Theta(d^\beta)$ for $\beta < 1/2$, by the same reasoning in the

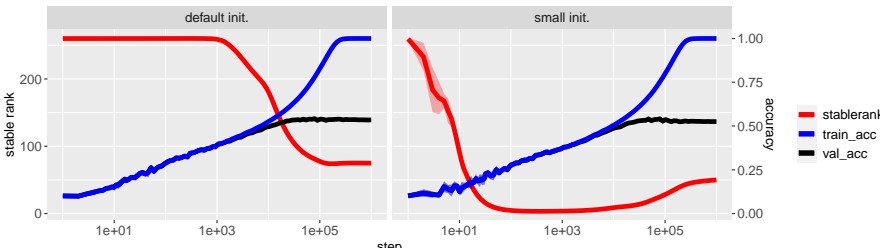

Figure 2: Stable rank of SGD-trained two-layer ReLU networks on CIFAR-10. Compared to the default TensorFlow initialization (left), a smaller initialization (right) results in a smaller stable rank, and this effect is especially pronounced before the very late stages of training. Remarkably, the train (blue) and test (black) accuracy behavior is essentially the same.

preceding paragraph the assumptions needed for Theorem 3.2 are satisfied provided $d \gg n^{\frac{1}{1-2\beta}} \vee n^2$. In this setting, regardless of the initialization, by Theorem 3.2 the limit of gradient flow produces a neural network which has a linear decision boundary and thus achieves 50% test error. In the appendix (see Fig. 6) we verify this with experiments.

Thus, the implicit bias can be beneficial in some settings and harmful in others. Theorem 4.2 and Lemma 3.3 suggest that the relationship between the input dimension and the number of samples, as well as the initialization variance, can influence how quickly gradient descent finds low-rank networks. In Figure 1 we examine these factors for two-layer nets trained on a Gaussian mixture model distribution (see Appendix F for experimental details). We see that the bias towards rank reduction increases as the dimension increases and the initialization scale decreases, as suggested by our theory. Moreover, it appears that the initialization scale is more influential for determining the rank reduction than training gradient descent for longer. In Appendix F we provide more detailed empirical investigations into this phenomenon.

In Figure 2, we investigate whether or not the initialization scale's effect on the rank reduction of gradient descent occurs in settings not covered by our theory, namely in two-layer ReLU networks with bias terms trained by SGD on CIFAR-10. We consider two different initialization schemes: (1) Glorot uniform, the default TensorFlow initialization scheme with standard deviation of order $1/\sqrt{m+d}$, and (2) a uniform initialization scheme with $50\times$ smaller standard deviation than that of the Glorot uniform initialization. In the default initialization scheme, it appears that a reduction in the rank of the network only comes in the late stages of training, and the smallest stable rank achieved by the network within $10^6$ steps is 74.0. On the other hand, with the smaller initialization scheme, the rank reduction comes rapidly, and the smallest stable rank achieved by the network is 3.25. It is also interesting to note that in the small initialization setting, after gradient descent rapidly produces low-rank weights, the rank of the trained network begins to increase only when the gap between the train and test accuracy begin to diverge.

## 6 CONCLUSION

In this work, we characterized the implicit bias of common gradient-based optimization algorithms for two-layer leaky ReLU networks when trained on high-dimensional datasets. For both gradient flow and gradient descent, we proved convergence to near-zero training loss and that there is an implicit bias towards low-rank networks. For gradient flow, we showed a number of additional implicit biases: the weights are (unique) global maxima of the associated margin maximization problem, and the decision boundary of the learned network is linear. For gradient descent, we provided experimental evidence which suggests that small initialization variance is important for gradient descent's ability to quickly produce low-rank networks.

There are many natural directions to pursue following this work. One question is whether or not a similar implicit bias towards low-rank weights in fully connected networks exists for networks with different activation functions or for data which is not nearly orthogonal. Our proofs relied heavily upon the near-orthogonality of the data, and the 'leaky' behavior of the leaky ReLU, namely that there is some $\gamma > 0$ such that $\phi'(z) \geq \gamma$ for all $z \in \mathbb{R}$. We conjecture that some of the properties we showed in Theorem 3.2 (e.g., a linear decision boundary) may not hold for non-leaky activations, like the ReLU, or without the near-orthogonality assumption.

ACKNOWLEDGEMENTS

We thank Matus Telgarsky for helpful discussions. This work was done in part while the authors were visiting the Simons Institute for the Theory of Computing as a part of the Deep Learning Theory Summer Cluster. SF, GV, PB, and NS acknowledge the support of the NSF and the Simons Foundation for the Collaboration on the Theoretical Foundations of Deep Learning through awards DMS-2031883 and #814639.

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

CONTENTS

## A  PRELIMINARIES ON THE CLARKE SUBDIFFERENTIAL AND THE KKT CONDITIONS

Below we review the definition of the KKT conditions for non-smooth optimization problems (cf. Lyu & Li (2019); Dutta et al. (2013)).

Let $f : \mathbb{R}^d \to \mathbb{R}$ be a locally Lipschitz function. The Clarke subdifferential (Clarke et al., 2008) at $x \in \mathbb{R}^d$ is the convex set

$$\partial^\circ f(x) := \text{conv} \left\{ \lim_{i \to \infty} \nabla f(x_i) \,\Big|\, \lim_{i \to \infty} x_i = x, \ f \text{ is differentiable at } x_i \right\} .$$

If $f$ is continuously differentiable at $x$ then $\partial^\circ f(x) = \{\nabla f(x)\}$. For the Clarke subdifferential the chain rule holds as an inclusion rather than an equation. That is, for locally Lipschitz functions

$z_1, \ldots, z_n : \mathbb{R}^d \to \mathbb{R}$ and $f : \mathbb{R}^n \to \mathbb{R}$, we have

$$\partial^{\circ}(f \circ z)(x) \subseteq \text{conv} \left\{ \sum_{i=1}^{n} \alpha_i h_i : \alpha \in \partial^{\circ} f(z_1(x), \ldots, z_n(x)), h_i \in \partial^{\circ} z_i(x) \right\} .$$

Consider the following optimization problem

$$\min f(x) \quad \text{s.t.} \quad \forall n \in [N] \ g_n(x) \leq 0 , \tag{5}$$

where $f, g_1, \ldots, g_n : \mathbb{R}^d \to \mathbb{R}$ are locally Lipschitz functions. We say that $x \in \mathbb{R}^d$ is a *feasible point* of Problem (5) if $x$ satisfies $g_n(x) \leq 0$ for all $n \in [N]$. We say that a feasible point $x$ is a *KKT point* if there exists $\lambda_1, \ldots, \lambda_N \geq 0$ such that

1. $\mathbf{0} \in \partial^{\circ} f(x) + \sum_{n \in [N]} \lambda_n \partial^{\circ} g_n(x)$;

2. For all $n \in [N]$ we have $\lambda_n g_n(x) = 0$.

## B   PROOF OF THEOREM 3.2

We start with some notations. We denote $p = \max_{i \neq j} |\langle x_i, x_j \rangle|$. Thus, our assumption on $n$ can be written as $n \leq \frac{\gamma^3}{3} \cdot \frac{R_{\min}^2}{p} \cdot \frac{R_{\min}^2}{R_{\max}^2}$. Since $W$ satisfies the KKT conditions of Problem (1), then there are $\lambda_1, \ldots, \lambda_n$ such that for every $j \in [m_1]$ we have

$$v_j = \sum_{i \in I} \lambda_i \nabla_{v_j} (y_i f(x_i; W)) = \frac{1}{\sqrt{m}} \sum_{i \in I} \lambda_i y_i \phi'_{i,v_j} x_i , \tag{6}$$

where $\phi'_{i,v_j}$ is a subgradient of $\phi$ at $v_j^{\top} x_i$, i.e., if $v_j^{\top} x_i > 0$ then $\phi'_{i,v_j} = 1$, if $v_j^{\top} x_i < 0$ then $\phi'_{i,v_j} = \gamma$ and otherwise $\phi'_{i,v_j}$ is some value in $[\gamma, 1]$. Also we have $\lambda_i \geq 0$ for all $i$, and $\lambda_i = 0$ if $y_i f(x_i; W) \neq 1$. Likewise, for all $j \in [m_2]$ we have

$$u_j = \sum_{i \in I} \lambda_i \nabla_{u_j} (y_i f(x_i; W)) = \frac{1}{\sqrt{m}} \sum_{i \in I} \lambda_i (-y_i) \phi'_{i,u_j} x_i , \tag{7}$$

where $\phi'_{i,u_j}$ is defined similarly to $\phi'_{i,v_j}$. The proof of the theorem follows from the following lemmas.

**Lemma B.1.** *For all $i \in I$ we have $\sum_{j \in [m_1]} \lambda_i \phi'_{i,v_j} + \sum_{j \in [m_2]} \lambda_i \phi'_{i,u_j} < \frac{3m}{2\gamma R_{min}^2}$. Furthermore, $\lambda_i < \frac{3}{2\gamma^2 R_{min}^2}$ for all $i \in I$.*

*Proof.* Let $\xi = \max_{q \in I} \left( \sum_{j \in [m_1]} \lambda_q \phi'_{q,v_j} + \sum_{j \in [m_2]} \lambda_q \phi'_{q,u_j} \right)$ and suppose that $\xi \geq \frac{3m}{2\gamma R_{min}^2}$. Let $r = \text{argmax}_{q \in I} \left( \sum_{j \in [m_1]} \lambda_q \phi'_{q,v_j} + \sum_{j \in [m_2]} \lambda_q \phi'_{q,u_j} \right)$. Since $\xi \geq \frac{3m}{2\gamma R_{min}^2} > 0$ then $\lambda_r > 0$, and hence by the KKT conditions we must have $y_r f(x_r; W) = 1$.

We consider two cases:

**Case 1:** Assume that $r \in I_-$. Using (6) and (7), we have

$$\sqrt{m}f(x_r; W) = \sum_{j \in [m_1]} \phi(v_j^\top x_r) - \sum_{j \in [m_2]} \phi(u_j^\top x_r)$$

$$= \sum_{j \in [m_1]} \phi\left(\frac{1}{\sqrt{m}} \sum_{q \in I} \lambda_q y_q \phi'_{q,v_j} x_q^\top x_r\right) - \sum_{j \in [m_2]} \phi\left(\frac{1}{\sqrt{m}} \sum_{q \in I} \lambda_q(-y_q) \phi'_{q,u_j} x_q^\top x_r\right)$$

$$= \sum_{j \in [m_1]} \phi\left(\frac{1}{\sqrt{m}} \lambda_r y_r \phi'_{r,v_j} x_r^\top x_r + \frac{1}{\sqrt{m}} \sum_{q \in I \setminus \{r\}} \lambda_q y_q \phi'_{q,v_j} x_q^\top x_r\right)$$

$$- \sum_{j \in [m_2]} \phi\left(\frac{1}{\sqrt{m}} \lambda_r(-y_r) \phi'_{r,u_j} x_r^\top x_r + \frac{1}{\sqrt{m}} \sum_{q \in I \setminus \{r\}} \lambda_q(-y_q) \phi'_{q,u_j} x_q^\top x_r\right)$$

$$\leq \sum_{j \in [m_1]} \phi\left(-\frac{1}{\sqrt{m}} \lambda_r \phi'_{r,v_j} R_{\min}^2 + \frac{1}{\sqrt{m}} \sum_{q \in I \setminus \{r\}} \lambda_q y_q \phi'_{q,v_j} x_q^\top x_r\right)$$

$$- \sum_{j \in [m_2]} \phi\left(\frac{1}{\sqrt{m}} \lambda_r \phi'_{r,u_j} R_{\min}^2 + \frac{1}{\sqrt{m}} \sum_{q \in I \setminus \{r\}} \lambda_q(-y_q) \phi'_{q,u_j} x_q^\top x_r\right) .$$

Since the derivative of $\phi$ is lower bounded by $\gamma$, we know $\phi(z_1) - \phi(z_2) \geq \gamma(z_1 - z_2)$ for all $z_1, z_2 \in \mathbb{R}$. Using this and the definition of $\xi$, the above is at most

$$\sum_{j \in [m_1]} \left[\phi\left(\frac{1}{\sqrt{m}} \sum_{q \in I \setminus \{r\}} \lambda_q y_q \phi'_{q,v_j} x_q^\top x_r\right) - \frac{1}{\sqrt{m}} \gamma \cdot \lambda_r \phi'_{r,v_j} R_{\min}^2\right]$$

$$- \sum_{j \in [m_2]} \left[\phi\left(\frac{1}{\sqrt{m}} \sum_{q \in I \setminus \{r\}} \lambda_q(-y_q) \phi'_{q,u_j} x_q^\top x_r\right) + \frac{1}{\sqrt{m}} \gamma \cdot \lambda_r \phi'_{r,u_j} R_{\min}^2\right]$$

$$\leq -\frac{1}{\sqrt{m}} \gamma \xi R_{\min}^2 + \sum_{j \in [m_1]} \left|\frac{1}{\sqrt{m}} \sum_{q \in I \setminus \{r\}} \lambda_q y_q \phi'_{q,v_j} x_q^\top x_r\right| + \sum_{j \in [m_2]} \left|\frac{1}{\sqrt{m}} \sum_{q \in I \setminus \{r\}} \lambda_q(-y_q) \phi'_{q,u_j} x_q^\top x_r\right|$$

$$\leq -\frac{1}{\sqrt{m}} \gamma \xi R_{\min}^2 + \frac{1}{\sqrt{m}} \sum_{j \in [m_1]} \sum_{q \in I \setminus \{r\}} \left|\lambda_q y_q \phi'_{q,v_j} x_q^\top x_r\right| + \frac{1}{\sqrt{m}} \sum_{j \in [m_2]} \sum_{q \in I \setminus \{r\}} \left|\lambda_q(-y_q) \phi'_{q,u_j} x_q^\top x_r\right| .$$

Using $|x_q^\top x_r| \leq p$ for $q \neq r$, the above is at most

$$-\frac{1}{\sqrt{m}} \gamma \xi R_{\min}^2 + \frac{1}{\sqrt{m}} \sum_{j \in [m_1]} \sum_{q \in I \setminus \{r\}} \lambda_q \phi'_{q,v_j} p + \frac{1}{\sqrt{m}} \sum_{j \in [m_2]} \sum_{q \in I \setminus \{r\}} \lambda_q \phi'_{q,u_j} p$$

$$= -\frac{1}{\sqrt{m}} \gamma \xi R_{\min}^2 + \frac{p}{\sqrt{m}} \sum_{q \in I \setminus \{r\}} \left(\sum_{j \in [m_1]} \lambda_q \phi'_{q,v_j} + \sum_{j \in [m_2]} \lambda_q \phi'_{q,u_j}\right)$$

$$\leq -\frac{1}{\sqrt{m}} \gamma \xi R_{\min}^2 + \frac{p}{\sqrt{m}} \cdot |I| \cdot \max_{q \in I} \left(\sum_{j \in [m_1]} \lambda_q \phi'_{q,v_j} + \sum_{j \in [m_2]} \lambda_q \phi'_{q,u_j}\right)$$

$$= -\frac{1}{\sqrt{m}} \gamma \xi R_{\min}^2 + \frac{p}{\sqrt{m}} n \xi = -\frac{\xi}{\sqrt{m}} (\gamma R_{\min}^2 - np) .$$

By our assumption on $n$, we can bound the above expression by

$$-\frac{\xi}{\sqrt{m}}\left(\gamma R_{\min}^2 - p \cdot \frac{\gamma^3}{3} \cdot \frac{R_{\min}^2}{p} \cdot \frac{R_{\min}^2}{R_{\max}^2}\right) = -\frac{\xi R_{\min}^2}{\sqrt{m}}\left(\gamma - \frac{\gamma^3}{3} \cdot \frac{R_{\min}^2}{R_{\max}^2}\right)$$

$$< -\frac{\xi R_{\min}^2}{\sqrt{m}}\left(\gamma - \frac{\gamma}{3}\right)$$

$$= -\frac{\xi R_{\min}^2}{\sqrt{m}} \cdot \frac{2\gamma}{3}$$

$$\leq -\frac{3m}{2\gamma R_{\min}^2} \cdot \frac{R_{\min}^2}{\sqrt{m}} \cdot \frac{2\gamma}{3} = -\sqrt{m} \ .$$

Thus, we obtain $f(x_r; W) < -1$ in contradiction to $y_r f(x_r; W) = 1$.

**Case 2:**   Assume that $r \in I_+$. A similar calculation to the one given in case 1 (which we do not repeat for conciseness) implies that $f(x_r; W) > 1$, in contradiction to $y_r f(x_r; W) = 1$. It concludes the proof of $\xi < \frac{3m}{2\gamma R_{\min}^2}$.

Finally, since $\xi < \frac{3m}{2\gamma R_{\min}^2}$ and the derivative of $\phi$ is lower bounded by $\gamma$, then for all $i \in I$ we have

$$\frac{3m}{2\gamma R_{\min}^2} > \sum_{j\in[m_1]} \lambda_i \phi'_{i,v_j} + \sum_{j\in[m_2]} \lambda_i \phi'_{i,u_j} \geq m\lambda_i \gamma \ ,$$

and hence $\lambda_i < \frac{3}{2\gamma^2 R_{\min}^2}$. $\qquad\square$

**Lemma B.2.** *For all $i \in I$ we have $\sum_{j\in[m_1]} \lambda_i \phi'_{i,v_j} + \sum_{j\in[m_2]} \lambda_i \phi'_{i,u_j} > \frac{m}{2R_{\max}^2}$. Furthermore, $\lambda_i > \frac{1}{2R_{\max}^2}$ for all $i \in I$.*

*Proof.* Suppose that there is $i \in I$ such that $\sum_{j\in[m_1]} \lambda_i \phi'_{i,v_j} + \sum_{j\in[m_2]} \lambda_i \phi'_{i,u_j} \leq \frac{m}{2R_{\max}^2}$. Using (6) and (7), we have

$$\sqrt{m} \leq \left|\sqrt{m} f(x_i; W)\right| = \left|\sum_{j\in[m_1]} \phi(v_j^\top x_i) - \sum_{j\in[m_2]} \phi(u_j^\top x_i)\right| \leq \sum_{j\in[m_1]} \left|v_j^\top x_i\right| + \sum_{j\in[m_2]} \left|u_j^\top x_i\right|$$

$$= \sum_{j\in[m_1]} \left|\frac{1}{\sqrt{m}} \sum_{q\in I} \lambda_q y_q \phi'_{q,v_j} x_q^\top x_i\right| + \sum_{j\in[m_2]} \left|\frac{1}{\sqrt{m}} \sum_{q\in I} \lambda_q(-y_q) \phi'_{q,u_j} x_q^\top x_i\right|$$

$$\leq \frac{1}{\sqrt{m}} \sum_{j\in[m_1]} \left(\left|\lambda_i y_i \phi'_{i,v_j} x_i^\top x_i\right| + \sum_{q\in I\setminus\{i\}} \left|\lambda_q y_q \phi'_{q,v_j} x_q^\top x_i\right|\right)$$

$$+ \frac{1}{\sqrt{m}} \sum_{j\in[m_2]} \left(\left|\lambda_i(-y_i) \phi'_{i,u_j} x_i^\top x_i\right| + \sum_{q\in I\setminus\{i\}} \left|\lambda_q(-y_q) \phi'_{q,u_j} x_q^\top x_i\right|\right) \ .$$

Using $|x_q^\top x_i| \le p$ for $q \ne i$ and $x_i^\top x_i \le R_{\max}^2$, the above is at most

$$\frac{1}{\sqrt{m}} \sum_{j \in [m_1]} \left( \lambda_i \phi'_{i,v_j} R_{\max}^2 + \sum_{q \in I \setminus \{i\}} \lambda_q \phi'_{q,v_j} p \right) + \frac{1}{\sqrt{m}} \sum_{j \in [m_2]} \left( \lambda_i \phi'_{i,u_j} R_{\max}^2 + \sum_{q \in I \setminus \{i\}} \lambda_q \phi'_{q,u_j} p \right)$$

$$= \frac{1}{\sqrt{m}} \left( \sum_{j \in [m_1]} \lambda_i \phi'_{i,v_j} R_{\max}^2 + \sum_{j \in [m_2]} \lambda_i \phi'_{i,u_j} R_{\max}^2 \right) + \frac{1}{\sqrt{m}} \sum_{q \in I \setminus \{i\}} \left( \sum_{j \in [m_1]} \lambda_q \phi'_{q,v_j} p + \sum_{j \in [m_2]} \lambda_q \phi'_{q,u_j} p \right)$$

$$= \frac{R_{\max}^2}{\sqrt{m}} \left( \sum_{j \in [m_1]} \lambda_i \phi'_{i,v_j} + \sum_{j \in [m_2]} \lambda_i \phi'_{i,u_j} \right) + \frac{p}{\sqrt{m}} \sum_{q \in I \setminus \{i\}} \left( \sum_{j \in [m_1]} \lambda_q \phi'_{q,v_j} + \sum_{j \in [m_2]} \lambda_q \phi'_{q,u_j} \right)$$

$$\le \frac{R_{\max}^2}{\sqrt{m}} \cdot \frac{m}{2 R_{\max}^2} + \frac{p}{\sqrt{m}} \cdot |I| \cdot \max_{q \in I} \left( \sum_{j \in [m_1]} \lambda_q \phi'_{q,v_j} + \sum_{j \in [m_2]} \lambda_q \phi'_{q,u_j} \right) \,.$$

Combining the above with our assumption on $n$, we get

$$\max_{q \in I} \left( \sum_{j \in [m_1]} \lambda_q \phi'_{q,v_j} + \sum_{j \in [m_2]} \lambda_q \phi'_{q,u_j} \right) \ge \frac{m}{2np} \ge \frac{m}{2p} \cdot \frac{3p}{\gamma^3 R_{\min}^2} \cdot \frac{R_{\max}^2}{R_{\min}^2} > \frac{3m}{2\gamma R_{\min}^2} \,,$$

in contradiction to Lemma B.1. It concludes the proof of $\sum_{j \in [m_1]} \lambda_i \phi'_{i,v_j} + \sum_{j \in [m_2]} \lambda_i \phi'_{i,u_j} > \frac{m}{2 R_{\max}^2}$.

Finally, since $\sum_{j \in [m_1]} \lambda_i \phi'_{i,v_j} + \sum_{j \in [m_2]} \lambda_i \phi'_{i,u_j} > \frac{m}{2 R_{\max}^2}$ and the derivative of $\phi$ is upper bounded by $1$, then for all $i \in I$ we have

$$\frac{m}{2 R_{\max}^2} < \sum_{j \in [m_1]} \lambda_i \phi'_{i,v_j} + \sum_{j \in [m_2]} \lambda_i \phi'_{i,u_j} \le m \lambda_i \,,$$

and hence $\lambda_i > \frac{1}{2 R_{\max}^2}$. $\qquad\square$

**Lemma B.3.** *For all $i \in I$ we have $y_i f(x_i; W) = 1$.*

*Proof.* By Lemma B.2 we have $\lambda_i > 0$ for all $i \in I$, and hence by the KKT conditions we must have $y_i f(x_i; W) = 1$. $\qquad\square$

**Lemma B.4.** *We have*

$$v_1 = \ldots = v_{m_1} = \frac{1}{\sqrt{m}} \sum_{i \in I_+} \lambda_i x_i - \frac{\gamma}{\sqrt{m}} \sum_{i \in I_-} \lambda_i x_i \,,$$

*and*

$$u_1 = \ldots = u_{m_2} = \frac{1}{\sqrt{m}} \sum_{i \in I_-} \lambda_i x_i - \frac{\gamma}{\sqrt{m}} \sum_{i \in I_+} \lambda_i x_i \,.$$

*Moreover, for all $i \in I$ we have: $y_i v_j^\top x_i > 0$ for every $j \in [m_1]$, and $y_i u_j^\top x_i < 0$ for every $j \in [m_2]$.*

*Proof.* Fix $j \in [m_1]$. By (6) for all $i \in I_+$ we have

$$v_j^\top x_i = \frac{1}{\sqrt{m}} \sum_{q \in I} \lambda_q y_q \phi'_{q,v_j} x_q^\top x_i$$

$$= \frac{1}{\sqrt{m}} \lambda_i y_i \phi'_{i,v_j} x_i^\top x_i + \frac{1}{\sqrt{m}} \sum_{q \in I \setminus \{i\}} \lambda_q y_q \phi'_{q,v_j} x_q^\top x_i$$

$$\ge \frac{1}{\sqrt{m}} \lambda_i \phi'_{i,v_j} R_{\min}^2 - \frac{1}{\sqrt{m}} \sum_{q \in I \setminus \{i\}} \lambda_q \phi'_{q,v_j} p \,.$$

By Lemma B.1 and Lemma B.2, and using $\phi'_{q,v_j} \in [\gamma, 1]$ for all $q \in I$, the above is larger than

$$\frac{1}{\sqrt{m}} \cdot \frac{1}{2R_{\max}^2} \cdot \gamma R_{\min}^2 - \frac{1}{\sqrt{m}} \cdot n \cdot \frac{3}{2\gamma^2 R_{\min}^2} \cdot p \geq \frac{\gamma R_{\min}^2}{2\sqrt{m}R_{\max}^2} - \frac{1}{\sqrt{m}} \cdot \frac{\gamma^3}{3} \cdot \frac{R_{\min}^2}{p} \cdot \frac{R_{\min}^2}{R_{\max}^2} \cdot \frac{3}{2\gamma^2 R_{\min}^2} \cdot p$$

$$= \frac{\gamma R_{\min}^2}{2\sqrt{m}R_{\max}^2} - \frac{\gamma R_{\min}^2}{2\sqrt{m}R_{\max}^2} = 0 \ .$$

Thus, $v_j^\top x_i > 0$, which implies $\phi'_{i,v_j} = 1$.

Similarly, for all $i \in I_-$ we have

$$v_j^\top x_i = \frac{1}{\sqrt{m}} \sum_{q \in I} \lambda_q y_q \phi'_{q,v_j} x_q^\top x_i$$

$$= \frac{1}{\sqrt{m}} \lambda_i y_i \phi'_{i,v_j} x_i^\top x_i + \frac{1}{\sqrt{m}} \sum_{q \in I \setminus \{i\}} \lambda_q y_q \phi'_{q,v_j} x_q^\top x_i$$

$$\leq -\frac{1}{\sqrt{m}} \lambda_i \phi'_{i,v_j} R_{\min}^2 + \frac{1}{\sqrt{m}} \sum_{q \in I \setminus \{i\}} \lambda_q \phi'_{q,v_j} p \ .$$

By Lemma B.1 and Lemma B.2, and using $\phi'_{q,v_j} \in [\gamma, 1]$ for all $q \in I$, the above is smaller than

$$-\frac{1}{\sqrt{m}} \cdot \frac{1}{2R_{\max}^2} \cdot \gamma R_{\min}^2 + \frac{1}{\sqrt{m}} \cdot n \cdot \frac{3}{2\gamma^2 R_{\min}^2} \cdot p \leq -\frac{\gamma R_{\min}^2}{2\sqrt{m}R_{\max}^2} + \frac{1}{\sqrt{m}} \cdot \frac{\gamma^3}{3} \cdot \frac{R_{\min}^2}{p} \cdot \frac{R_{\min}^2}{R_{\max}^2} \cdot \frac{3}{2\gamma^2 R_{\min}^2} \cdot p$$

$$= -\frac{\gamma R_{\min}^2}{2\sqrt{m}R_{\max}^2} + \frac{\gamma R_{\min}^2}{2\sqrt{m}R_{\max}^2} = 0 \ .$$

Thus, $v_j^\top x_i < 0$, which implies $\phi'_{i,v_j} = \gamma$.

Using (6) again we conclude that

$$v_j = \frac{1}{\sqrt{m}} \sum_{i \in I} \lambda_i y_i \phi'_{i,v_j} x_i = \frac{1}{\sqrt{m}} \sum_{i \in I_+} \lambda_i x_i - \frac{\gamma}{\sqrt{m}} \sum_{i \in I_-} \lambda_i x_i \ .$$

Since the above expression holds for all $j \in [m_1]$ then we have $v_1 = \ldots = v_{m_1}$.

By similar arguments (which we do not repeat for conciseness) we also get

$$u_1 = \ldots = u_{m_2} = \frac{1}{\sqrt{m}} \sum_{i \in I_-} \lambda_i x_i - \frac{\gamma}{\sqrt{m}} \sum_{i \in I_+} \lambda_i x_i \ .$$

and $y_i u_j^\top x_i < 0$ for all $i \in I$ and $j \in [m_2]$. $\qquad\square$

By the above lemma, we may denote $v := v_1 = \ldots = v_{m_1}$ and $u := u_1 = \ldots = u_{m_2}$, and denote $z := \frac{m_1}{\sqrt{m}} v - \frac{m_2}{\sqrt{m}} u$.

**Lemma B.5.** *The pair $v, u$ is a unique global optimum of the Problem (3).*

*Proof.* First, we remark that a variant of the this lemma appears in Sarussi et al. (2021). They proved the claim under an assumption called *Neural Agreement Regime (NAR)*, and Lemma B.4 implies that this assumption holds in our setting.

Note that the objective in Problem (3) is strictly convex and the constraints are affine. Hence, its KKT conditions are sufficient for global optimality, and the global optimum is unique. It remains to show that $v, u$ satisfy the KKT conditions.

Firstly, note that $v, u$ satisfy the constraints. Indeed, by Lemma B.4, for every $i \in I_+$ we have $v^\top x_i > 0$ and $u^\top x_i < 0$. Combining it with Lemma B.3 we get

$$1 = f(x_i; W) = \frac{m_1}{\sqrt{m}} \phi(v^\top x_i) - \frac{m_2}{\sqrt{m}} \phi(u^\top x_i) = \frac{m_1}{\sqrt{m}} v^\top x_i - \gamma \frac{m_2}{\sqrt{m}} u^\top x_i \ . \tag{8}$$

Similarly, for every $i \in I_-$ we have $v^\top x_i < 0$ and $u^\top x_i > 0$. Together with Lemma B.3 we get

$$-1 = f(x_i; W) = \frac{m_1}{\sqrt{m}}\phi(v^\top x_i) - \frac{m_2}{\sqrt{m}}\phi(u^\top x_i) = \gamma\frac{m_1}{\sqrt{m}}v^\top x_i - \frac{m_2}{\sqrt{m}}u^\top x_i \,. \tag{9}$$

Next, we need to show that there are $\mu_1, \ldots, \mu_n \geq 0$ such that

$$m_1 v = \sum_{i \in I_+} \mu_i \frac{m_1}{\sqrt{m}} x_i + \sum_{i \in I_-} \mu_i \left(-\gamma \frac{m_1}{\sqrt{m}} x_i\right) \,,$$

$$m_2 u = \sum_{i \in I_+} \mu_i \left(-\gamma \frac{m_2}{\sqrt{m}} x_i\right) + \sum_{i \in I_-} \mu_i \frac{m_2}{\sqrt{m}} x_i \,.$$

By setting $\mu_i = \lambda_i$ for all $i \in I$, Lemma B.4 implies that the above equations hold.

Finally, we need to show that $\mu_i = 0$ for all $i \in I$ where the corresponding constraint holds with a strict inequality. However, by (8) and (9) all constraints hold with an equality. $\qquad\square$

**Lemma B.6.** *The weight matrix $W$ is a unique global optimum of Problem (1).*

*Proof.* Let $\tilde{W}$ be a weight matrix that satisfies the KKT conditions of Problem (1), and let $\tilde{v}_1, \ldots, \tilde{v}_{m_1}, \tilde{u}_1, \ldots, \tilde{u}_{m_2}$ be the corresponding positive and negative weight vectors. We first show that $\tilde{W} = W$, i.e., there is a unique KKT point for Problem (1). Indeed, by Lemma B.4, for every such $\tilde{W}$ we have $\tilde{v}_1 = \ldots = \tilde{v}_{m_1} := \tilde{v}$ and $\tilde{u}_1 = \ldots = \tilde{u}_{m_2} := \tilde{u}$, and by Lemma B.5 the vectors $\tilde{v}, \tilde{u}$ are a unique global optimum of Problem (3). Since by Lemma B.5 the vectors $v, u$ are also a unique global optimum of Problem (3), then we must have $v = \tilde{v}$ and $u = \tilde{u}$.

Now, let $W^*$ be a global optimum of Problem (1). By Lyu & Li (2019), the KKT conditions of this problem are necessary for optimality, and hence they are satisfied by $W^*$. Therefore, we have $W^* = W$. Thus, $W$ is a unique global optimum. $\qquad\square$

**Lemma B.7.** *For every $x \in \mathbb{R}^d$ we have $\mathrm{sign}\,(f(x; W)) = \mathrm{sign}(z^\top x)$.*

*Proof.* First, We remark that a variant of the this lemma appears in Sarussi et al. (2021). They proved the claim under an assumption called *Neural Agreement Regime (NAR)*, and Lemma B.4 implies that this assumption holds in our setting.

Let $x \in \mathbb{R}^d$. Consider the following cases:

**Case 1:** If $v^\top x \geq 0$ and $u^\top x \geq 0$ then $f(x; W) = \frac{m_1}{\sqrt{m}}v^\top x - \frac{m_2}{\sqrt{m}}u^\top x = z^\top x$, and thus $\mathrm{sign}\,(f(x; W)) = \mathrm{sign}(z^\top x)$.

**Case 2:** If $v^\top x \geq 0$ and $u^\top x < 0$ then $f(x; W) = \frac{m_1}{\sqrt{m}}v^\top x - \frac{m_2}{\sqrt{m}}\gamma u^\top x > 0$ and $z^\top x = \frac{m_1}{\sqrt{m}}v^\top x - \frac{m_2}{\sqrt{m}}u^\top x > 0$.

**Case 3:** If $v^\top x < 0$ and $u^\top x \geq 0$ then $f(x; W) = \frac{m_1}{\sqrt{m}}\gamma v^\top x - \frac{m_2}{\sqrt{m}}u^\top x < 0$ and $z^\top x = \frac{m_1}{\sqrt{m}}v^\top x - \frac{m_2}{\sqrt{m}}u^\top x < 0$.

**Case 4:** If $v^\top x < 0$ and $u^\top x < 0$ then $f(x; W) = \frac{m_1}{\sqrt{m}}\gamma v^\top x - \frac{m_2}{\sqrt{m}}\gamma u^\top x = \gamma z^\top x$, and thus $\mathrm{sign}\,(f(x; W)) = \mathrm{sign}(z^\top x)$. $\qquad\square$

**Lemma B.8.** *The vector $z$ may not be an $\ell_2$-max-margin linear predictor.*

*Proof.* We give an example of a setting that satisfies the theorem's assumptions, but the corresponding vector $z$ is not an $\ell_2$-max-margin linear predictor. Let $\gamma = \frac{1}{2}$ and suppose that $m_1 = m_2 := m'$. Let $x_1 = (-1, 0, 0)^\top$, $x_2 = (\epsilon, \sqrt{1 - \epsilon^2}, 0)^\top$, and $x_3 = (0, 0, 1)^\top$, where $\epsilon > 0$ is sufficiently small such that the theorem's assumption holds. Namely, since we need $n \leq \frac{\gamma^3}{3} \cdot \frac{R_{\min}^2}{p} \cdot \frac{R_{\min}^2}{R_{\max}^2}$ and we have $R_{\min} = R_{\max} = 1$ and $p = \epsilon$, then $\epsilon$ should satisfy $3 \leq \frac{1}{8 \cdot 3\epsilon}$. We also let $y_1 = -1$, $y_2 = y_3 = 1$. Let $W$ be a KKT point of Problem (1) w.r.t. the dataset $\{(x_i, y_i)\}_{i=1}^3$, and let $v_1, \ldots, v_{m'}, u_1, \ldots, u_{m'}$ be the corresponding weight vectors. By Lemma B.4 and Lemma B.5 we have $v = v_1 = \ldots = v_{m'}$

and $u = u_1 = \ldots = u_{m'}$ where $v, u$ are a solution of Problem (3). Moreover, by Lemma B.4 and Lemma B.2 we have

$$v = \frac{1}{\sqrt{2m'}} \left( \lambda_2 x_2 + \lambda_3 x_3 - \gamma \lambda_1 x_1 \right) = \frac{1}{\sqrt{2m'}} \left( \lambda_2 x_2 + \lambda_3 x_3 - \frac{1}{2} \cdot \lambda_1 x_1 \right) , \tag{10}$$

$$u = \frac{1}{\sqrt{2m'}} \left( \lambda_1 x_1 - \gamma \lambda_2 x_2 - \gamma \lambda_3 x_3 \right) = \frac{1}{\sqrt{2m'}} \left( \lambda_1 x_1 - \frac{1}{2} \cdot \lambda_2 x_2 - \frac{1}{2} \cdot \lambda_3 x_3 \right) , \tag{11}$$

where $\lambda_i > 0$ for all $i$. Since $x_1, x_2, x_3$ are linearly independent, then given $v, u$ there is a unique choice of $\lambda_1, \lambda_2, \lambda_3$ that satisfy the above equations.

Since $v, u$ satisfy the KKT conditions of Problem (3), we can find $\lambda_1, \lambda_2, \lambda_3$ as follows. Let $\mu_1, \mu_2, \mu_3 \geq 0$ be such that the KKT conditions of Problem (3) hold. From the stationarity condition we have

$$m'v = \mu_2 \frac{m'}{\sqrt{2m'}} x_2 + \mu_3 \frac{m'}{\sqrt{2m'}} x_3 - \gamma \mu_1 \frac{m'}{\sqrt{2m'}} x_1 ,$$

$$m'u = \mu_1 \frac{m'}{\sqrt{2m'}} x_1 - \gamma \mu_2 \frac{m'}{\sqrt{2m'}} x_2 - \gamma \mu_3 \frac{m'}{\sqrt{2m'}} x_3 .$$

Since $x_1, x_2, x_3$ are linearly independent, combining the above with (10) and (11) implies $\mu_i = \lambda_i > 0$ for all $i$. Therefore, all constraints in Problem (3) must hold with an equality. Namely, we have

$$\frac{\sqrt{2m'}}{m'} = \left( u^\top - \frac{1}{2} v^\top \right) x_1$$

$$= \frac{1}{\sqrt{2m'}} \left[ \lambda_1 x_1 - \frac{1}{2} \cdot \lambda_2 x_2 - \frac{1}{2} \cdot \lambda_3 x_3 - \frac{1}{2} \left( \lambda_2 x_2 + \lambda_3 x_3 - \frac{1}{2} \cdot \lambda_1 x_1 \right) \right]^\top x_1$$

$$= \frac{1}{\sqrt{2m'}} \left( \frac{5}{4} \cdot \lambda_1 x_1 - \lambda_2 x_2 - \lambda_3 x_3 \right)^\top x_1 = \frac{1}{\sqrt{2m'}} \left( \frac{5}{4} \cdot \lambda_1 \cdot 1 - \lambda_2(-\epsilon) - \lambda_3 \cdot 0 \right)$$

$$= \frac{1}{\sqrt{2m'}} \left( \frac{5}{4} \cdot \lambda_1 + \lambda_2 \epsilon \right) ,$$

$$\frac{\sqrt{2m'}}{m'} = \left( v^\top - \frac{1}{2} u^\top \right) x_2$$

$$= \frac{1}{\sqrt{2m'}} \left[ \lambda_2 x_2 + \lambda_3 x_3 - \frac{1}{2} \cdot \lambda_1 x_1 - \frac{1}{2} \left( \lambda_1 x_1 - \frac{1}{2} \cdot \lambda_2 x_2 - \frac{1}{2} \cdot \lambda_3 x_3 \right) \right]^\top x_2$$

$$= \frac{1}{\sqrt{2m'}} \left( \frac{5}{4} \cdot \lambda_2 x_2 + \frac{5}{4} \cdot \lambda_3 x_3 - \lambda_1 x_1 \right)^\top x_2 = \frac{1}{\sqrt{2m'}} \left( \frac{5}{4} \cdot \lambda_2 + 0 - \lambda_1(-\epsilon) \right)$$

$$= \frac{1}{\sqrt{2m'}} \left( \frac{5}{4} \cdot \lambda_2 + \lambda_1 \epsilon \right) ,$$

and

$$\frac{\sqrt{2m'}}{m'} = \left( v^\top - \frac{1}{2} u^\top \right) x_3 = \frac{1}{\sqrt{2m'}} \left( \frac{5}{4} \cdot \lambda_2 x_2 + \frac{5}{4} \cdot \lambda_3 x_3 - \lambda_1 x_1 \right)^\top x_3 = \frac{1}{\sqrt{2m'}} \cdot \frac{5}{4} \cdot \lambda_3 .$$

Solving the above equations, we get $\lambda_1 = \lambda_2 = \frac{8}{4\epsilon+5}$, and $\lambda_3 = \frac{8}{5}$.

Thus, a KKT point of Problem (1) must satisfy (10) and (11) with the above $\lambda_i$'s. Now, consider

$$
\begin{aligned}
z &= \frac{m'}{\sqrt{2m'}} v - \frac{m'}{\sqrt{2m'}} u = \frac{m'}{\sqrt{2m'}} (v - u) \\
&= \frac{m'}{\sqrt{2m'}} \cdot \frac{1}{\sqrt{2m'}} \left( \sum_{i \in I_+} \lambda_i x_i - \gamma \sum_{i \in I_-} \lambda_i x_i - \sum_{i \in I_-} \lambda_i x_i + \gamma \sum_{i \in I_+} \lambda_i x_i \right) \\
&= \frac{1 + \gamma}{2} \left( \sum_{i \in I_+} \lambda_i x_i - \sum_{i \in I_-} \lambda_i x_i \right) \\
&= \frac{3}{4} \left( \frac{8}{4\epsilon + 5} \cdot x_2 + \frac{8}{5} \cdot x_3 - \frac{8}{4\epsilon + 5} \cdot x_1 \right) \\
&= \left( \frac{6}{4\epsilon + 5} \cdot x_2 + \frac{6}{5} \cdot x_3 - \frac{6}{4\epsilon + 5} \cdot x_1 \right) .
\end{aligned}
$$

We need to show that $z$ does not satisfy the KKT conditions of the problem

$$
\min_{\tilde{z}} \frac{1}{2} \|\tilde{z}\|^2 \quad \text{s.t.} \quad \forall i \in \{1, 2, 3\} \quad y_i \tilde{z}^\top x_i \geq \beta , \tag{12}
$$

for any margin $\beta > 0$. A KKT point $\tilde{z}$ of the above problem must satisfy $\tilde{z} = -\lambda_1' x_1 + \lambda_2' x_2 + \lambda_3' x_3$, where $\lambda_i' \geq 0$ for all $i$, and $\lambda_i' = 0$ if $y_i \tilde{z}^\top x_i \neq \beta$. Since $z$ is a linear combination of the three independent vectors $x_1, x_2, x_3$ where the coefficients are non-zero, then if $z$ is a KKT point of Problem (12) we must have $\lambda_i' \neq 0$ for all $i$, which implies $y_i z^\top x_i = \beta$ for all $i$. Therefore, in order to conclude that $z$ is not a KKT point, it suffices to show that $z^\top x_2 \neq z^\top x_3$.

We have

$$
z^\top x_2 = \left( \frac{6}{4\epsilon + 5} \cdot x_2 + \frac{6}{5} \cdot x_3 - \frac{6}{4\epsilon + 5} \cdot x_1 \right)^\top x_2 = \frac{6}{4\epsilon + 5} + 0 + \frac{6\epsilon}{4\epsilon + 5} = \frac{6(\epsilon + 1)}{4\epsilon + 5} ,
$$

and

$$
z^\top x_3 = \left( \frac{6}{4\epsilon + 5} \cdot x_2 + \frac{6}{5} \cdot x_3 - \frac{6}{4\epsilon + 5} \cdot x_1 \right)^\top x_3 = \frac{6}{5} .
$$

Using the above equations, it is easy to verify that $z^\top x_2 \neq z^\top x_3$ for all $\epsilon > 0$. $\qquad \square$

**Lemma B.9.** *For all $i \in I$ we have $y_i z^\top x_i \geq 1$, and $\|z\| \leq \frac{2}{\kappa + \gamma} \|z^*\|$, where $z^* = \operatorname{argmin}_{\tilde{z}} \|\tilde{z}\|$ s.t. $y_i \tilde{z}^\top x_i \geq 1$ for all $i \in I$.*

*Proof.* By Lemma B.4, for all $i \in I_+$ we have $v^\top x_i > 0$ and $u^\top x_i < 0$. Hence

$$
\begin{aligned}
1 \leq f(x_i; W) &= \frac{m_1}{\sqrt{m}} \phi(v^\top x_i) - \frac{m_2}{\sqrt{m}} \phi(u^\top x_i) \\
&= \frac{m_1}{\sqrt{m}} v^\top x_i - \frac{m_2}{\sqrt{m}} \gamma u^\top x_i \\
&\leq \frac{m_1}{\sqrt{m}} v^\top x_i - \frac{m_2}{\sqrt{m}} u^\top x_i = z^\top x_i .
\end{aligned}
$$

Likewise, by Lemma B.4, for all $i \in I_-$ we have $v^\top x_i < 0$ and $u^\top x_i > 0$. Hence

$$
\begin{aligned}
-1 \geq f(x_i; W) &= \frac{m_1}{\sqrt{m}} \phi(v^\top x_i) - \frac{m_2}{\sqrt{m}} \phi(u^\top x_2) \\
&= \frac{m_1}{\sqrt{m}} \gamma v^\top x_i - \frac{m_2}{\sqrt{m}} u^\top x_i \\
&\geq \frac{m_1}{\sqrt{m}} v^\top x_i - \frac{m_2}{\sqrt{m}} u^\top x_i = z^\top x_i .
\end{aligned}
$$

Thus, it remains to obtain an upper bound for $\|z\|$.

Assume w.l.o.g. that $m_1 \geq m_2$ (the proof for the case $m_1 \leq m_2$ is similar). Thus, $\kappa = \sqrt{\frac{m_2}{m_1}}$. Let $z^* \in \mathbb{R}^d$ such that $y_i(z^*)^\top x_i \geq 1$ for all $i \in I$. Let

$$v^* = z^* \cdot \frac{\sqrt{m}}{m_1} \cdot \frac{1}{\kappa + \gamma} ,$$

$$u^* = -z^* \cdot \frac{\sqrt{m}}{m_2} \cdot \frac{\kappa}{\kappa + \gamma} .$$

Note that $v^*, u^*$ satisfy the constraints in Problem (3). Indeed, for $i \in I_-$ we have

$$\frac{m_2}{\sqrt{m}}(u^*)^\top x_i - \gamma \frac{m_1}{\sqrt{m}}(v^*)^\top x_i = -\frac{\kappa(z^*)^\top x_i}{\kappa + \gamma} - \gamma \cdot \frac{(z^*)^\top x_i}{\kappa + \gamma} \geq \frac{\kappa}{\kappa + \gamma} + \gamma \cdot \frac{1}{\kappa + \gamma} = 1 .$$

For $i \in I_+$ we have

$$\frac{m_1}{\sqrt{m}}(v^*)^\top x_i - \gamma \frac{m_2}{\sqrt{m}}(u^*)^\top x_i = \frac{(z^*)^\top x_i}{\kappa + \gamma} + \gamma \cdot \frac{\kappa(z^*)^\top x_i}{\kappa + \gamma} \geq \frac{1}{\kappa + \gamma} + \gamma \cdot \frac{\kappa}{\kappa + \gamma} = \frac{1 + \gamma \kappa}{\kappa + \gamma} \geq 1 ,$$

where the last inequality is since $0 \leq (1 - \kappa)(1 - \gamma) = 1 + \kappa \gamma - \kappa - \gamma$.

By Lemma B.5 the pair $v, u$ is a global optimum of Problem (3). Hence

$$m_1 \|v\|^2 + m_2 \|u\|^2 \leq m_1 \|v^*\|^2 + m_2 \|u^*\|^2$$

$$= m_1 \cdot \frac{m}{m_1^2} \cdot \frac{1}{(\kappa + \gamma)^2} \|z^*\|^2 + m_2 \cdot \frac{m}{m_2^2} \cdot \frac{\kappa^2}{(\kappa + \gamma)^2} \|z^*\|^2$$

$$= \frac{m \|z^*\|^2}{(\kappa + \gamma)^2} \left[ \frac{1}{m_1} + \frac{\kappa^2}{m_2} \right]$$

$$= \frac{m \|z^*\|^2}{(\kappa + \gamma)^2} \cdot \frac{2}{m_1} .$$

Therefore, we have

$$\|m_1 v\|^2 + \|m_2 u\|^2 \leq m_1^2 \|v\|^2 + m_1 m_2 \|u\|^2 \leq \frac{m \|z^*\|^2}{(\kappa + \gamma)^2} \cdot 2 .$$

Hence,

$$\|z\|^2 = \left\| \frac{m_1}{\sqrt{m}} v - \frac{m_2}{\sqrt{m}} u \right\|^2 \leq 2 \left( \left\| \frac{m_1}{\sqrt{m}} v \right\|^2 + \left\| \frac{m_2}{\sqrt{m}} u \right\|^2 \right) = \frac{2}{m} \left( \|m_1 v\|^2 + \|m_2 u\|^2 \right) \leq \frac{4 \|z^*\|^2}{(\kappa + \gamma)^2} ,$$

which implies $\|z\| \leq \frac{2\|z^*\|}{\kappa + \gamma}$ as required. $\square$

## C  PROOF OF LEMMA 3.3

*Proof of Lemma 3.3.* According to the distribution assumption in the lemma, we can write $x_i = \Sigma^{1/2} \bar{x}_i$ where $\bar{x}_i \sim \mathsf{N}(0, I_d)$.[2] By Hanson-Wright inequality (Rudelson & Vershynin, 2013, Theorem 2.1), we have for any $t \geq 0$,

$$\Pr \left[ \left| \left\| \Sigma^{1/2} \bar{x}_i \right\| - \|\Sigma^{1/2}\|_F \right| > t \right] \leq 2 \exp \left( -\Omega \left( \frac{t^2}{\left\| \Sigma^{1/2} \right\|_2^2} \right) \right) ,$$

i.e.,

$$\Pr \left[ \left| \|x_i\| - \sqrt{d} \right| > t \right] \leq 2 \exp \left( -\Omega \left( t^2 \right) \right) .$$

Let $t = C\sqrt{\log n}$ for a sufficiently large constant $C > 0$. Taking a union bound over all $i \in [n]$, we have that with probability at least $1 - n^{-20}$, $\|x_i\| = \sqrt{d} \pm O(\sqrt{\log n})$ for all $i \in [n]$ simultaneously.

---

[2] The proof below holds more generally when $\bar{x}_i$ has independent subgaussian entries.

For $i \neq j$, we have $\langle x_i, x_j \rangle | x_j \sim \mathsf{N}(0, x_j^\top \Sigma x_j)$. Hence we can apply a standard tail bound to obtain

$$\Pr\left[|\langle x_i, x_j \rangle| > t \,|\, x_j\right] \leq 2 \exp\left(-\frac{t^2}{2 x_j^\top \Sigma x_j}\right).$$

Because we have known that $x_j^\top \Sigma x_j = O(\|x_j\|^2) = O(d + \log n) = O(d)$ with probability at least $1 - n^{-20}$, we have

$$\Pr\left[|\langle x_i, x_j \rangle| > t\right] \leq n^{-20} + 2 \exp\left(-\Omega\left(\frac{t^2}{d}\right)\right).$$

Then we can take $t = C\sqrt{d \log n}$ for a sufficiently large constant $C$ and apply a union bound over all $i, j$, which gives $|\langle x_i, x_j \rangle| = O(\sqrt{d \log n})$ for all $i \neq j$ with probability at least

$$1 - n^2 \left(n^{-20} + 2\exp(-\Omega(C^2 \log n))\right) \geq 1 - n^{-15}.$$

This completes the proof. $\qquad\qquad\qquad\qquad\qquad\qquad\qquad\qquad\qquad\qquad\qquad\qquad\square$

## D  PROOF OF THEOREM 3.4

To prove Theorem 3.4, we need to show that for some $t_0 > 0$, $\widehat{L}(W(t)) < \log 2/n$ for all $t \geq t_0$. To do so, we will first show a proxy PL inequality (Frei & Gu, 2021), and then use this to argue that the loss must eventually be smaller than $\log 2/n$.

We begin by showing that the vector $\widehat{\mu} := \sum_{i=1}^n y_i x_i$ correctly classifies the training data with a positive margin. To see this, note that for any $k \in [n]$,

$$
\begin{aligned}
\left\langle \sum_{i=1}^n y_i x_i, y_k x_k \right\rangle &= \|x_k\|^2 + \sum_{i \neq k} \langle y_i x_i, y_k x_k \rangle \\
&\geq \min_i \|x_i\|^2 - n \max_{i \neq j} |\langle x_i, x_j \rangle| \\
&\overset{(i)}{\geq} \left(1 - \frac{\gamma^3}{3}\right) \min_i \|x_i\|^2 \\
&\overset{(ii)}{\geq} \frac{2}{3} \min_i \|x_i\|^2.
\end{aligned}
\tag{13}
$$

Inequality $(i)$ uses the theorem's assumption that $3n \max_{i \neq j} |\langle x_i, x_j \rangle| \leq \gamma^3$. Inequality $(ii)$ uses that $\gamma \leq 1$. To show how large of a margin $\widehat{\mu}$ gets on the training data, we bound its norm. We have,

$$
\begin{aligned}
\left\| \sum_{i=1}^n y_i x_i \right\|^2 &\leq \sum_{i=1}^n \|x_i\|^2 + \sum_{i \neq j} |\langle x_i, x_j \rangle| \\
&= \sum_{i=1}^n \left[ \|x_i\|^2 + \sum_{j \neq i} |\langle x_i, x_j \rangle| \right] \\
&\leq \sum_{i=1}^n \left[ \|x_i\|^2 + n \max_{i \neq j} |\langle x_i, x_j \rangle| \right] \\
&\leq \sum_{i=1}^n \left[ \|x_i\|^2 + \frac{\gamma^3}{3} \min_j \|x_j\|^2 \right] \\
&\leq 2n \max_i \|x_i\|^2.
\end{aligned}
$$

Denoting $R_{\min} := \min_i \|x_i\|$, $R_{\max} = \max_i \|x_i\|$, and $R = R_{\max}/R_{\min}$, substituting the above display into (13) we get for any $k \in [n]$,

$$\left\langle \frac{\widehat{\mu}}{\|\widehat{\mu}\|}, y_k x_k \right\rangle \geq \frac{2/3 R_{\min}^2}{\sqrt{2n R_{\max}^2}} = \frac{\sqrt{2} R_{\min}}{3 R \sqrt{n}}. \tag{14}$$

Let us now define the matrix $Z \in \mathbb{R}^{m \times d}$ with rows,

$$z_j := \frac{\widehat{\mu}}{\|\widehat{\mu}\|} a_j.$$

Since $a_j^2 = 1/m$ for each $j$, we have $\|Z\|_F^2 = 1$, and moreover we have for any $k \in [n]$ and $W \in \mathbb{R}^{m \times d}$,

$$\begin{aligned}
y_k \langle \nabla f(x_k; W), Z \rangle &= \sum_{j=1}^{m} a_j^2 \phi'(\langle w_j, x_k \rangle) \left\langle \frac{\widehat{\mu}}{\|\widehat{\mu}\|}, y_k x_k \right\rangle \\
&\geq \frac{\sqrt{2} R_{\min}}{3 R \sqrt{n}} \frac{1}{m} \sum_{j=1}^{m} \phi'(\langle w_j, x_k \rangle) \\
&\geq \frac{\sqrt{2} R_{\min} \gamma}{3 R \sqrt{n}},
\end{aligned}$$

where the first inequality uses (14) and the last inequality uses that $\phi'(z) \geq \gamma$. If $\ell$ is the logistic or exponential loss and we define

$$g(z) = -\ell'(z), \quad \widehat{G}(W(t)) := \frac{1}{n} \sum_{k=1}^{n} g(y_k f(x_k; W(t))),$$

then since $g(z) > 0$ the above allows for the following proxy-PL inequality,

$$\begin{aligned}
\|\nabla \widehat{L}(W(t))\|_F &\geq \left\langle \nabla \widehat{L}(W(t)), -Z \right\rangle \\
&= \frac{1}{n} \sum_{k=1}^{n} -\ell'(y_k f(x_k; W(t))) y_k \langle \nabla f(x_k; W(t)), Z \rangle \\
&\geq \frac{\sqrt{2} R_{\min} \gamma}{3 R \sqrt{n}} \widehat{G}(W(t)).
\end{aligned} \tag{15}$$

By the chain rule, the above implies

$$\begin{aligned}
\frac{\mathrm{d}}{\mathrm{d}t} \widehat{L}(W(t)) &= -\|\nabla \widehat{L}(W(t))\|_F^2 \\
&\leq -\left( \frac{\sqrt{2} R_{\min} \gamma}{3 R \sqrt{n}} \widehat{G}(W(t)) \right)^2.
\end{aligned}$$

Let us now calculate how long until we reach the point where $\widehat{G}(W(t)) < \log 2/(3n)$. Define

$$\tau = \inf\{t : \widehat{G}(W(t)) < \log 2/(3n)\}.$$

Then for any $t < \tau$ we have

$$\frac{\mathrm{d}}{\mathrm{d}t} \widehat{L}(W(t)) \leq -\left( \frac{\sqrt{2} R_{\min} \gamma}{3 R \sqrt{n}} \cdot \frac{\log 2}{3n} \right)^2.$$

Integrating, we see that

$$\widehat{L}(W(t)) \leq \widehat{L}(W(0)) - \frac{2 R_{\min}^2 \gamma^2 \log^2(2) t}{81 R^2 n^3}.$$

Since $\widehat{L}(W(t)) \geq 0$, this means that $\tau \leq 81 \widehat{L}(W(0)) R^2 n^3 / (2 \gamma^2 R_{\min}^2 \log^2(2)) \leq 85 \widehat{L}(W(0)) R^2 n^3 / (\gamma^2 R_{\min}^2)$. At time $\tau$, we know that $\widehat{G}(W(\tau)) \leq \log 2/(3n)$ and thus $y_i f(x_i; W(\tau)) > 0$ for each $i$. For $z > 0$, both the logistic loss and the exponential loss satisfy $\ell(z) \leq 2 \cdot -\ell'(z)$, and so for either loss, we have

$$\widehat{L}(W(\tau)) = \frac{1}{n} \sum_{i=1}^{n} \ell(y_i f(x_i; W(\tau))) \leq \frac{2}{n} \sum_{i=1}^{n} -\ell'(y_i f(x_i; W(\tau))) = 2\widehat{G}(W(\tau)) \leq \frac{2}{3} \cdot \frac{\log 2}{n}.$$

Since $\widehat{L}(W(t))$ is decreasing, we thus have for all times $t \geq \tau$, we have $\widehat{L}(W(t)) \leq \widehat{L}(W(\tau)) < \log(2)/n$.

# E  PROOF OF THEOREM 4.2

In this section, we provide a proof of Theorem 4.2. An overview of our proof is as follows.

1. In Section E.1 we provide basic concentration arguments about the random initialization.

2. In Section E.2 we show that the neural network output and the logistic loss objective function are smooth as a function of the parameters.

3. In Section E.3 we prove a structural result on how gradient descent weights the samples throughout the training trajectory. In particular, we show that throughout gradient descent, the sigmoid losses $-\ell'(y_i f(x_i; W^{(t)}))$ grow at approximately the same rate for all samples.

4. In Section E.4 we leverage the above structural result to provide a tighter upper bound on $\|W^{(t)}\|_F$ than is possible with a naïve application of the triangle inequality.

5. In Section E.5 we provide a lower bound for $\|W^{(t)}\|_2$.

6. In Section E.6 we show that a proxy-PL inequality is satisfied.

7. We conclude the proof of Theorem 4.2 in Section E.7 by putting together the preceding items to bound the stable rank $\mathsf{StableRank}(W^{(t)}) = \|W^{(t)}\|_F^2 / \|W^{(t)}\|_2^2$ and to show that $\widehat{L}(W^{(t)}) \to 0$.

Let us denote by $C_R := 10R^2/\gamma^2 + 10$, where $R = R_{\max}/R_{\min}$ and $R_{\max} = \max_i \|x_i\|$, $R_{\min} = \min_i \|x_i\|$. For a given probability threshold $\delta \in (0, 1)$, we make the following assumptions moving forward:

(A1) Step-size $\alpha \leq \gamma^2 \left(5nR_{\max}^2 R^2 C_R \max(1, H)\right)^{-1}$, where $\phi$ is $H$-smooth and $\gamma$-leaky.

(A2) Initialization variance satisfies $\omega_{\mathrm{init}} \leq \alpha \gamma^2 R_{\min} \left(72RC_R n \sqrt{md \log(4m/\delta)}\right)^{-1}$.

We shall also use the following notation to refer to the sigmoid losses that appear throughout the analysis of gradient descent training for the logistic loss,

$$g(z) = -\ell'(z) = \frac{1}{1 + \exp(z)}, \quad \widehat{G}(W) = \frac{1}{n}\sum_{i=1}^{n} g\big(y_i f(x_i; W)\big), \quad g_i^{(t)} := g\big(y_i f(x_i; W^{(t)})\big).$$

(16)

## E.1  CONCENTRATION FOR RANDOM INITIALIZATION

The following lemma characterizes the $\ell_2$-norm of each neuron at intialization. It also characterizes how large the projection of each neuron along the direction $\widehat{\mu} := \sum_{i=1}^{n} y_i x_i$ can be at initialization. We shall see in Lemma E.13 that gradient descent forces the weights to align with this direction. In the proof of Theorem 4.2, we will argue that by taking a single step of gradient descent with a sufficiently large step-size and small initialization variance, the gradient descent update dominates the behavior of each neuron at initialization, so that after one step the $\widehat{\mu}$ direction becomes dominant for each neuron. This will form the basis of showing that $W^{(t)}$ has small stable rank for $t \geq 1$.

**Lemma E.1.** *With probability at least $1-\delta$ over the random initialization, the following holds. First, we have the following upper bounds for the spectral norm and per-neuron norms at initialization,*

$$\|W^{(0)}\|_2 \leq C_0 \omega_{\mathrm{init}}(\sqrt{m} + \sqrt{d}), \quad and \quad for\ all\ j \in [m],\ \|w_j^{(0)}\|^2 \leq 5\omega_{\mathrm{init}}^2 d \log(4m/\delta).$$

*Second, if we denote by $\bar{\mu} \in \mathbb{R}^d$ be the vector $\sum_{i=1}^{n} y_i x_i / \|\sum_{i=1}^{n} y_i x_i\|$, then we have*

$$|\langle w_j^{(0)}, \bar{\mu}\rangle| \leq 2\omega_{\mathrm{init}}\sqrt{\log(4m/\delta)}.$$

*Proof.* For the first part of the lemma, note that for fixed $j \in [m]$, there are i.i.d. $z_i \sim \mathsf{N}(0, 1)$ such that

$$\|w_j^{(0)}\|^2 = \sum_{i=1}^{d}(w_j^{(0)})_i^2 = \omega_{\mathrm{init}}^2 \sum_{i=1}^{d} z_i^2 \sim \omega_{\mathrm{init}}^2 \cdot \chi^2(d).$$

By concentration of the $\chi^2$ distribution (Laurent & Massart, 2000, Lemma 1), for any $t > 0$,

$$\mathbb{P}\left(\frac{1}{\omega_{\text{init}}^2}\|w_j^{(0)}\|^2 - d \geq 2\sqrt{dt} + 2t\right) \leq \exp(-t).$$

In particular, if we let $t = \log(4m/\delta)$, we have that with probability at least $1 - \delta/4$, for all $j \in [m]$,

$$\|w_j^{(0)}\|^2 \leq \omega_{\text{init}}^2\left(d + 2\sqrt{d\log(4m/\delta)} + 2\log(4m/\delta)\right) \leq 5\omega_{\text{init}}^2 d\log(4m/\delta).$$

For the second part, note that $\langle w_j^{(0)}, \bar{\mu}\rangle \sim \mathsf{N}(0, \omega_{\text{init}}^2)$. We therefore have $\mathbb{P}(|\langle w_j^{(0)}, \bar{\mu}\rangle| \geq t) \leq 2\exp(-t^2/2\omega_{\text{init}}^2)$. Choosing $t = \omega_{\text{init}}\sqrt{\log(4m/\delta)}$ we see that with probability at least $1 - \delta/2$, for all $j$, $|\langle w_j^{(0)}, \bar{\mu}\rangle| \leq 2\omega_{\text{init}}\sqrt{\log(4m/\delta)}$. Taking a union bound over both events completes the proof. $\qquad\square$

## E.2 SMOOTHNESS OF NETWORK OUTPUT AND LOSS

In this sub-section, we show that the network output and the logistic loss satisfy a number of smoothness properties, owing to the fact that $\phi$ is $H$-smooth (i.e., $\phi''$ exists and $|\phi''(z)| \leq H$).

**Lemma E.2.** *For an $H$-smooth activation $\phi$ and any $W, V \in \mathbb{R}^{m \times d}$ and $x \in \mathbb{R}^d$,*

$$|f(x; W) - f(x; V) - \langle\nabla f(x; V), W - V\rangle| \leq \frac{H\|x\|^2}{2\sqrt{m}}\|W - V\|_2^2.$$

*Proof.* This was shown in Frei et al. (2022a, Lemma 4.5). $\qquad\square$

We next show that the empirical risk is smooth, in the sense that the gradient norm is bounded by the loss itself and that the gradients are Lipschitz.

**Lemma E.3.** *For an $H$-smooth, 1-Lipschitz activation $\phi$ and any $W, V \in \mathbb{R}^{m \times d}$, if $\|x_i\| \leq R_{max}$ for all $i$,*

$$\frac{1}{R_{max}}\|\nabla\widehat{L}(W)\|_F \leq \widehat{G}(W) \leq \widehat{L}(W) \wedge 1,$$

*where $\widehat{G}(W)$ is defined in (16). Additionally,*

$$\|\nabla\widehat{L}(W) - \nabla\widehat{L}(V)\|_F \leq R_{max}^2\left(1 + \frac{H}{\sqrt{m}}\right)\|W - V\|_2.$$

*Proof.* This follows by Frei et al. (2022a, Lemma 4.6). The only difference is that in that paper, the authors use $\|x_i\|^2 \leq C_1 p$ (in their work, $x_i \in \mathbb{R}^p$) to go from equations (5) and (6) to equation (7), while we instead use that $\|x_i\|^2 \leq R_{\max}^2$. $\qquad\square$

## E.3 LOSS RATIO BOUND

In this section, we prove a key structural result which we will refer to as a 'loss ratio bound'.

**Lemma E.4.** *Let $\phi$ be a $\gamma$-leaky, $H$-smooth activation. Define $R = \max_{i,j}\|x_i\|/\|x_j\|$, and let us denote $C_R = 10R^2\gamma^{-2} + 10$. Suppose that for all $i \in [n]$, we have,*

$$\|x_i\|^2 \geq 5\gamma^{-2}C_R n\max_{k \neq i}|\langle x_i, x_k\rangle|.$$

*Then under Assumptions (A1) and (A2), we have with probability at least $1 - \delta$,*

$$\sup_{t \geq 0}\left\{\max_{i,j \in [n]}\frac{\ell'\left(y_i f(x_i; W^{(t)})\right)}{\ell'\left(y_j f(x_j; W^{(t)})\right)}\right\} \leq C_R.$$

This lemma shows that regardless of the relationship between $x$ and $y$, the ratio of the *sigmoid* losses $-\ell'(y_i f(x_i; W^{(t)}))$, where $-\ell'(z) = 1/(1 + \exp(z))$, grows at essentially the same rate for all examples.

Our proof largely follows that used by Frei et al. (2022a), who showed a loss ratio bound for gradient descent-trained two-layer networks with $\gamma$-leaky, $H$-smooth activations when the data comes from a mixture of isotropic log-concave distributions. We generalize their proof technique to accommodate general training data for which the samples are nearly orthogonal in the sense that $\|x_i\|^2 \gg n \max_{k \neq i} |\langle x_i, x_k \rangle|$. Additionally, we provide a more general proof technique that illustrates how a loss ratio bound could hold for activations $\phi$ for which $\phi'(z)$ is not bounded from below by an absolute constant (like the ReLU), as well as for training data which are not necessarily nearly-orthogonal. We begin by describing two conditions which form the basis of this more general proof technique. The first condition concerns near-orthogonality of the *gradients* of the network, rather than the *samples* as in the assumption for Theorem 4.2.

**Condition E.5** (Near-orthogonality of gradients). *We say that* near-orthogonality of gradients holds at time $t$ if, for a some absolute constant $C' > 1$, for any $i \in [n]$,

$$\|\nabla f(x_i; W^{(t)})\|^2 \geq C'n \max_{k \neq i} |\langle \nabla f(x_i; W^{(t)}), \nabla f(x_k; W^{(t)}) \rangle|.$$

Note that for linear classifiers—i.e., $m = 1$ with $\phi(z) = z$—near-orthogonality of gradients is equivalent to near-orthogonality of samples, since in this setting $\nabla f(x_i; W) = x_i$. It is clear that this is a more general condition than near-orthogonality of samples.

The next condition we call *gradient persistence*, which roughly states that the gradients of the network with respect to a sample has large norm whenever that sample has large norm.

**Condition E.6** (Gradient persistence). *We say that* gradient persistence holds at time $t$ *if there is a constant* $c > 0$ *such that for all* $i \in [n]$,

$$\|\nabla f(x_i; W^{(t)})\|_F^2 \geq c\|x_i\|^2.$$

Gradient persistence essentially states that there is no possibility of a 'vanishing gradient' problem.

Next, we show that Lipschitz activation functions that are also 'leaky' in the sense that $\phi'(z) \geq \gamma > 0$ everywhere, allow for both gradient persistence and, when the samples are nearly-orthogonal, near-orthogonality of gradients.

**Fact E.7.** *Suppose $\phi$ is such that $\phi'(z) \in [\gamma, 1]$ for all $z$ for some absolute constant $\gamma > 0$. Suppose that for some $C > \gamma^{-2}$, for all $i \in [n]$ we have,*

$$\|x_i\|^2 \geq Cn \max_{k \neq i} |\langle x_i, x_k \rangle|.$$

*Then for all times $t \geq 0$, the gradients are nearly-orthogonal (Condition E.5) with $C' = C\gamma^2$ and gradient persistence (Condition E.6) holds for $c = \gamma^2$.*

*Proof.* For any samples $i, k \in [n]$ and any $W \in \mathbb{R}^{m \times d}$,

$$\langle \nabla f(x_i; W), \nabla f(x_k; W) \rangle = \langle x_i, x_k \rangle \cdot \frac{1}{m} \sum_{j=1}^m \phi'(\langle w_j, x_i \rangle)\phi'(\langle w_j, x_k \rangle).$$

Since $\phi'(z) \in [\gamma, 1]$ for all $z$, we therefore see that gradient persistence holds with $c = \gamma^2$:

$$\|\nabla f(x_k; W)\|_F^2 = \|x_k\|^2 \cdot \frac{1}{m} \sum_{j=1}^m \phi'(\langle w_j, x_k \rangle)^2 \geq \gamma^2 \|x_k\|^2.$$

Similarly, we see that the gradients are nearly-orthogonal, since

$$Cn \max_{i \neq k} |\langle \nabla f(x_i; W), \nabla f(x_k; W) \rangle| \overset{(i)}{\leq} Cn \max_{i \neq k} |\langle x_i, x_k \rangle| \overset{(ii)}{\leq} \|x_k\|^2 \leq \gamma^{-2} \|\nabla f(x_k; W)\|_F^2,$$

where $(i)$ uses that $\phi$ is 1-Lipschitz and $(ii)$ uses the assumption on the near-orthogonality of the samples. $\square$

We can now begin to prove Lemma E.4. We remind the reader of the notation for the sigmoid loss,

$$g(z) := -\ell'(z) = \frac{1}{1 + \exp(z)}, \qquad g_i^{(t)} := g\big(y_i f(x_i; W^{(t)})\big).$$

We follow the same proof technique of Frei et al. (2022a), whereby in order to control the ratio of the sigmoid losses we show instead that the ratio of the exponential losses is small and that this suffices for showing the sigmoid losses is small. As we mention above, we generalize their analysis to emphasize that near-orthogonality of gradients and gradient persistence suffice for showing the loss ratio does not grow significantly.

**Lemma E.8.** *Denote $R := R_{max}/R_{min}$ where $R_{max} = \max_i \|x_i\|$ and $R_{min} = \min_i \|x_i\|$, and let $\phi$ be an arbitrary 1-Lipschitz and $H$-smooth activation. Suppose that near-orthogonality of gradients (Condition E.5) holds for some $C' > 1$ and gradient persistence (Condition E.6) hold at time $t$ for some $c > 0$. Provided $\alpha \leq [5HR_{max}^2 n(10R^2/c + 10)]^{-1}$ and $C' \geq 25R^2/c + 25$, then for any $i, j \in [n]$ we have,*

$$\frac{\exp\big(-y_i f(x_i; W^{(t+1)})\big)}{\exp\big(-y_j f(x_j; W^{(t+1)})\big)} \leq \frac{\exp\big(-y_i f(x_i; W^{(t)})\big)}{\exp\big(-y_j f(x_j; W^{(t)})\big)}$$
$$\times \exp\left(-\frac{g_j^{(t)} \alpha c R_{min}^2}{n}\left(\frac{g_i^{(t)}}{g_j^{(t)}} - \frac{R^2}{c}\right)\right)$$
$$\times \exp\left(\frac{\alpha R_{max}^2}{(10R^2/c + 10)n} \cdot \widehat{G}(W^{(t)})\right)$$

*Proof.* It suffices to consider $i = 1$ and $j = 2$. For notational simplicity denote

$$A_t := \frac{\exp(-y_1 f(x_1; W^{(t)}))}{\exp(-y_2 f(x_2; W^{(t)}))}.$$

We now calculate the exponential loss ratio between two samples at time $t + 1$ in terms of the exponential loss ratio at time $t$.

Recall the notation $g_i^{(t)} := -\ell'(y_i f(x_i; W^{(t)}))$, and introduce the notation

$$\nabla f_i^{(t)} := \nabla f(x_i; W^{(t)}).$$

We can calculate,

$$
\begin{aligned}
A_{t+1} &= \frac{\exp(-y_1 f(x_1; W^{(t+1)}))}{\exp(-y_2 f(x_2; W^{(t+1)}))} \\
&= \frac{\exp\left(-y_1 f_1\left(W^{(t)} - \alpha \nabla \widehat{L}(W^{(t)})\right)\right)}{\exp\left(-y_2 f_2\left(W^{(t)} - \alpha \nabla \widehat{L}(W^{(t)})\right)\right)} \\
&\overset{(i)}{\leq} \frac{\exp\left(-y_1 f\left(x_1; W^{(t)}\right) + y_1 \alpha \left\langle \nabla f_1^{(t)}, \nabla \widehat{L}(W^{(t)})\right\rangle\right)}{\exp\left(-y_2 f\left(x_2; W^{(t)}\right) + y_2 \alpha \left\langle \nabla f_2^{(t)}, \nabla \widehat{L}(W^{(t)})\right\rangle\right)} \exp\left(\frac{H R_{\max}^2 \alpha^2}{\sqrt{m}} \|\nabla \widehat{L}(W^{(t)})\|^2\right) \\
&\overset{(ii)}{=} A_t \cdot \frac{\exp\left(y_1 \alpha \left\langle \nabla f_1^{(t)}, \nabla \widehat{L}(W^{(t)})\right\rangle\right)}{\exp\left(y_2 \alpha \left\langle \nabla f_2^{(t)}, \nabla \widehat{L}(W^{(t)})\right\rangle\right)} \exp\left(\frac{H R_{\max}^2 \alpha^2}{\sqrt{m}} \|\nabla \widehat{L}(W^{(t)})\|^2\right) \\
&= A_t \cdot \frac{\exp\left(-\frac{\alpha}{n} \sum_{k=1}^n g_k^{(t)} \langle y_1 \nabla f_1^{(t)}, y_k \nabla f_k^{(t)} \rangle\right)}{\exp\left(-\frac{\alpha}{n} \sum_{k=1}^n g_k^{(t)} \langle y_1 \nabla f_2^{(t)}, y_k \nabla f_k^{(t)} \rangle\right)} \exp\left(\frac{H R_{\max}^2 \alpha^2}{\sqrt{m}} \|\nabla \widehat{L}(W^{(t)})\|^2\right) \\
&= A_t \cdot \exp\left(-\frac{\alpha}{n}\left(g_1^{(t)} \|\nabla f_1^{(t)}\|_F^2 - g_2^{(t)} \|\nabla f_2^{(t)}\|_F^2\right)\right) \\
&\quad \times \exp\left(-\frac{\alpha}{n}\left(\sum_{k \neq 2} g_k^{(t)} \langle y_2 \nabla f_2^{(t)}, y_k \nabla f_k^{(t)} \rangle - \sum_{k \neq 1} g_k^{(t)} \langle y_1 \nabla f_1^{(t)}, y_k \nabla f_k^{(t)} \rangle\right)\right) \\
&\quad \times \exp\left(\frac{H R_{\max}^2 \alpha^2}{\sqrt{m}} \|\nabla \widehat{L}(W^{(t)})\|^2\right).
\end{aligned}
\tag{17}
$$

Inequality $(i)$ uses Lemma E.2 while $(ii)$ uses the definition of $A_t$. We now proceed in a manner similar to Frei et al. (2022a) to bound each of the three terms in the product separately. For the first term, since gradient persistence (Condition E.6) holds at time $t$, we have for any $i \in [n]$,

$$
\|\nabla f_i^{(t)}\|_F^2 \geq c\|x_i\|^2 \geq c R_{\min}^2.
$$

On the other hand, since $\phi$ is 1-Lipschitz we also have

$$
\|\nabla f_i^{(t)}\|_F^2 = \|x_i\|^2 \frac{1}{m} \sum_{i=1}^m \phi'(\langle w_j^{(t)}, x_i \rangle)^2 \leq \|x_i\|^2 \leq R_{\max}^2.
$$

Putting the preceding two displays together, we get

$$
c R_{\min}^2 \leq \|\nabla f_i^{(t)}\|_F^2 \leq R_{\max}^2.
\tag{18}
$$

Therefore, we have

$$
\begin{aligned}
\exp\left(-\frac{\alpha}{n}\left(g_1^{(t)} \|\nabla f_1^{(t)}\|_F^2 - g_2^{(t)} \|\nabla f_2^{(t)}\|_F^2\right)\right) &= \exp\left(-\frac{g_2^{(t)} \alpha}{n}\left(\frac{g_1^{(t)}}{g_2^{(t)}} \|\nabla f_1^{(t)}\|_F^2 - \|\nabla f_2^{(t)}\|_F^2\right)\right) \\
&\overset{(i)}{\leq} \exp\left(-\frac{g_2^{(t)} \alpha}{n}\left(\frac{g_1^{(t)}}{g_2^{(t)}} \cdot c R_{\min}^2 - R_{\max}^2\right)\right) \\
&= \exp\left(-\frac{g_2^{(t)} \alpha c R_{\min}^2}{n}\left(\frac{g_1^{(t)}}{g_2^{(t)}} - \frac{R^2}{c}\right)\right).
\end{aligned}
\tag{19}
$$

Inequality $(i)$ uses (18), and the equality uses the definition $R = R_{\max}/R_{\min}$. This bounds the first term in (17).

For the second term, we use the fact that the gradients are nearly orthogonal at time $t$ (Condition E.5) and the lemma's assumption on $C'$ to get for any $i \neq k$,

$$
\|\nabla f_i^{(t)}\|_F^2 \geq C' n \max_{k \neq i} |\langle \nabla f_i^{(t)}, \nabla f_k^{(t)} \rangle| \geq (25 R^2/c + 25) n \max_{k \neq i} |\langle \nabla f_i^{(t)}, \nabla f_k^{(t)} \rangle|.
\tag{20}
$$

This allows for us to bound,

$$
\exp\left(-\frac{\alpha}{n}\left(\sum_{k\neq 2}g_k^{(t)}\langle y_2\nabla f_2^{(t)}, y_k\nabla f_k^{(t)}\rangle - \sum_{k\neq 1}g_k^{(t)}\langle y_1\nabla f_1^{(t)}, y_k\nabla f_k^{(t)}\rangle\right)\right)
$$

$$
\overset{(i)}{\leq} \exp\left(\frac{\alpha}{n}\sum_{k\neq 1}g_k^{(t)}|\langle\nabla f_1^{(t)}, \nabla f_k^{(t)}\rangle| + \frac{\alpha}{n}\sum_{k\neq 2}g_k^{(t)}|\langle\nabla f_2^{(t)}, \nabla f_k^{(t)}\rangle|\right)
$$

$$
\overset{(ii)}{\leq} \exp\left(\frac{\alpha}{n}\sum_{k\neq 1}g_k^{(t)}\cdot\frac{1}{(25R^2/c+25)n}\cdot\|\nabla f_1^{(t)}\|_F^2 + \frac{\alpha}{n}\sum_{k\neq 2}g_k^{(t)}\cdot\frac{1}{(25R^2/c+25)n}\cdot\|\nabla f_2^{(t)}\|^2\right)
$$

$$
\overset{(iii)}{\leq} \exp\left(\frac{\alpha}{n}\sum_{k\neq 1}g_k^{(t)}\cdot\frac{1}{(25R^2/c+25)n}\cdot R_{\max}^2 + \frac{\alpha}{n}\sum_{k\neq 2}g_k^{(t)}\cdot\frac{1}{(25R^2/c+25)n}\cdot R_{\max}^2\right)
$$

$$
\leq \exp\left(\frac{2\alpha R_{\max}^2}{(25R^2/c+25)n}\cdot\widehat{G}(W^{(t)})\right). \tag{21}
$$

Inequality $(i)$ uses the triangle inequality. Inequality $(ii)$ uses (20). The inequality $(iii)$ uses (18).

Finally, for the third term of (17), we have

$$
\exp\left(\frac{HR_{\max}^2\alpha^2}{\sqrt{m}}\|\nabla\widehat{L}(W^{(t)})\|^2\right) \overset{(i)}{\leq} \exp\left(\frac{HR_{\max}^4\alpha^2}{\sqrt{m}}\widehat{G}(W^{(t)})\right)
$$

$$
\overset{(ii)}{\leq} \exp\left(\frac{\alpha R_{\max}^2}{2(25R^2/c+25)n}\cdot\widehat{G}(W^{(t)})\right). \tag{22}
$$

Inequality $(i)$ uses Lemma E.3, while $(ii)$ uses the lemma's assumption that $\alpha$ is smaller than $[5HR_{\max}^2 n(10R^2/c+10)]^{-1}$. Putting (19), (21) and (22) into (17), we get

$$
A_{t+1} \leq A_t \cdot \exp\left(-\frac{g_2^{(t)}\alpha c R_{\min}^2}{n}\left(\frac{g_1^{(t)}}{g_2^{(t)}} - \frac{R^2}{c}\right)\right)
$$

$$
\times \exp\left(\frac{2\alpha R_{\max}^2}{(25R^2/c+25)n}\cdot\widehat{G}(W^{(t)})\right)
$$

$$
\times \exp\left(\frac{\alpha R_{\max}^2}{2(25R^2/c+25)n}\cdot\widehat{G}(W^{(t)})\right)
$$

$$
= A_t \cdot \exp\left(-\frac{g_2^{(t)}\alpha c R_{\min}^2}{n}\left(\frac{g_1^{(t)}}{g_2^{(t)}} - \frac{R^2}{c}\right)\right)\cdot\exp\left(\frac{5\alpha R_{\max}^2}{2(25R^2/c+25)n}\cdot\widehat{G}(W^{(t)})\right) \tag{23}
$$

This completes the proof. $\qquad\square$

Lemma E.8 shows that if the sigmoid loss ratio $g_i^{(t)}/g_j^{(t)}$ is large, then for a small-enough step-size, the exponential loss ratio will contract at the following interation. This motivates understanding how the exponential loss ratios relate to the sigmoid loss ratios. We recall the following fact, shown in Frei et al. (2022a, Fact A.2).

**Fact E.9.** *For any $z_1, z_2 \in \mathbb{R}$,*

$$
\frac{g(z_1)}{g(z_2)} \leq \max\left(2, 2\frac{\exp(-z_1)}{\exp(-z_2)}\right),
$$

*and if $z_1, z_2 > 0$, then we also have*

$$
\frac{\exp(-z_1)}{\exp(-z_2)} \leq 2\frac{g(z_1)}{g(z_2)}.
$$

This fact demonstrates that if we can ensure that the inputs to the losses is positive, then we can essentially treat the sigmoid and exponential losses interchangeably. Thus, if the network is able

to interpolate the training data at a given time $t$, we can swap the sigmoid loss ratio appearing in Lemma E.8 with the exponential loss, and argue that if the exponential loss is too large at a given iteration, it will contract the following one. This allows for the exponential losses to be bounded throughout gradient descent. We formalize this in the following lemma.

**Proposition E.10.** *Denote $R := R_{max}/R_{min}$ where $R_{max} = \max_i \|x_i\|$ and $R_{min} = \min_i \|x_i\|$. Let $\phi$ be an arbitrary 1-Lipschitz and $H$-smooth activation. Suppose that,*

- *Gradient persistence (Condition E.6) holds at time $t$ for some $c > 0$, and*

- *Near-orthogonality of gradients (Condition E.5) holds at time $t$ for some $C' > 25R^2/c + 25$,*

- *For some $\rho \geq 5R^2/c + 5$, an exponential loss ratio bound holds at time $t$ with,*

$$\max_{i,j} \frac{\exp\left(-y_i f(x_i; W^{(t)})\right)}{\exp\left(-y_j f(x_j; W^{(t)})\right)} \leq \rho.$$

- *The network interpolates the training data at time $t$: $y_i f(x_i; W^{(t)}) > 0$ for all $i$.*

*Then, provided the learning rate satisfies $\alpha \leq [5HR_{max}^2 n(10R^2/c + 10)]^{-1}$, we have an exponential loss ratio bound at time $t + 1$ as well,*

$$\max_{i,j} \frac{\exp\left(-y_i f(x_i; W^{(t+1)})\right)}{\exp\left(-y_j f(x_j; W^{(t+1)})\right)} \leq \rho.$$

*Proof.* As in the proof of Lemma E.8, it suffices to prove that the ratio of the exponential loss for the first sample to the exponential loss for the second sample is bounded by $\rho$. Let us again denote

$$A_t := \frac{\exp(-y_1 f(x_1; W^{(t)}))}{\exp(-y_2 f(x_2; W^{(t)}))},$$

and recall the notation $g_i^{(t)} := -\ell'(y_i f(x_i; W^{(t)}))$ and $\nabla f_i^{(t)} := \nabla f(x_i; W^{(t)})$. By Lemma E.8, we have,

$$A_{t+1} \leq A_t \cdot \exp\left(-\frac{g_2^{(t)} \alpha c R_{min}^2}{n}\left(\frac{g_1^{(t)}}{g_2^{(t)}} - \frac{R^2}{c}\right)\right) \cdot \exp\left(\frac{\alpha R_{max}^2}{(10R^2/c + 10)n} \cdot \widehat{G}(W^{(t)})\right) \quad (24)$$

We now consider two cases.

**Case 1:** $g_1^{(t)}/g_2^{(t)} \leq \frac{2}{5}\rho$. Continuing from (24), we have,

$$A_{t+1} \overset{(i)}{\leq} A_t \cdot \exp\left(\frac{g_2^{(t)} \alpha R_{min}^2 R^2}{n}\right) \cdot \exp\left(\frac{\alpha R_{max}^2}{(10R^2/c + 10)n}\widehat{G}(W^{(t)})\right)$$

$$= A_t \cdot \exp\left(\alpha \cdot \left(\frac{g_2^{(t)} R_{max}^2}{n} + \frac{R_{max}^2 \widehat{G}(W^{(t)})}{(10R^2/c + 10)n}\right)\right)$$

$$\overset{(ii)}{\leq} 1.2 A_t$$

$$\overset{(iii)}{\leq} 2.4 \frac{g_1^{(t)}}{g_2^{(t)}}$$

$$\overset{(iv)}{\leq} 2.4 \cdot \frac{2}{5}\rho \leq \rho.$$

Above, inequality $(i)$ follows since $g_1^{(t)}/g_2^{(t)} > 0$. The equality uses that $R = R_{max}/R_{min}$. Inequality $(ii)$ uses that $g_i^{(t)} < 1$, the lemma's assumption on the step-size, $\alpha \leq [5HR_{max}^2 n(10R^2/c + 10)]^{-1}$, and that $\exp(0.1) \leq 1.2$. The inequality $(iii)$ uses the proposition's assumption that the network interpolates the training data at time $t$, so that the ratio of exponential losses is at most twice the ratio of the sigmoid losses by Fact E.9. The final inequality $(iv)$ follows by the case assumption that $g_1^{(t)}/g_2^{(t)} \leq \frac{2}{5}\rho$.

**Case 2:** $g_1^{(t)}/g_2^{(t)} > \frac{2}{5}\rho.$ Continuing from (24), we have,

$$
A_{t+1} \leq A_t \cdot \exp\left(-\frac{g_2^{(t)}\alpha c R_{\min}^2}{n}\left(\frac{g_1^{(t)}}{g_2^{(t)}} - \frac{R^2}{c}\right)\right) \cdot \exp\left(\frac{\alpha R_{\max}^2}{(10R^2/c + 10)n} \cdot \widehat{G}(W^{(t)})\right)
$$

$$
\stackrel{(i)}{\leq} A_t \cdot \exp\left(-\frac{g_2^{(t)}\alpha c R_{\min}^2}{n}\cdot\left(\frac{2}{5}\rho - \frac{R^2}{c}\right)\right) \cdot \exp\left(\frac{\alpha R_{\max}^2}{(10R^2/c + 10)n} \cdot \widehat{G}(W^{(t)})\right)
$$

$$
= A_t \cdot \exp\left(-\frac{g_2^{(t)}\alpha c R_{\min}^2}{n}\cdot\left(\frac{2}{5}\rho - \frac{R^2}{c}\right)\right) \cdot \exp\left(\frac{\alpha R_{\max}^2}{(10R^2/c + 10)n} \cdot g_2^{(t)} \cdot \frac{1}{n}\sum_{i=1}^{n}\frac{g_i^{(t)}}{g_2^{(t)}}\right)
$$

$$
\stackrel{(ii)}{\leq} A_t \cdot \exp\left(-\frac{g_2^{(t)}\alpha c R_{\min}^2}{n}\cdot\left(\frac{2}{5}\rho - \frac{R^2}{c}\right)\right) \cdot \exp\left(\frac{\alpha R_{\max}^2}{(10R^2/c + 10)n} \cdot g_2^{(t)} \cdot 2\rho\right)
$$

$$
= A_t \exp\left(-\frac{g_2^{(t)}\alpha c R_{\min}^2}{n}\cdot\left(\frac{2}{5}\rho - \frac{R^2}{c} - \frac{R^2}{c}\cdot\frac{1}{5R^2/c + 5}\cdot\rho\right)\right)
$$

$$
\stackrel{(iii)}{\leq} A_t \leq \rho.
$$

Inequality $(i)$ uses the Case 2 assumption that $g_1^{(t)}/g_2^{(t)} > \frac{2}{5}\rho$. Inequality $(ii)$ uses the proposition's assumption that the exponential loss ratio at time $t$ is at most $\rho$, so that the sigmoid loss ratio is at most $2\rho$ by Fact E.9 (note that the sigmoid loss ratio is at least $2\rho/5 > 2$ by the case assumption and as $\rho > 5$). The equality uses that $R = R_{\max}/R_{\min}$. The final inequality $(iii)$ follows as we can write

$$
\frac{2}{5}\rho - \frac{R^2}{c} - \frac{R^2}{c}\cdot\frac{1}{5R^2/c + 5}\cdot\rho = \frac{2}{5}\rho\left(1 - \frac{1}{2}\cdot\frac{5R^2/c}{5(R^2/c + 1)}\right) - \frac{R^2}{c}
$$

$$
\geq \frac{2}{5}\rho\cdot\frac{1}{2} - \frac{R^2}{c}
$$

$$
> 0.
$$

The first inequality above uses that $|x/(1+x)| \leq 1$ for $x > 0$, and the final inequality follows by the assumption that $\rho \geq 5R^2/c + 5 > 5R^2/c$. This proves $(iii)$ above, so that in Case 2, the exponential loss ratio decreases at the following iteration.

$\square$

In summary, the preceding proposition demonstrates that a loss ratio bound can hold for general Lipschitz and smooth activations provided the following four conditions hold for some time $t_0$:

(1) an exponential loss ratio bound holds at time $t_0$;

(2) near-orthogonality of the gradients holds for all times $t \geq t_0$;

(3) gradient persistence holds at all times $t \geq t_0$; and

(4) the network interpolates the training data for all times $t \geq t_0$.

This is because the proposition guarantees that once you interpolate the training data, if the gradients are nearly-orthogonal and gradient persistence holds, the maximum ratio of the exponential losses does not become any larger than the maximum ratio at time $t_0$. Note that the above proof outline does not rely upon the training data being nearly orthogonal, nor that the activations are 'leaky', and thus may be applicable to more general settings than the ones we consider in this work.

On the other hand, when the training data is nearly-orthogonal and the activations are $\gamma$-leaky and $H$-smooth activations, Fact E.7 shows that (2) and (3) above hold for all times $t \geq 0$. Thus, to show a loss ratio bound in this setting, the main task is to show items (1) and (4) above. Towards this end, we present the final auxiliary lemma that will be used in the proof of Lemma E.4. A similar lemma appeared in Frei et al. (2022a, Lemma A.3), and our proof is only a small modification of their proof. For completeness, we provide its proof in detail here.

**Lemma E.11.** *Let $\phi$ be a $\gamma$-leaky, $H$-smooth activation. Then the following hold with probability at least $1 - \delta$ over the random initialization.*

(a) *An exponential loss ratio bound holds at initialization:*

$$\max_{i,j} \frac{\exp(-y_i f(x_i; W^{(0)}))}{\exp(-y_j f(x_j; W^{(0)}))} \leq \exp(2).$$

(b) *If there is an absolute constant $C'_R > 1$ such that at time $t$ we have $\max_{i,j}\{g_i^{(t)}/g_j^{(t)}\} \leq C'_R$, and if for all $k \in [n]$ we have*

$$\|x_k\|^2 \geq 2\gamma^{-2} C'_R n \max_{i \neq k} |\langle x_i, x_k \rangle|,$$

*then for $\alpha \leq \gamma^2/(2HC'_R R^2 R_{max}^2 n)$, we have*

$$\text{for all } k \in [n], \qquad y_k[f(x_k; W^{(t+1)}) - f(x_k; W^{(t)})] \geq \frac{\alpha \gamma^2 R_{min}^2}{4C'_R n} \widehat{G}(W^{(t)}).$$

(c) *If for all $k \in [n]$ we have $\|x_k\|^2 \geq 8\gamma^{-2} n \max_{i \neq k} |\langle x_i, x_k \rangle|$, then under Assumptions (A1) and (A2), at time $t = 1$ and for all samples $k \in [n]$, we have $y_k f(x_k; W^{(t)}) > 0$.*

*Proof.* We shall prove each part of the lemma in sequence.

**Part (a).** Since $\phi$ is 1-Lipschitz and $\phi(0) = 0$, Cauchy–Schwarz implies

$$|f(x; W)| = \left| \sum_{j=1}^{m} a_j \phi(\langle w_j, x \rangle) \right| \leq \sqrt{\sum_{j=1}^{m} a_j^2} \sqrt{\sum_{j=1}^{m} \langle w_j, x \rangle^2} = \|Wx\|_2.$$

Applying this bound to the network output for each sample at initialization, we get

$$|f(x_i; W^{(0)})| \leq \|W^{(0)}\|_F \|x_i\| \overset{(i)}{\leq} \sqrt{5}\omega_{\text{init}} \sqrt{md \log(4m/\delta)} R_{\max} \overset{(ii)}{\leq} \frac{\sqrt{5}\alpha R_{\max}^2}{72n} \overset{(iii)}{\leq} \frac{1}{50}. \quad (25)$$

Inequality $(i)$ uses Lemma E.1, while inequality $(ii)$ and $(iii)$ follow by Assumptions (A2) and (A1), respectively. We therefore have,

$$\max_{i,j=1,\ldots,n} \frac{\exp(-y_i f(x_i; W^{(0)}))}{\exp(-y_j f(x_j; W^{(0)}))} \leq \exp(2). \quad (26)$$

**Part (b).** Let $k \in [n]$. Let us re-introduce the notation $\nabla f_i^{(t)} := \nabla f(x_i; W^{(t)})$. By Lemma E.2, we know

$$y_k[f(x_k; W^{(t+1)}) - f(x_k; W^{(t)})] \geq \left[ \frac{\alpha}{n} \sum_{i=1}^{n} g_i^{(t)} \langle y_i \nabla f_i^{(t)}, y_k \nabla f_k^{(t)} \rangle \right] - \frac{HR_{\max}^2 \alpha^2}{2\sqrt{m}} \|\nabla \widehat{L}(W^{(t)})\|_2^2.$$

By definition,

$$\langle \nabla f_i^{(t)}, \nabla f_k^{(t)} \rangle = \langle x_i, x_k \rangle \cdot \underbrace{\frac{1}{m} \sum_{j=1}^{m} \phi'(\langle w_j^{(t)}, x_i \rangle) \phi'(\langle w_j^{(t)}, x_k \rangle)}_{\in [\gamma^2, 1]}. \quad (27)$$

We can thus calculate,

$$y_k[f(x_k; W^{(t+1)}) - f(x_k; W^{(t)})]$$

$$\overset{(i)}{\geq} \frac{\alpha}{n} \left[ \sum_{i=1}^{n} g_i^{(t)} \langle y_i \nabla f_i^{(t)}, y_k \nabla f_k^{(t)} \rangle - \frac{H R_{\max}^4 \alpha n}{2\sqrt{m}} \widehat{G}(W^{(t)}) \right]$$

$$= \frac{\alpha}{n} \left[ g_k^{(t)} \|\nabla f_k^{(t)}\|_F^2 + \sum_{i \neq k} g_i^{(t)} \langle y_i \nabla f_i^{(t)}, y_k \nabla f_k^{(t)} \rangle - \frac{H R_{\max}^4 \alpha n}{2\sqrt{m}} \widehat{G}(W^{(t)}) \right]$$

$$\geq \frac{\alpha}{n} \left[ g_k^{(t)} \|\nabla f_k^{(t)}\|^2 - \max_j g_j^{(t)} \sum_{i \neq k} |\langle \nabla f_i^{(t)}, \nabla f_k^{(t)} \rangle| - \frac{H R_{\max}^4 \alpha n}{2\sqrt{m}} \widehat{G}(W^{(t)}) \right]$$

$$= \frac{\alpha}{n} \left[ g_k^{(t)} \left( \|\nabla f_k^{(t)}\|^2 - \frac{\max_j g_j^{(t)}}{g_k^{(t)}} \sum_{i \neq k} |\langle \nabla f_i^{(t)}, \nabla f_k^{(t)} \rangle| \right) - \frac{H R_{\max}^4 \alpha n}{2\sqrt{m}} \widehat{G}(W^{(t)}) \right].$$

where Inequality $(i)$ uses Lemma E.3. Continuing we get that

$$y_k[f(x_k; W^{(t+1)}) - f(x_k; W^{(t)})]$$

$$\overset{(i)}{\geq} \frac{\alpha}{n} \left[ g_k^{(t)} \left( \|\nabla f_k^{(t)}\|^2 - C_R' \sum_{i \neq k} |\langle \nabla f_i^{(t)}, \nabla f_k^{(t)} \rangle| \right) - \frac{H R_{\max}^4 \alpha n}{2\sqrt{m}} \widehat{G}(W^{(t)}) \right]$$

$$\overset{(ii)}{\geq} \frac{\alpha}{n} \left[ g_k^{(t)} \cdot \left( \gamma^2 \|x_k\|^2 - C_R' \sum_{i \neq k} |\langle x_i, x_k \rangle| \right) - \frac{H R_{\max}^4 \alpha n}{2\sqrt{m}} \widehat{G}(W^{(t)}) \right]$$

$$\overset{(iii)}{\geq} \frac{\alpha}{n} \left[ g_k^{(t)} \cdot \frac{1}{2} \gamma^2 \|x_k\|^2 - \frac{H R_{\max}^4 \alpha n}{2\sqrt{m}} \widehat{G}(W^{(t)}) \right]$$

$$\overset{(iv)}{\geq} \frac{\alpha}{n} \left[ g_k^{(t)} \cdot \frac{1}{2} \gamma^2 R_{\min}^2 - \frac{H R_{\max}^4 \alpha n}{2\sqrt{m}} \widehat{G}(W^{(t)}) \right]$$

$$\overset{(v)}{\geq} \frac{\alpha}{n} \left[ \frac{\gamma^2 R_{\min}^2}{2 C_R'} \widehat{G}(W^{(t)}) - \frac{H R_{\max}^4 \alpha n}{2\sqrt{m}} \widehat{G}(W^{(t)}) \right]$$

$$\overset{(vi)}{\geq} \frac{\alpha \gamma^2 R_{\min}^2}{4 C_R' n} \widehat{G}(W^{(t)})$$

Inequality $(i)$ uses the lemma's assumption that $\max_{i,j}\{g_i^{(t)}/g_j^{(t)}\} \leq C_R'$. Inequality $(ii)$ uses that $\phi$ is $\gamma$-leaky and 1-Lipschitz (see eq. (27)). Inequality $(iii)$ uses that the assumption that the samples are nearly-orthogonal,

$$\|x_k\|^2 \geq 2\gamma^{-2} C_R' n \max_{i \neq k} |\langle x_i, x_k \rangle| \geq 2\gamma^{-2} C_R' \sum_{i \neq k} |\langle x_i, x_k \rangle|.$$

Inequality $(iv)$ uses the definition $R_{\min} = \min_i \|x_i\|$. Inequality $(v)$ again uses the lemma's assumption of a sigmoid loss ratio bound, so that

$$g_k^{(t)} = \frac{1}{n} \sum_{i=1}^{n} \frac{g_i^{(t)}}{g_k^{(t)}} g_k^{(t)} \geq \frac{1}{C_R'} \frac{1}{n} \sum_{i=1}^{n} g_i^{(t)} = \frac{1}{C_R'} \widehat{G}(W^{(t)}).$$

The final inequality $(vi)$ follows since the step-size $\alpha \leq \gamma^2/(2HC_R'R^2R_{\max}^2 n)$ is small enough. This completes part (b) of this lemma.

**Part (c).** Note that by (25), $|f(x_k; W^{(0)})| \leq 1/50$. Since $g$ is monotone this implies the sigmoid losses at initialization satisfy $g_i^{(0)} \in [(1 + \exp(0.02))^{-1}, (1 + \exp(-0.02))^{-1}] \subset [0.49, 0.51]$ and so

$$\widehat{G}(W^{(0)}) \geq \frac{49}{100}, \qquad \text{and} \qquad \max_{i,j} \frac{g_i^{(0)}}{g_j^{(0)}} \leq \frac{51}{49}. \tag{28}$$

Thus, the assumption that $\|x_k\|^2 \geq 8\gamma^{-2} n \max_{i \neq k} |\langle x_i, x_k \rangle|$ and Assumption (A1) allow for us to apply part (b) of this lemma as follows,

$$
\begin{aligned}
y_k f(x_k; W^{(1)}) &= y_k f(x_k; W^{(1)}) - y_k f(x_k; W^{(0)}) + f(x_k; W^{(0)}) \\
&\geq y_k f(x_k; W^{(1)}) - y_k f(x_k; W^{(0)}) - |f(x_k; W^{(0)})| \\
&\overset{(i)}{\geq} \frac{\alpha \gamma^2 R_{\min}^2}{4n \cdot {}^{51}/_{49}} \cdot \widehat{G}(W^{(0)}) - \sqrt{5} \omega_{\text{init}} \sqrt{md \log(4m/\delta)} R_{\max} \\
&\overset{(ii)}{\geq} \frac{\gamma^2 \alpha R_{\min}^2}{16n} - \sqrt{5} \omega_{\text{init}} \sqrt{md \log(4m/\delta)} R_{\max} \\
&= \frac{\gamma^2 \alpha R_{\min}^2}{16n} \left[ 1 - \frac{16\sqrt{5} \omega_{\text{init}} Rn \sqrt{md \log(4m/\delta)}}{\gamma^2 \alpha R_{\min}} \right] \\
&\overset{(iii)}{\geq} \frac{\gamma^2 \alpha R_{\min}^2}{32n}.
\end{aligned}
$$

The first term in inequality $(i)$ uses the lower bound provided in part (b) of this lemma as well as (28), while the second term uses the upper bound on $|f(x_k; W^{(0)})|$ in (25). Inequality $(ii)$ uses (28). Inequality $(iii)$ uses Assumption (A2) so that $\omega_{\text{init}} \leq \alpha \gamma^2 R_{\min} \cdot (72 R C_R n \sqrt{md \log(4m/\delta)})^{-1}$ and that $16\sqrt{5} < 36$. □

We now have all of the pieces necessary to prove Lemma E.4.

*Proof of Lemma E.4.* In order to show that the ratio of the $g(\cdot)$ losses is bounded, it suffices to show that the ratio of exponential losses $\exp(-(\cdot))$ is bounded, since by Fact E.9,

$$
\max_{i,j=1,\ldots,n} \frac{g(y_i f(x_i; W^{(t)}))}{g(y_j f(x_j; W^{(t)}))} \leq \max \left( 2, 2 \cdot \max_{i,j=1,\ldots,n} \frac{\exp(-y_i f(x_i; W^{(t)}))}{\exp(-y_j f(x_j; W^{(t)}))} \right). \tag{29}
$$

We will prove the lemma by first showing an exponential loss ratio holds at time $t = 0$ and $t = 1$, and then use an inductive argument based on Proposition E.10 with $\rho = 5R^2/\gamma^5 + 5 = \frac{1}{2} C_R$.

By part (a) of Lemma E.11, the exponential loss ratio at time $t = 0$ is at most $\exp(2)$. To see the loss ratio holds at time $t = 1$, first note that by assumption, we have that the samples satisfy,

$$
\|x_i\|^2 \geq 5\gamma^{-2} C_R n \max_{k \neq i} |\langle x_i, x_k \rangle| = 2\gamma^{-2}(25R^2\gamma^{-2} + 25) n \max_{k \neq i} |\langle x_i, x_k \rangle|. \tag{30}
$$

Because $\phi$ is a $\gamma$-leaky, $H$-smooth activation, by Fact E.7 this implies that gradient persistence (Condition E.6) holds with $c = \gamma^2$ and near-orthogonality of gradients (Condition E.5) holds for all times $t \geq 0$ with $C' > 2(25R^2\gamma^{-2} + 25)$. By Assumption (A1), we can therefore apply Lemma E.8 at time $t = 0$, so that we have for any $i, j$,

$$
\begin{aligned}
\frac{\exp\left(-y_i f(x_i; W^{(1)})\right)}{\exp\left(-y_j f(x_j; W^{(1)})\right)} &\leq \exp(2) \cdot \exp\left( -\frac{g_j^{(0)} \alpha c R_{\min}^2}{n} \left( \frac{g_i^{(0)}}{g_j^{(0)}} - \frac{R^2}{\gamma^2} \right) \right) \\
&\quad \times \exp\left( \frac{\alpha R_{\max}^2}{(10R^2/\gamma^2 + 10)n} \cdot \widehat{G}(W^{(0)}) \right) \\
&\overset{(i)}{\leq} \exp(2) \cdot \exp\left( \frac{R^2 R_{\min}^2 \alpha}{n} \right) \cdot \exp\left( \frac{\alpha R_{\max}^2}{(10R^2/\gamma^2 + 10)n} \right) \\
&= \exp(2) \cdot \exp\left( \alpha \left( \frac{R_{\max}^2}{n} + \frac{R_{\max}^2}{(10R^2/\gamma^2 + 10)n} \right) \right) \\
&\overset{(ii)}{\leq} \exp(2.1) \leq 9.
\end{aligned}
$$

Inequality $(i)$ uses that $g_i^{(t)} < 1$, while inequality $(ii)$ uses that the step-size is sufficiently small $\alpha \leq 1/20 R_{\max}^2$ by Assumption (A1). Therefore, the exponential loss ratio at times $t = 0$ and $t = 1$ is at most $9 \leq 5R^2/\gamma^2 + 5$.

Now suppose by induction that at times $\tau = 1, \ldots, t$, the exponential loss ratio is at most $5R^2/\gamma^2 + 5$, and consider $t+1$. (The cases $t = 0$ and $t = 1$ were just proved above.) By the induction hypothesis and (29), the sigmoid loss ratio from times $0, \ldots, t$ is at most $10R^2/\gamma^2 + 10$. By Assumption (A1), the step-size satisfies

$$\alpha \leq \gamma^2 [5nR_{\max}^2 R^2 (10R^2 \gamma^{-2} + 10) \max(1, H)]^{-1} \leq \gamma^2 [2HC_R R_{\max}^2 R^2 n]^{-1}.$$

Further, the samples satisfy (30), so that

$$\|x_k\|^2 \geq 2\gamma^{-2} (10R^2 \gamma^{-2} + 10) n \max_{i \neq k} |\langle x_i, x_k \rangle| = 2\gamma^{-2} C_R n \max_{i \neq k} |\langle x_i, x_k \rangle|.$$

Thus all parts of Lemma E.11 hold with $C_R' = C_R = 10R^2 \gamma^{-2} + 10$. By part (b) of that lemma, the unnormalized margin for each sample increased for every time $\tau = 0, \ldots, t$:

$$\text{for all } \tau = 1, \ldots, t, \quad y_k [f(x_k; W^{(\tau+1)}) - f(x_k; W^{(\tau)})] > 0. \tag{31}$$

Since the network interpolates the training data at time $t = 1$ by part (c) of Lemma E.11, this implies

$$\text{for all } \tau = 1, \ldots, t, \quad y_k f(x_k; W^{(\tau)}) > 0.$$

Finally, since the learning rate satisfies $\alpha \leq \gamma^2 [5nR_{\max}^2 R^2 C_R \max(1, H)]^{-1}$, all of the conditions necessary to apply Proposition E.10 hold. This proposition shows that the exponential loss ratio at time $t + 1$ is at most $5R^2/\gamma^2 + 5$. This completes the induction so that the exponential loss ratio is at most $5R^2/\gamma^2 + 5$ throughout gradient descent, which by (29) implies that the sigmoid loss ratio is at most $10R^2/\gamma^2 + 10$. $\qquad\square$

### E.4 Upper bound for the Frobenius norm

In this section we prove an upper bound for the Frobenius norm of the first-layer weights (recall that $\text{StableRank}(W) = \|W\|_F^2 / \|W\|_2^2$). The proof is a modification of Frei et al. (2022a, Lemma 4.10) to accommodate more general data. Note that the lemma is a strict improvement over the triangle inequality, as we are able to reduce the growth term by a factor of $1/\sqrt{n}$. The proof crucially relies upon the loss ratio bound proved in Lemma E.4.

**Lemma E.12.** *Let $R_{min} = \min_i \|x_i\|$, $R_{max} := \max_i \|x_i\|$, $R = R_{max}/R_{min}$, and denote $C_R = 10R^2/\gamma^2 + 10$ as the upper bound on the sigmoid loss ratio from Lemma E.4. Suppose that for all $i \in [n]$ the training data satisfy,*

$$\|x_i\|^2 \geq 5\gamma^{-2} C_R n \max_{k \neq i} |\langle x_i, x_k \rangle|.$$

*Then under Assumptions (A1) and (A2), with probability at least $1 - \delta$, for any $t \geq 1$,*

$$\|W^{(t)}\|_F \leq \|W^{(0)}\|_F + \frac{\sqrt{2C_R} R_{max} \alpha}{\sqrt{n}} \sum_{s=0}^{t-1} \widehat{G}(W^{(s)}).$$

*Proof.* We prove an upper bound on the $\ell_2$ norm of each neuron and then use this to derive an upper bound on the Frobenius norm of the first layer weight matrix. First note that the lemma's assumptions guarantee that Lemma E.4 holds. Next, by the triangle inequality, we have

$$\|w_j^{(t)}\| = \left\| w_j^{(0)} + \alpha \sum_{s=0}^{t-1} \nabla_j \widehat{L}(W^{(s)}) \right\|_F \leq \|w_j^{(0)}\| + \alpha \sum_{s=0}^{t-1} \|\nabla_j \widehat{L}(W^{(s)})\|_F. \tag{32}$$

We now consider the squared gradient norm with respect to the $j$-th neuron:

$$\|\nabla_j \widehat{L}(W^{(s)})\|^2$$

$$= \frac{1}{n^2} \left\| \sum_{i=1}^{n} g_i^{(s)} y_i \nabla_j f(x_i; W^{(s)}) \right\|^2$$

$$= \frac{1}{n^2} \left[ \sum_{i=1}^{n} \left( g_i^{(s)} \right)^2 \left\| \nabla_j f(x_i; W^{(s)}) \right\|^2 + \sum_{k \neq i \in [n]} g_i^{(s)} g_k^{(s)} y_i y_j \langle \nabla_j f(x_i; W^{(s)}), \nabla_j f(x_k; W^{(s)}) \rangle \right]$$

$$\leq \frac{1}{n^2} \left[ \sum_{i=1}^{n} \left( g_i^{(s)} \right)^2 \left\| \nabla_j f(x_i; W^{(s)}) \right\|^2 + \sum_{k \neq i \in [n]} g_i^{(s)} g_k^{(s)} \left| \langle \nabla_j f(x_i; W^{(s)}), \nabla_j f(x_k; W^{(s)}) \rangle \right| \right]$$

$$\overset{(i)}{\leq} \frac{a_j^2}{n^2} \left[ \sum_{i=1}^{n} \left( g_i^{(s)} \right)^2 \phi'(\langle w_j^{(t)}, x_i \rangle)^2 \|x_i\|^2 + \sum_{k \neq i \in [n]} g_i^{(s)} g_k^{(s)} \phi'(\langle w_j^{(t)}, x_i \rangle) \phi'(\langle w_j^{(t)}, x_k \rangle) |\langle x_i, x_k \rangle| \right]$$

$$\overset{(ii)}{\leq} \frac{a_j^2}{n^2} \left[ \sum_{i=1}^{n} \left( g_i^{(s)} \right)^2 \|x_i\|^2 + \sum_{k \neq i \in [n]} g_i^{(s)} g_k^{(s)} |\langle x_i, x_k \rangle| \right]$$

$$= \frac{a_j^2}{n^2} \sum_{i=1}^{n} \left( \left( g_i^{(s)} \right)^2 \left[ \|x_i\|^2 + \sum_{k \neq i} \frac{g_k^{(s)}}{g_i^{(s)}} |\langle x_i, x_k \rangle| \right] \right)$$

$$\overset{(iii)}{\leq} \frac{a_j^2}{n^2} \sum_{i=1}^{n} \left( \left( g_i^{(s)} \right)^2 \left[ \|x_i\|^2 + C_R \sum_{k \neq i} |\langle x_i, x_k \rangle| \right] \right)$$

$$\overset{(iv)}{\leq} \frac{2 a_j^2}{n^2} \sum_{i=1}^{n} \left( g_i^{(s)} \right)^2 \|x_i\|^2.$$

Above, inequality $(i)$ uses that $\nabla_j f(x_i; W) = a_j \phi'(\langle w_j, x_i \rangle) x_i$. Inequality $(ii)$ uses that $\phi$ is 1-Lipschitz. Inequality $(iii)$ uses the loss ratio bound in Lemma E.4, and inequality $(iv)$ uses the lemma's assumption about the near-orthogonality of the samples. We can thus continue,

$$\|\nabla_j \widehat{L}(W^{(s)})\|^2 \leq \frac{2 a_j^2}{n^2} \sum_{i=1}^{n} \left( g_i^{(s)} \right)^2 \|x_i\|^2$$

$$\leq \frac{2 a_j^2 R_{\max}^2}{n^2} \cdot \left( \max_k g_k^{(s)} \right) \cdot \sum_{i=1}^{n} g_i^{(s)}$$

$$= \frac{2 a_j^2 R_{\max}^2}{n} \cdot \left( \max_k g_k^{(s)} \right) \widehat{G}(W^{(s)})$$

$$\overset{(i)}{\leq} \frac{2 a_j^2 R_{\max}^2 C_R}{n} \left( \widehat{G}(W^{(s)}) \right)^2. \tag{33}$$

The final inequality uses the loss ratio bound so that we have

$$\max_{k \in [n]} g_k^{(s)} = \frac{1}{n} \sum_{i=1}^{n} \left( \frac{\max_k g_k^{(s)}}{g_i^{(s)}} g_i^{(s)} \right) \leq \frac{C_R}{n} \sum_{i=1}^{n} g_i^{(s)} = C_R \widehat{G}(W^{(s)}).$$

Finally, taking square roots of (33) and applying this bound on the norm in Inequality (32) above we conclude that

$$\|w_j^{(t)}\| \leq \|w_j^{(0)}\| + \frac{\sqrt{2 C_R} |a_j| R_{\max} \alpha}{\sqrt{n}} \sum_{s=0}^{t-1} \widehat{G}(W^{(s)}),$$

establishing our claim for the upper bound on $\|w_j^{(t)}\|$. For the bound on the Frobenius norm, we have an analogue of (32),

$$\|W^{(t)}\|_F \leq \|W^{(0)}\|_F + \alpha \sum_{s=0}^{t-1} \|\nabla \widehat{L}(W^{(s)})\|_F,$$

and we can simply use that $a_j^2 = 1/m$ and

$$\|\nabla\widehat{L}(W^{(s)})\|_F^2 = \sum_{j=1}^{m} \|\nabla_j\widehat{L}(W^{(s)})\|_F^2.$$

$\square$

### E.5 LOWER BOUND FOR THE SPECTRAL NORM

We next show that the spectral norm is large. The proof follows by showing that after the first step of gradient descent, every neuron is highly correlated with the vector $\widehat{\mu} := \sum_{i=1}^{n} y_i x_i$.

**Lemma E.13.** *Let* $R_{max} = \max_i \|x_i\|$, $R_{min} = \min_i \|x_i\|$ *and* $R := R_{max}/R_{min}$. *Let* $C_R = 10R^2\gamma^{-2} + 10$. *Suppose that for all* $i \in [n]$ *the training data satisfy,*

$$\|x_i\|^2 \geq 5\gamma^{-2}C_R n \max_{k \neq i} |\langle x_i, x_k\rangle|.$$

*Then, under Assumptions (A1) and (A2), we have with probability at least* $1 - \delta$, *we have the following lower bound for the spectral norm of the weights for any* $t \geq 1$:

$$\|W^{(t)}\|_2 \geq \frac{\alpha\gamma R_{min}}{4\sqrt{2}R\sqrt{n}} \sum_{s=0}^{t-1} \widehat{G}(W^{(s)}).$$

*Proof.* We shall show that every neuron is highly correlated with the vector $\widehat{\mu} := \sum_{i=1}^{n} y_i x_i$. By definition,

$$\langle w_j^{(t+1)} - w_j^{(t)}, \widehat{\mu}\rangle = \frac{\alpha a_j}{n} \sum_{i=1}^{n} g_i^{(t)} \phi'(\langle w_j^{(t)}, x_i\rangle) \left\langle y_i x_i, \sum_{k=1}^{n} y_k x_k\right\rangle$$

$$= \frac{\alpha a_j}{n} \sum_{i=1}^{n} g_i^{(t)} \phi'(\langle w_j^{(t)}, x_i\rangle) \left[\|x_i\|^2 + \sum_{k \neq i}\langle y_i x_i, y_k x_k\rangle\right].$$

**Positive neurons.** If $a_j > 0$, then we have,

$$\langle w_j^{(t+1)} - w_j^{(t)}, \widehat{\mu}\rangle \geq \frac{\alpha|a_j|}{n} \sum_{i=1}^{n} g_i^{(t)} \phi'(\langle w_j^{(t)}, x_i\rangle) \left[\|x_i\|^2 - \sum_{k \neq i}|\langle x_i, x_k\rangle|\right]$$

$$\overset{(i)}{\geq} \frac{\alpha|a_j|}{n} \sum_{i=1}^{n} g_i^{(t)} \phi'(\langle w_j^{(t)}, x_i\rangle) \cdot \frac{1}{2}\|x_i\|^2$$

$$\geq \frac{\alpha|a_j|R_{min}^2}{2n} \sum_{i=1}^{n} g_i^{(t)} \phi'(\langle w_j^{(t)}, x_i\rangle)$$

$$\overset{(ii)}{\geq} \frac{\alpha\gamma|a_j|R_{min}^2}{2} \widehat{G}(W^{(t)}).$$

Inequality $(i)$ uses the lemma's assumption that $\|x_i\|^2 \gg n \max_{k \neq i} |\langle x_i, x_k\rangle|$. Inequality $(ii)$ uses that $\phi$ is $\gamma$-leaky and $g_i^{(t)} \geq 0$. Telescoping, we get

$$\langle w_j^{(t)} - w_j^{(0)}, \widehat{\mu}\rangle \geq \frac{\alpha\gamma|a_j|R_{min}^2}{2} \sum_{s=0}^{t-1} \widehat{G}(W^{(s)}) = \frac{\alpha\gamma R_{min}^2}{2\sqrt{m}} \sum_{s=0}^{t-1} \widehat{G}(W^{(s)}). \tag{34}$$

We now show that we can ignore the $\langle w_j^{(0)}, \widehat{\mu}\rangle$ term by taking $\alpha$ large relative to $\omega_{\text{init}}$. By the calculation in (25), we know that $|f(x_i; W^{(0)})| \leq 1$ for each $i$ and thus

$$\widehat{G}(W^{(0)}) = \frac{1}{n} \sum_{i=1}^{n} \frac{1}{1 + \exp(-y_i f(x_i; W^{(0)}))} \geq 1/4. \tag{35}$$

On the other hand, by Lemma E.1, we know that

$$|\langle w_j^{(0)}, \widehat{\mu}\rangle| \le 2\omega_{\text{init}}\|\widehat{\mu}\|\sqrt{\log(4m/\delta)}.$$

By the lemma's assumption that $\|x_i\|^2 \gg n \max_{k\neq i}|\langle x_i, x_k\rangle|$, we have

$$\|\widehat{\mu}\|^2 = \sum_{i=1}^n \left[\|x_i\|^2 + \sum_{k:k\neq i}^n \langle y_i x_i, y_k x_k\rangle\right] \le \sum_{i=1}^n \left[\|x_i\|^2 + n\max_{k\neq i}|\langle x_i, x_k\rangle|\right] \le 2nR_{\text{max}}^2. \quad (36)$$

Substituting this inequality into the previous display, we get

$$|\langle w_j^{(0)}, \widehat{\mu}\rangle| \le 4R_{\text{max}}\omega_{\text{init}}\sqrt{n\log(4m/\delta)}. \quad (37)$$

We thus have

$$\begin{aligned}
\frac{\alpha\gamma R_{\text{min}}^2}{2\sqrt{m}}\widehat{G}(W^{(0)}) &\overset{(i)}{\ge} \frac{\alpha\gamma R_{\text{min}}^2}{8\sqrt{m}} \\
&\overset{(ii)}{\ge} 8R_{\text{max}}\omega_{\text{init}}\sqrt{n\log(4m/\delta)} \\
&\overset{(iii)}{\ge} 2|\langle w_j^{(0)}, \widehat{\mu}\rangle|. \quad (38)
\end{aligned}$$

where $(i)$ uses (35), $(ii)$ uses Assumption (A2) and that $C_R > 1$ so that,

$$\alpha \ge 64\omega_{\text{init}}\gamma^{-1}C_R(R_{\text{max}}/R_{\text{min}}^2)\sqrt{nm\log(4m/\delta)} = 64\omega_{\text{init}}\gamma^{-1}C_R(R/R_{\text{min}})\sqrt{nm\log(4m/\delta)},$$

and $(iii)$ uses (37). Continuing from (34) we get

$$\begin{aligned}
\langle w_j^{(t)}, \widehat{\mu}\rangle &\ge \langle w_j^{(t)} - w_j^{(0)}, \widehat{\mu}\rangle - |\langle w_j^{(0)}, \widehat{\mu}\rangle| \\
&\ge \frac{\alpha\gamma|a_j|R_{\text{min}}^2}{2}\sum_{s=0}^{t-1}\widehat{G}(W^{(s)}) - |\langle w_j^{(0)}, \widehat{\mu}\rangle| \\
&\ge \frac{\alpha\gamma|a_j|R_{\text{min}}^2}{4}\sum_{s=0}^{t-1}\widehat{G}(W^{(s)}), \quad (39)
\end{aligned}$$

where the last inequality uses (38).

**Negative neurons.** The argument in this case is essentially identical. If $a_j < 0$, then

$$\begin{aligned}
\langle w_j^{(t+1)} - w_j^{(t)}, \widehat{\mu}\rangle &\le -\frac{\alpha|a_j|}{n}\sum_{i=1}^n g_i^{(t)}\phi'(\langle w_j^{(t)}, x_i\rangle)\left[\|x_i\|^2 - \sum_{k\neq i}|\langle x_i, x_k\rangle|\right] \\
&\overset{(i)}{\le} -\frac{\alpha|a_j|R_{\text{min}}^2}{2n}\sum_{i=1}^n g_i^{(t)}\phi'(\langle w_j^{(t)}, x_i\rangle) \\
&\overset{(iii)}{\le} -\frac{\alpha\gamma|a_j|R_{\text{min}}^2}{2}\widehat{G}(W^{(t)}),
\end{aligned}$$

where the inequalities $(i)$ and $(ii)$ follow using an identical logic to the positive neuron case. We therefore have for negative neurons,

$$\langle w_j^{(t)} - w_j^{(0)}, \widehat{\mu}\rangle \le -\frac{\alpha\gamma R_{\text{min}}^2}{2\sqrt{m}}\sum_{s=0}^{t-1}\widehat{G}(W^{(s)}). \quad (40)$$

An identical argument used for the positive neurons to derive (39) shows a similar bound for $\langle w_j^{(t)}, \widehat{\mu}\rangle$.

To see the claim about the spectral norm, first note that since $R_{\min} > 0$, $|\langle w_j^{(t)}, \widehat{\mu} \rangle| > 0$ and hence $\widehat{\mu} \neq 0$. We thus can calculate,

$$
\begin{aligned}
\|W^{(t)}\|_2^2 &\overset{(i)}{\geq} \|W^{(t)}\widehat{\mu}/\|\widehat{\mu}\|\|_2^2 \\
&= \|\widehat{\mu}\|^{-2} \sum_{j=1}^{m} \langle w_j^{(t)}, \widehat{\mu} \rangle^2 \\
&\geq \|\widehat{\mu}\|^{-2} \sum_{j=1}^{m} \left( \frac{\alpha\gamma|a_j|R_{\min}^2}{4} \sum_{s=0}^{t-1} \widehat{G}(W^{(s)}) \right)^2 \\
&= \|\widehat{\mu}\|^{-2} \left( \frac{\alpha\gamma R_{\min}^2}{4} \sum_{s=0}^{t-1} \widehat{G}(W^{(s)}) \right)^2 \\
&\overset{(ii)}{\geq} \left( \frac{\alpha\gamma R_{\min}^2}{4\sqrt{2}R_{\max}\sqrt{n}} \sum_{s=0}^{t-1} \widehat{G}(W^{(s)}) \right)^2
\end{aligned}
$$

Inequality $(i)$ uses (34) and (40), and inequality $(ii)$ uses the upper bound for $\|\widehat{\mu}\|$ given in (36). This completes the proof. $\qquad\square$

### E.6 PROXY PL INEQUALITY

Our final task for the proof of Theorem 4.2 is to show that $\widehat{L}(W^{(t)}) \to 0$. We do so by establishing a variant of the Polyak–Lojasiewicz (PL) inequality called a *proxy PL inequality* (Frei & Gu, 2021, Definition 1.2).

**Lemma E.14.** *Let $R_{max} = \max_i \|x_i\|$, $R_{min} = \min_i \|x_i\|$, and $R := R_{max}/R_{min}$. Let $C_R = 10R^2\gamma^{-2} + 10$. Suppose the training data satisfy, for all $i \in [n]$,*

$$
\|x_i\|^2 \geq 5\gamma^{-2}C_R n \max_{k \neq i} |\langle x_i, x_k \rangle|.
$$

*For a $\gamma$-leaky activation, the following proxy-PL inequality holds for any $t \geq 0$:*

$$
\left\| \nabla \widehat{L}(W^{(t)}) \right\| \geq \frac{\gamma R_{min}}{2\sqrt{2}R\sqrt{n}} \widehat{G}(W^{(t)}).
$$

*Proof.* By definition, for any matrix $V \in \mathbb{R}^{m \times d}$ with $\|V\|_F \leq 1$ we have

$$
\|\nabla \widehat{L}(W)\| \geq \langle \nabla \widehat{L}(W), -V \rangle = \frac{1}{n} \sum_{i=1}^{n} g_i^{(t)} y_i \langle \nabla f(x_i; W), V \rangle.
$$

Let $\widehat{\mu} := \sum_{i=1}^{n} y_i x_i$ and define the matrix $V$ as having rows $a_j\widehat{\mu}/\|\widehat{\mu}\|$. Then, $\|V\|_F^2 = \sum_{j=1}^{m} a_j^2 = 1$, and we have for each $j$,

$$
\begin{aligned}
y_i \langle \nabla f(x_i; W), V \rangle &= \frac{1}{m} \sum_{j=1}^{m} \phi'(\langle w_j, x_i \rangle) \langle y_i x_i, \widehat{\mu}/\|\widehat{\mu}\| \rangle \\
&= \frac{1}{\|\widehat{\mu}\|m} \sum_{j=1}^{m} \phi'(\langle w_j, x_i \rangle) \left[ \|x_i\|^2 + \sum_{k \neq i} \langle y_i x_i, y_k x_k \rangle \right] \\
&\overset{(i)}{\geq} \frac{1}{\|\widehat{\mu}\|m} \sum_{j=1}^{m} \phi'(\langle w_j, x_i \rangle) \cdot \frac{1}{2}\|x_i\|^2 \\
&\overset{(ii)}{\geq} \frac{R_{\min}^2}{2\|\widehat{\mu}\|m} \sum_{j=1}^{m} \phi'(\langle w_j, x_i \rangle) \\
&\overset{(iii)}{\geq} \frac{\gamma R_{\min}^2}{2\|\widehat{\mu}\|}.
\end{aligned}
$$

Inequality $(i)$ uses the lemma's assumption that $\|x_i\|^2 \gg n \max_{k \neq i} |\langle x_i, x_k \rangle|$. Inequality $(ii)$ uses that $\|x_i\|^2 \geq R_{\min}^2$, and inequality $(iii)$ uses that $\phi'(z) \geq \gamma$. We therefore have,

$$\|\nabla \widehat{L}(W^{(t)})\| \geq \frac{1}{n} \sum_{i=1}^{n} g_i^{(t)} y_i \langle \nabla f(x_i; W^{(t)}), V \rangle$$

$$\geq \frac{\gamma R_{\min}^2}{2\|\widehat{\mu}\|} \widehat{G}(W^{(t)})$$

$$\geq \frac{\gamma R_{\min}^2}{2\sqrt{2} R_{\max} \sqrt{n}} \widehat{G}(W^{(t)}) = \frac{\gamma R_{\min}}{2\sqrt{2} R \sqrt{n}} \widehat{G}(W^{(t)}),$$

where the final inequality uses the calculation (36).

$\square$

### E.7  PROOF OF THEOREM 4.2

We are now in a position to provide the proof of Theorem 4.2. For the reader's convenience, we re-state the theorem below.

**Theorem 4.2.** *Suppose that $\phi$ is a $\gamma$-leaky, $H$-smooth activation. For training data $\{(x_i, y_i)\}_{i=1}^n \subset \mathbb{R}^d \times \{\pm 1\}$, let $R_{max} = \max_i \|x_i\|$ and $R_{min} = \min_i \|x_i\|$, and suppose $R = R_{max}/R_{min}$ is at most an absolute constant. Denote by $C_R := 10R^2/\gamma^2 + 10$. Assume the training data satisfies,*

$$R_{min}^2 \geq 5\gamma^{-2} C_R n \max_{i \neq j} |\langle x_i, x_j \rangle|.$$

*There exist absolute constants $C_1, C_2 > 1$ (independent of $m$, $d$, and $n$) such that the following holds. For any $\delta \in (0, 1)$, if the step-size satisfies $\alpha \leq \gamma^2(5nR_{max}^2 R^2 C_R \max(1, H))^{-1}$, and $\omega_{init} \leq \alpha \gamma^2 R_{min}(72RC_R n \sqrt{md \log(4m/\delta)})^{-1}$, then with probability at least $1 - \delta$ over the random initialization of gradient descent, the trained network satisfies:*

1. *The empirical risk under the logistic loss satisfies $\widehat{L}(W^{(t)}) \leq \sqrt{C_1 n / R_{min}^2 \alpha t}$ for $t \geq 1$.*

2. *The $\ell_2$ norm of each neuron grows to infinity: for all $j$, $\|w_j^{(t)}\|_2 \to \infty$.*

3. *The stable rank of the weights is bounded: $\sup_{t \geq 1} \left\{ \mathsf{StableRank}(W^{(t)}) \right\} \leq C_2$.*

*Proof.* We prove the theorem in parts.

**Empirical risk driven to zero.** This is a simple consequence of the proxy-PL inequality given in Lemma E.14 since $\phi$ is smooth; a small modification of the proof of Frei et al. (2022a, Lemma 4.12) suffices. In particular, since by Lemma E.3 the loss $\widehat{L}(w)$ has $R_{\max}^2(1 + H/\sqrt{m})$-Lipschitz gradients, we have

$$\widehat{L}(W^{(t+1)}) \leq \widehat{L}(W^{(t)}) - \alpha \|\nabla \widehat{L}(W^{(t)})\|_F^2 + R_{\max}^2 \max(1, H/\sqrt{m}) \alpha^2 \|\nabla \widehat{L}(W^{(t)})\|_F^2.$$

Applying the proxy-PL inequality of Lemma E.14 and using that $\alpha \leq [2\max(1, H/\sqrt{m})R_{\max}^2]^{-1}$ we thus have

$$\frac{\gamma^2 R_{\min}^2}{8R^2 n} \widehat{G}(W^{(t)})^2 \leq \|\nabla \widehat{L}(W^{(t)})\|_F^2 \leq \frac{2}{\alpha} \left[ \widehat{L}(W^{(t+1)}) - \widehat{L}(W^{(t)}) \right].$$

Telescoping the above, we get

$$\min_{t < T} \widehat{G}(W^{(t)})^2 \leq \frac{1}{T} \sum_{t=0}^{T-1} \widehat{G}(W^{(t)})^2 \leq \frac{2\widehat{L}(W^{(0)})}{\alpha T} \cdot \frac{8nR^2}{\gamma^2 R_{\min}^2}.$$

We know from the proof of Lemma E.4 (see (31)) that the unnormalized margin increases for each sample for all times. Since $g$ is monotone, this implies $\widehat{G}(W^{(t)})$ is decreasing and hence so is $\widehat{G}(W^{(t)})^2$, which implies

$$\widehat{G}(W^{(T-1)}) = \min_{t < T} \widehat{G}(W^{(t)}) \leq \sqrt{\frac{16\widehat{L}(W^{(0)}) n R^2}{\gamma^2 R_{\min}^2 \alpha T}}.$$

Since $\ell(z) \le 2g(z)$ for $z > 0$ and we know that the network interpolates the training data for all times $t \ge 1$, we know that $\widehat{L}(W^{(t)}) \le \widehat{2G}(W^{(t)})$ for $t \ge 1$, so that for $T \ge 2$,

$$\widehat{L}(W^{(T-1)}) \le \widehat{2G}(W^{(T-1)}) \le 2\sqrt{\frac{16\widehat{L}(W^{(0)})nR^2}{\gamma^2 R_{\min}^2 \alpha T}}.$$

**Norms driven to infinity.** We showed in Lemma E.13 (see (39) and (36)) that for each $t \ge 1$ and for each $j$,

$$\|w_j^{(t)}\| \ge \langle w_j^{(t)}, \widehat{\mu}/\|\widehat{\mu}\|\rangle \ge \frac{\alpha|a_j|R_{\min}^2}{4\sqrt{2}R_{\max}\sqrt{n}} \sum_{s=0}^{t-1} \widehat{G}(W^{(s)}).$$

It therefore suffices to show that $\sum_{s=0}^{t-1} \widehat{G}(W^{(s)}) \to \infty$. Suppose this is not the case, so that there exists some $\beta > 0$ such that $\sum_{s=0}^{t-1} \widehat{G}(W^{(s)}) \le \beta$ for all $t$. By Lemma E.12, this implies that for all $t$, $\|W^{(t)}\|_F \le \|W^{(0)}\|_F + 2\sqrt{C_R R_{\max}\alpha/n}\beta$. In particular, $\|W^{(t)}\|_F$ is bounded independently of $t$. But this contradicts the fact that $\widehat{L}(W^{(t)}) \to 0$ and $\ell > 0$ everywhere, and thus $\|w_j^{(t)}\| \gtrsim \sum_{s=0}^{t-1} \widehat{G}(W^{(s)}) \to \infty$.

**Stable rank is constant.** By definition,

$$\mathsf{StableRank}(W^{(t)}) = \frac{\|W^{(t)}\|_F^2}{\|W^{(t)}\|_2^2}.$$

We consider two cases.

**Case 1:** $\|W^{(t)}\|_F > 2\|W^{(0)}\|_F$. In this instance, by Lemma E.12, we have the chain of inequalities,

$$2\|W^{(0)}\|_F < \|W^{(t)}\|_F \le \|W^{(0)}\|_F + \frac{\sqrt{2C_R}R_{\max}\alpha}{\sqrt{n}} \sum_{s=0}^{t-1} \widehat{G}(W^{(s)}).$$

In particular, we have

$$\|W^{(0)}\|_F < \frac{\sqrt{2C_R}R_{\max}\alpha}{\sqrt{n}} \sum_{s=0}^{t-1} \widehat{G}(W^{(s)}).$$

We can thus use Lemma E.13 and Lemma E.12 to bound the ratio of the Frobenius norm to the spectral norm:

$$
\begin{aligned}
\frac{\|W^{(t)}\|_F}{\|W^{(t)}\|_2} &\le \frac{\|W^{(0)}\|_F + \frac{\sqrt{2C_R}R_{\max}\alpha}{\sqrt{n}} \sum_{s=0}^{t-1} \widehat{G}(W^{(t)})}{\frac{\alpha\gamma R_{\min}}{4\sqrt{2}R\sqrt{n}} \sum_{s=0}^{t-1} \widehat{G}(W^{(t)})} \\
&\le \frac{\frac{2\sqrt{2C_R}R_{\max}\alpha}{\sqrt{n}} \sum_{s=0}^{t-1} \widehat{G}(W^{(t)})}{\frac{\alpha\gamma R_{\min}}{4\sqrt{2}R\sqrt{n}} \sum_{s=0}^{t-1} \widehat{G}(W^{(t)})} \\
&= 16C_R^{1/2}R^2\gamma^{-1}.
\end{aligned}
\tag{41}
$$

**Case 2:** $\|W^{(t)}\|_F \le 2\|W^{(0)}\|_F$. Again using Lemma E.13, we have

$$
\begin{aligned}
\frac{\|W^{(t)}\|_F}{\|W^{(t)}\|_2} &\le \frac{2\|W^{(0)}\|_F}{\frac{\alpha\gamma R_{\min}}{4\sqrt{2}R\sqrt{n}} \sum_{s=0}^{t-1} \widehat{G}(W^{(t)})} \\
&\overset{(i)}{\le} \frac{\sqrt{5}\omega_{\mathrm{init}}\sqrt{md\log(4m/\delta)}}{\frac{\alpha\gamma R_{\min}}{4\sqrt{2}R\sqrt{n}}\widehat{G}(W^{(0)})} \\
&\overset{(ii)}{\le} \frac{4\sqrt{5}\omega_{\mathrm{init}}\sqrt{md\log(4m/\delta)}}{\frac{\alpha\gamma R_{\min}}{4\sqrt{2}R\sqrt{n}}} \\
&= 16\sqrt{10}C_R\gamma^{-1}RR_{\min}^{-1}\sqrt{n}\alpha^{-1}\omega_{\mathrm{init}}\sqrt{md\log(4m/\delta)} \\
&\overset{(iii)}{\le} \gamma/\sqrt{n} \le 16C_R^{1/2}R^2\gamma^{-1}.
\end{aligned}
\tag{42}
$$

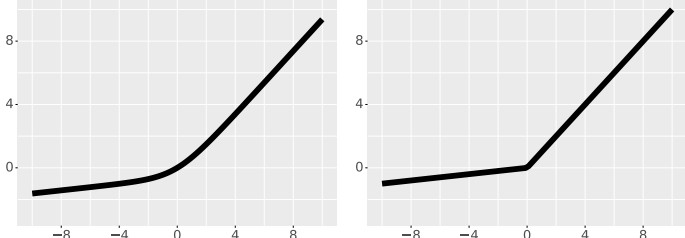

Figure 3: The 0.1-leaky, $\frac{1}{4}$-smooth leaky activation $\phi(z) = 0.1z + 0.9 \log\left(\frac{1}{2}(1 + \exp(z))\right)$ (left) and the standard leaky ReLU $\phi(z) = \max(0.1z, z)$ (right).

Inequality $(i)$ uses Lemma E.1. Inequality $(ii)$ uses that $\widehat{G}(W^{(0)}) \geq 1/4$ by the calculation (35). The final inequality $(iii)$ uses Assumption (A2) so that $\omega_{\text{init}} \leq \alpha \gamma^2 R_{\min}(72RC_R n\sqrt{md\log(4m/\delta)})^{-1}$. Thus, (42) yields the following upper bound for the stable rank,

$$\mathsf{StableRank}(W^{(t)}) \leq 16^2 C_R R^4 \gamma^{-2} = 16^2(10R^2/\gamma^2 + 10)R^4\gamma^{-2} =: C^*.$$

□

## F    EXPERIMENT DETAILS

We describe below the two experimental settings we consider.

### F.1    BINARY CLUSTER DATA

In Figure 1, we consider the binary cluster distribution described in (4). We consider a neural network with $m = 512$ neurons with activation $\phi(z) = \gamma z + (1 - \gamma)\log\left(\frac{1}{2}(1 + \exp(z))\right)$ for $\gamma = 0.1$, which is a 0.1-leaky, $1/4$-smooth leaky ReLU activation (see Figure 3). We fix $n = 100$ samples with mean separation $\|\mu\| = d^{0.26}$ with each entry of $\mu$ identical and positive. We introduce label noise by making 15% of the labels in each cluster share the opposing cluster label (i.e., samples from cluster mean $+\mu_1$ have label $+1$ with probability 0.85 and $-1$ with probability 0.15). Concurrent with the set-up in Section 4, we do not use biases and we keep the second layer fixed at the values $\pm 1/\sqrt{m}$, with exactly half of the second-layer weights positive and the other half negative. For the figure on the left, the initialization is standard normal distribution with standard deviation that is $50\times$ smaller than the TensorFlow default initialization, that is, $\omega_{\text{init}} = 1/50 \times \omega_{\text{init}}^{\mathsf{TF}}$ where $\omega_{\text{init}}^{\mathsf{TF}} = \sqrt{2/(m + d)}$. For the figure on the right, we fix $d = 10^4$ and vary the initialization standard deviation for different multiples of $\omega_{\text{init}}^{\mathsf{TF}}$, so that the variance is between $(10^{-2}\omega_{\text{init}}^{\mathsf{TF}})^2$ and $(10^2\omega_{\text{init}}^{\mathsf{TF}})^2$. For the experiment on the effect of dimension, we use a fixed learning rate of $\alpha = 0.01$, while for the experiment on the effect of the initialization scale we use a learning rate of $\alpha = 0.16$. In Figure 1, we show the stable rank of the first-layer weights scaled by the initial stable rank of the network (i.e., we plot $\mathsf{StableRank}(W^{(t)})/\mathsf{StableRank}(W^{(0)})$). The line shows the average over 5 independent random initializations with error bars (barely visible) corresponding to plus or minus one standard deviation.

In Figure 4, we provide additional empirical observations on how the learning rate can affect the initialization scale's influence on the stable rank of the trained network as we showed in Figure 1. We fix $d = 10^4$ and otherwise use the same setup for Figure 1 described in the previous paragraph. When the learning rate is the smaller value of $\alpha = 0.01$, training for longer can reduce the (stable) rank of the network, while for the larger learning rate of $\alpha = 0.32$ most of the rank reduction occurs in the first step of gradient descent.

In Figure 5, we examine the training accuracy, test accuracy, and stable rank of networks trained on the binary cluster distribution described above. Here we fix $d = 10^4$ and $\alpha = 0.01$ and otherwise use the same setup described in the first paragraph. We again consider two settings of the initialization scale: either a standard deviation of $\omega_{\text{init}}^{\mathsf{TF}}$ or $1/50 \times \omega_{\text{init}}^{\mathsf{TF}}$. We again see that the stable rank decreases much more rapidly when using a small initialization. Note that in both settings we observe a benign

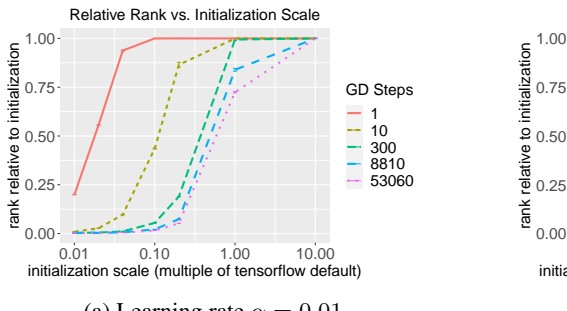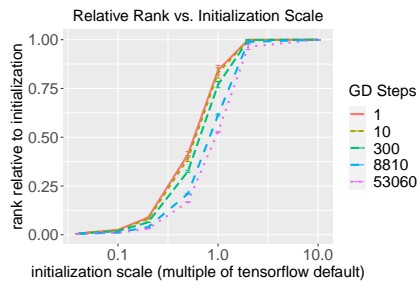

(a) Learning rate $\alpha = 0.01$      (b) Learning rate $\alpha = 0.32$

Figure 4: With larger learning rates, most of the rank reduction occurs in the first step of gradient descent. With smaller learning rates, training for longer can reduce the rank at most initialization scales.

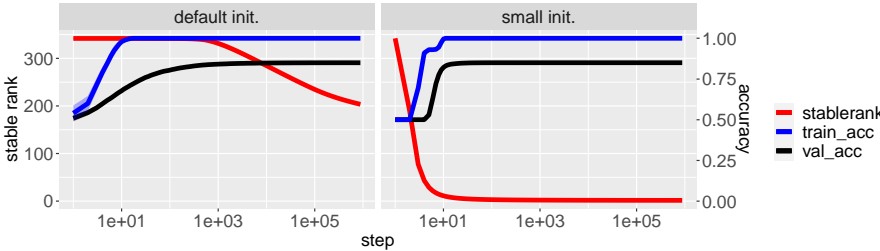

Figure 5: For the high-dimensional binary cluster data (cf. (4)), we see that using a small initialization scale leads to a rapid decrease in the stable rank of the network. A similar phenomenon occurs with CIFAR-10 (see Figure 2).

overfitting phenomenon as the training accuracy is 100% and the test accuracy is eventually the (optimal) 85%.

Finally, in Figure 6, we examine the training and test error of two-layer leaky ReLU networks trained by gradient descent with learning rate $\alpha = 0.01$ for the 2-XOR distribution described in Section 5 (with $n = 80$). We fix the number of neurons to $m = 512$. Theorem 3.2 suggests that if the leaky parameter $\gamma$ and $d$ are large enough relative to the number of samples, then the network will achieve a linear decision boundary. For the 2-XOR distribution, every classifier with a linear decision boundary achieves 50% test accuracy. We see that as $\gamma$ and $d$ increase, the test accuracy is indeed close to 50%, but for small $\gamma$ and $d$ the network achieves better performance and thus learns a nonlinear decision boundary.

### F.2 CIFAR10

We use the standard 10-class CIFAR10 dataset with pixel values normalized to be between 0 and 1 (dividing each pixel value by 255). We consider a standard two-layer network with 512 neurons with ReLU activations with biases and with second-layer weights trained. We train for $T = 10^6$ steps with SGD with batch size 128 and a learning rate of $\alpha = 0.01$. Figure 2 shows the average over 5 independent random initializations with shaded area corresponding to plus or minus one standard deviation.

For the second-layer initialization we use the standard TensorFlow Dense layer initialization, which uses Glorot Uniform with standard deviation $\sqrt{2/(m + 10)}$ (since the network has 10 outputs). For the first-layer initialization, we consider two different initialization schemes.

**Default initialization.** We use the standard Dense layer initialization in TensorFlow Keras. In this case the 'Glorot Uniform' initialization has standard deviation $\omega_{\text{init}}^{\mathsf{TF}} = \sqrt{2/(m + d)}$.

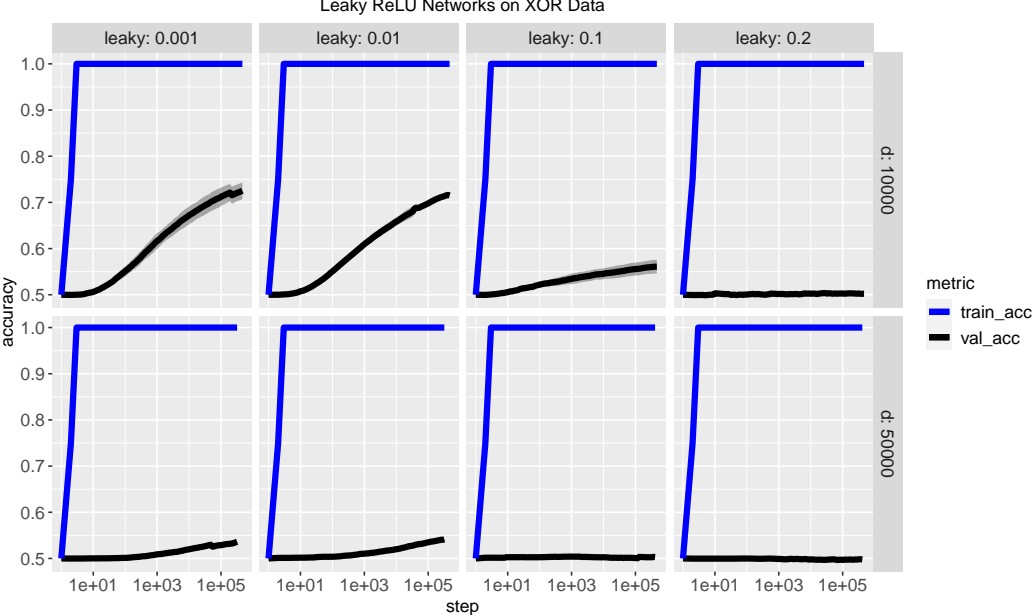

Figure 6: As $\gamma$ and $d$ increase, leaky ReLU networks trained by gradient descent fail to generalize well for the XOR distribution, as predicted by Theorem 3.2.

**Small initialization.** We use $\omega_{\mathrm{init}} = \omega_{\mathrm{init}}^{\mathsf{TF}}/50$.

