# OpenReview forum: "Implicit Bias in Leaky ReLU Networks Trained on High-Dimensional Data "
_ICLR.cc/2023/Conference — ICLR 2023 notable top 25%_

### Official Review · Reviewer_BzfX · 2022-10-23

**Confidence:** 4
**Correctness:** 4
**Technical Novelty And Significance:** 4
**Empirical Novelty And Significance:** 2
**Recommendation:** 10

**Clarity, Quality, Novelty And Reproducibility:**

As described above, the work is of high quality, very clear and has novel theoretical results.

**Strength And Weaknesses:**

Strengths:
1. Strong theoretical results in a challenging setup.
2. New theoretical techniques that might be useful in other settings.
3. Very well written.
4. Novel empirical findings.


Weaknesses:

Nothing major.
Two minor weaknesses:
1. The fact that “high dimensional data” is used throughout can be a bit of an over statement. For example, consider classifying MNIST digits, e.g., 0 digit vs. 1 digit. This problem is high dimensional, while I guess that the assumption in the paper is not satisfied. For example, there can be two data points of a 0 digit that have a large pairwise correlation. Maybe “random high dimensional data” is more accurate.
2. There are only a few experiments. There are many interesting experiments that can be included. For example, the theoretical finding on D_XOR is very interesting because usually networks solve the XOR problem. I guess that this is because of the high noise and low norm of the cluster centers. In any case, it would be interesting to see that the network indeed fails on this data empirically and converges to a linear classifier.




**Summary Of The Paper:**

This paper considers learning high dimensional linearly separable data with a one-hidden layer network with Leaky ReLU activations. The high dimensional data assumption entails that the norm of the data points is significantly larger than the pairwise correlation between different points. The assumption holds for random high dimensional data such as a Gaussian distribution. It is shown that a network where the first layer is trained with gradient flow has the following properties: (1) It converges to a zero training error solution (2) The solution has rank at most 2. (3) The solution has a linear decision boundary (4) It converges to a certain max margin solution which is not necessarily the max margin linear separator. Results for gradient descent are also presented where a bound on the stable rank is proved.

**Summary Of The Review:**

Strong and novel theoretical results in a challenging setting and very well written. Highly recommend for acceptance.

---

> ### Author Response · Authors · 2022-11-11
> **Author Response to Reviewer BzfX**
>
> We thank the reviewer for their very positive and encouraging review.
>
>
> Regarding the two points raised by the reviewer: we concur with the reviewer's opinion that we should be more careful with our usage of the word 'high-dimensional'.  As the reviewer notes, our usage comes from the fact that for random data, when $d$ is much larger than $n^2$, our results apply.  Similar to how 'high-dimensional' is used in the phrase 'high-dimensional statistics', we are using 'high-dimensional' to signify that we are in a regime where the number of samples is (much) smaller than the dimension.  In this meaning, MNIST is not considered `high-dimensional' since MNIST digits are 784 dimensional while the MNIST dataset has 60,000 (training) samples.  We have updated our manuscript to clarify our usage of the phrase 'high-dimensional'.
>
>
> Regarding the XOR experiments: we agree this is a quite interesting phenomenon, and the reviewer is correct that [previous work](https://arxiv.org/abs/2202.07626) has shown that two-layer nets with standard ReLU activations can generalize well for this distribution, even in the high-dimensional setting when $d\gg n$.  The usage of leaky ReLU, and the particular implicit bias towards linear decision boundaries that we identified when the dimension is particularly large relative to the leaky parameter, is the main reason for this behavior.  This is something that we observed experimentally but chose not to include in the submission, but we have revised the manuscript to include experiments  which verify that as the leaky parameter $\gamma$ increases and as the dimension increases, the network fails to generalize well for the XOR data problem (and also that the network interpolates the training data quickly).  See Figure 6 in the updated manuscript.

---

### Official Review · Reviewer_kRaJ · 2022-10-24

**Confidence:** 4
**Correctness:** 4
**Technical Novelty And Significance:** 3
**Empirical Novelty And Significance:** Not applicable
**Recommendation:** 6

**Clarity, Quality, Novelty And Reproducibility:**

Clarity:
- The paper is overall well-written and easy to follow.

Quality:
- The quality of paper is overall good. The results are clearly stated with proof ideas discussed in the main text and full proofs in the appendix.


Novelty:
- The results appear to be interesting and novel.

Reproducibility:
- This is a theoretical work so there are no experiments to be reproduced. The full proof are given in the appendix (I didn’t check them).


**Strength And Weaknesses:**

Strength:
- The paper is well-written and easy to follow.
- Understand the implicit bias of neural networks is an important problem and current paper provides an answer in the two-layer leaky-ReLU network with nearly orthogonal data setting.
- The results are clearly stated with proof ideas discussed in the main text and full proofs in the appendix.
- I found the section 5 that discusses the implication of the result quite interesting: it not only discusses the benefits of such implicit bias, but also gives examples where such implicit bias is harmful.

Weaknesses:
- As mentioned in the conclusion section by authors, the results heavily rely on leaky ReLU and cannot be applied to ReLU. Such leaky parameter $\gamma$ also has impact on the convergence point, such as point 7 in Theorem 3.2. It is unclear how to generalize the results to other activations and non-nearly-orthogonal data.


**Summary Of The Paper:**

In this paper, the authors studied the implicit bias of gradient flow and gradient descent for two-layer leaky-ReLU networks with nearly orthogonal data. It is shown that gradient flow will asymptotically converge to a network with rank at most 2 and linear decision boundary. Such a network is also approximately a max-margin linear predictor. For gradient descent, it is shown that with small enough initialization, a single step could reduce the rank of weight matrix to constant and the rank of weight matrix will remain constant throughout training. Experiments are also provided to support the theoretical results.

**Summary Of The Review:**

In summary, this work studied the implicit bias of two-layer leaky ReLU network with nearly orthogonal data. It shows that the convergence point has a low (stable) rank weight matrix and a linear decision boundary. The results seem to be new and interesting. Therefore, I’m currently leaning towards accept.

---

> ### Author Response · Authors · 2022-11-11
> **Author Response to Reviewer kRaJ**
>
> We thank the reviewer for their kind review.  We agree that understanding how the implicit bias of gradient descent changes when using a ReLU vs. leaky ReLU is an interesting question and we believe this is a natural direction for future research.  We have added experiments (see Fig. 6) that suggest the behavior is qualitatively different for ReLU.   Regarding extensions to not-nearly-orthogonal data, we believe the picture here is significantly more complicated as suggested by the experiments in Figure 2.
>
> If there are any additional points of contention that the reviewer has, we kindly ask that the reviewer let us know.  We hope that the reviewer may consider raising their score.

---

### Official Review · Reviewer_ewuA · 2022-10-24

**Confidence:** 4
**Correctness:** 4
**Technical Novelty And Significance:** 2
**Empirical Novelty And Significance:** 3
**Recommendation:** 8

**Clarity, Quality, Novelty And Reproducibility:**

The paper is clear, but the presentation can be improved. The quality of this paper looks fine, but some unclear "asumptions" on the data along with the treat of ReLU networks only undermine the contributions of the paper. Plus, lack of discussions of related works further makes it difficult to judge the novelty of the results.



**Strength And Weaknesses:**

Strengths: the paper characterizes the implicit bias of common gradient-based optimization algorithms for two-layer leaky ReLU networks trained on high-dimensional datasets. It presents that small initialization variance is important for gradient descent’s ability to quickly produce low-rank networks.
Weaknesses: The clear and precise definition of "high-dimensional data" is missing. What exactly are "high-dimensional data"? I understand random data in high-dimensions are almost always nearly orthogonal to each other. Please be more careful when stating the 'near-orthogonality' of data dealt with in the paper. Can we quantify the required "near-orthogonality" of data? Furthermore, the idea, analysis, and technical tools used in this paper are mainly adapted from the literature, in e.g., Lyu & Li (2019), Ji & Telgarsky (2020), Lyu et al. (2021). Third, I am not sure whether this so-called "implict bias" pertains to the network itself intrinsically, since if some of the assumptions on the data/gradient flow algorithms would make the "implict bias" not hold. Some existing works on two-layer ReLU or leaky ReLU learning are missing and closely related. See e.g., the landscaple and local/global optimality of (Leaky) ReLU networks in Laurent, et al, The multilinear structure of ReLU networks, in ICML, 2018; Laurent, et al, Deep linear networks with arbitrary loss: All local minima are global, in ICML, 2018; as well as the global optimality of SGD on training ReLU networks and the rank-2 solutions of regularized relu training in e.g., Wang, et al. Learning relu networks on linearly separable data: Algorithm, optimality, and generalization. IEEE TSP, 2019, and Yang, et al. Learning two-layer relu networks is nearly as easy as learning linear classifiers on separable data. IEEE TSP, 2021.  Ergen & Pilanci, (2020, June). Convex geometry of two-layer relu networks: Implicit autoencoding and interpretable models. In International Conference on Artificial Intelligence and Statistics. PMLR. Last, can the results generalzie to three-layer or deep ReLU networks of practical interest? Or please at least discuss its implications for training deep relu networks.

Comments:
i) This paper investigates the implicit bias of gradient flow and gradient descent in two-layer fully-connected neural networks with leaky ReLU activations when the training data are nearly-orthogonal. However, the paper still focuses on two-layer neural networks and also only the leaky ReLU activations. More comparison of the work in context shall be included, as well as its practical hints for ReLU networks and deeper networks.
ii) On Page 1, "We consider fully-connected two-layer networks with m neurons where the first layer weights are trained and the second layer weights are fixed at their random initialization." Although fixing the second-layer weights does seem to harm the representation of the 2-layer network, which is indeed used in most existing works, this (considerably) makes the learning problem much easier. Nonconvex optimization often comes from the symmetry or rotation, if one fixes the second-layer and breaks the symmetry/or elliminates the scaling issue. It would be much more meaningful to study the landscape and learning behaviour of the two-layer networks.
iii) Could you please list the pros and cons of your offline method compared with the online method in Lyu & Li (2019) and Ji & Telgarsky (2020)? In addition, could you explain how does different choices affect the result?
iv) In the experiments, this paper verifies the theoretical results in two-layer ReLU networks with bias terms trained by SGD only on CIFAR-10. More datasets and tests shall be conducted for validation.
v) There are some mistakes in grammar and sentences, such as "...and consider training that starts from a..." and "...when the labels y are some nonlinear function of t..." These indistinct expressions abate the readability of this paper.




**Summary Of The Paper:**

This paper is concerned with the implicit bias of gradient flow and gradient descent in two-layer fully connected neural networks with leaky ReLU activations when the training data are nearly-orthogonal. For gradient flow, this paper leverages the implicit bias for homogeneous neural networks to show that gradient flow produces a neural network with rank at most two. For gradient descent, this paper shows that a single step of gradient descent suffices to sufficiently reduce the rank of the network and the rank remains small throughout training. Finally, several experiments are provided to verify the effectiveness of the proposed method.


**Summary Of The Review:**

This paper is concerned with the implicit bias of gradient flow and gradient descent in two-layer fully connected neural networks with leaky ReLU activations when the training data are nearly-orthogonal. I believe the paper shalle be improved in providing a detailed comparison of related works on assumptions on the data, activation functions, global/local optimality, and other interesting findings. Overall, the contribution of the present paper is below the bar for ICLR.
------------------------------------------------------
I have looked at the the authors' replies as well as the comments of other reviewers. I agree that the paper contributes new ideas and technical proofs for understanding training of leaky ReLU networks. I have updated my score. It would be good to see all the corrections in the revision.

---

> ### Author Response · Authors · 2022-11-11
> **Author Response to Reviewer ewuA**
>
> We thank the reviewer for their effort.  The reviewer has listed a number of criticisms of our work but we find it difficult to ascertain the fundamental reason for their severely negative score.   Below, we try to respond fully to what we believe are the main points of disagreement.  We hope the reviewer will consider increasing their score.
>
>
> ## Technical contributions
>
> We strongly disagree with the reviewer's assertion that the ''idea, analysis, and technical tools used in this paper are mainly adapted from the literature, in e.g., Lyu \& Li (2019), Ji \& Telgarsky (2020), Lyu et al. (2021)''.  We are not sure what the reviewer is referring to.  It would be helpful if the reviewer could point us towards specific ideas or tools from these works.  We use Lyu and Li (2019) and Ji and Telgarsky (2020) to establish that gradient flow converges in direction to KKT points.  But this is only used to deduce Corollary 3.5 from Theorem 3.2 and 3.4 (note also that Theorem 4.2 does not use any aspect of the aforementioned works, since the activation functions in Section 4 are not necessarily homogeneous).   The tools used to prove Theorem 3.2 and Theorem 3.4 are original contributions of this work and do not use any tools from these works.  Our proof technique is also orthogonal to the work Lyu et al. (2021): their setting is one of ''symmetric'' datasets (which we note do not include high-dimensional Gaussian data such as the setting of Lemma 3.3).
>
>
> ## ''High-dimensonality'' and ''near-orthogonality''
>
> We agree that we can be more explicit about the precise meaning of the phrases ''near-orthogonality'' and ''high-dimensionality'' when used in the main text, and we have updated our manuscript to do so.
>
> We are not sure what the reviewer is referring to when they say `Can we quantify the required ''near-orthogonality'' of data?': in both Theorem 3.2 and Theorem 4.2, we are explicit about the precise conditions needed for our theorems to apply.  We have exact constants on how large the norms of samples must be ($R_{\mathrm{min}}^2$) relative to the maximum pairwise correlations between such samples.  In Lemma 3.3, we show that provided the dimension is large enough relative to the number of samples, well-conditioned Gaussians satisfy this condition with high probability.  This is where the phrase ''high-dimensional data'' comes from.  We have updated our manuscript to make this point more clear.
>
> ## Extension to deeper networks of ''practical interest''
>
> We agree that understanding deep ReLU networks is an important research problem.  However, we believe it is natural to develop a deeper theoretical understanding of two-layer neural networks before beginning to understand deeper networks, and we believe our work provides a significant contribution in this direction.
>
>
> ## Fixing second layer weights
>
> The reviewer asserts that fixing the second layer weights ''(considerably) makes the learning problem much easier. Nonconvex optimization often comes from the symmetry or rotation, if one fixes the second-layer and breaks the symmetry/or elliminates the scaling issue. It would be much more meaningful to study the landscape and learning behaviour of the two-layer networks.''
> We are having difficulty understanding the reviewer's criticism here.   Fixing the second layer weights does not eliminate symmetry inherent to the problem: there is still significant symmetry among 'positive' neurons and among 'negative' neurons.  We are not sure what ''scaling'' issue the reviewer is referring to.  We believe it is by now a well-established and accepted simplification to restrict to training first-layer weights, as this allows for all of the representational power of two-layer neural networks and allows for feature-learning and non-convex training dynamics.
>
> ## ''Offline method'' compared to ''online method''
>
> We are unsure what ''online'' method the reviewer is referring to.  Both Lyu \& Li (2019) and Ji \& Telgarsky (2020) only analyze gradient flow and gradient descent in the ''offline'' setting.  Neither provide an analysis of ''online'' methods (like online stochastic gradient descent) as the reviewer implies.
>
>
> ## Related works
>
> We thank the reviewer for pointing us towards these related works.
> We have updated our related work section.  We have added some of the works suggested by the reviewer, and we have put a greater emphasis on those works most closely related to ours, i.e., those related to the training dynamics of neural networks for linearly separable data.
>
> ## The terminology 'implicit bias'
>
>  The `implicit bias' refers to a property of an optimization algorithm when trained over a particular class of functions, and does not refer solely to the class of functions.  We believe this is standard terminology.

---

### Official Review · Reviewer_MUCK · 2022-10-24

**Confidence:** 4
**Clarity, Quality, Novelty And Reproducibility:** The paper is well-written.
**Correctness:** 4
**Technical Novelty And Significance:** 3
**Empirical Novelty And Significance:** 3
**Recommendation:** 8

**Strength And Weaknesses:**

This paper provides interesting and accurate characterizations of the implicit bias of gradient flow/descent with nearly-orthogonal data and leaky-ReLU activation. Specifically, although the network can be highly over-parameterized, this paper shows that the classical or stable rank stays small. One weakness is that the near orthogonality condition is a little restrictive, as it implies linear separability. Lemma 3.3 shows that it holds with Gaussian inputs, but it requires $d\ge n^2$, which does not hold in practice. Can you discuss how this assumption is used in your proof, and why linear separability is not enough?

**Summary Of The Paper:**

This paper analyzes the implicit bias of gradient flow/descent on a two-layer leaky-ReLU network. Specifically, assuming the data points are nearly orthogonal, Theorem 3.2 gives an accurate characterization of the KKT point of the margin maximization problem: all data points are support vectors, the first layer is of rank 2, and a closed-form solution is given. In Theorem 3.4, it is further shown that gradient flow converges to this solution. Section 4 focuses on gradient descent, and shows that with nearly orthogonal data and smoothed leaky-ReLU activation, the stable rank of the network is bounded by a universal constant. Empirical supports are also provided.

**Summary Of The Review:**

This paper gives interesting results on the implicit bias of gradient descent with nearly-orthogonal data and leaky-ReLU activation. I recommend for acceptance.

---

> ### Author Response · Authors · 2022-11-11
> **Author Response to Reviewer MUCK**
>
> We thank the reviewer for their kind review.
>
> Regarding the assumption about near-orthogonality to the data and its use in the proof.  The assumption is used crucially in a number of places, both in the gradient flow analysis and in the gradient descent analysis.  For gradient flow, we show that every sample satisfies the KKT constraints with an equality (item 1 in Theorem 3.2).  This generally does not happen for linearly separable data: if we consider linear classifiers and have data $S = \\{ (a, +1), (b, +1) \\}$ with $b>a>0$, then $(a,+1)$ will be a support vector but $(b,+1)$ will not.  A similar analysis shows that not all KKT constraints in leaky ReLU nets will be satisfied with an equality for linearly separable data.  For gradient descent, the ``loss ratio bound'' of Lemma E.4 will not generally hold for linearly separable data using a similar argument.  For that dataset $S$, the losses on the sample $(a,+1)$ will be much larger than the losses on the sample $(b, +1)$ throughout the gradient descent trajectory, since only $(a,+1)$ will be a support vector.
> We added a note on this issue to the manuscript (see the conclusion).

---

### Decision · Program_Chairs · 2023-01-20

**Decision:**

Accept: notable-top-25%

**Justification For Why Not Higher Score:**

Overall the quality of the paper is high. However, the assumption in the paper is a bit restrictive. If such restrictions were relaxed, this paper would deserve oral presentation.

**Justification For Why Not Lower Score:**

This paper gives fairly high quality theoretical analysis. The implicit bias of the gradient flow is one of the biggest issues in the literature and it plays an important role to analyze the effectiveness of feature learning. This paper gives a precise and clear characterization on this topic. Hence, I think this paper deserves the spotlight presentation.

**Metareview: Summary, Strengths And Weaknesses:**

This paper gives an accurate characterization of the implicit bias induced by the gradient flow to train two layer leaky-ReLU networks. The gradient flow induces a small stable rank (almost rank 2) at an early stage of the optimization and converges to a solution with an approximate-max-margin. In particular, it is shown that the small scale of initialization is important to have small stable rank implicit bias in the beginning of the optimization.

This paper gives a precise characterization of the implicit bias of the gradient flow in a high dimensional setting where the inputs are nearly orthogonal. It is also important to show the small stable rank implicit bias. This kind of phenomenon has also been pointed out by the existing literature in different settings, but this research gives another (important) example to the literature. Overall, the writing is good.
One of the main drawbacks of this paper is that it assumes near orthogonality of the input, which is a bit restrictive. Another weakness is its numerical experiment. The experiment is conducted on a toy synthetic dataset. However, these drawbacks do not hurt the value of this paper.

In summary, this paper gives a new and interesting theoretical work for the implicit bias of 2 layer neural networks. The theory is solid. The writing is good. Hence, I recommend acceptance in ICLR.



**Note From Pc:**

if the above contains the word "oral" or "spotlight" please see: "oral" presentation means -> notable-top-5% and "spotlight" means -> notable-top-25%. As stated in our emails, we are disassociating presentation type from AC recommendations